# Knowledge-guided machine learning can improve carbon cycle quantification in agroecosystems

Licheng Liu [1,13], Wang Zhou [2,3,13], Kaiyu Guan [2,3,4,5 ✉], Bin Peng [2,3], Shaoming Xu[6], Jinyun Tang [7], Qing Zhu [7], Jessica Till[1], Xiaowei Jia[8], Chongya Jiang[2,3], Sheng Wang [2,3,9], Ziqi Qin[2,3], Hui Kong[10], Robert Grant [11], Symon Mezbahuddin[11,12], Vipin Kumar[6] & Zhenong Jin [1 ✉]

Accurate and cost-effective quantification of the carbon cycle for agroecosystems at decision-relevant scales is critical to mitigating climate change and ensuring sustainable food production. However, conventional process-based or data-driven modeling approaches alone have large prediction uncertainties due to the complex biogeochemical processes to model and the lack of observations to constrain many key state and flux variables. Here we propose a Knowledge-Guided Machine Learning (KGML) framework that addresses the above challenges by integrating knowledge embedded in a process-based model, high-resolution remote sensing observations, and machine learning (ML) techniques. Using the U.S. Corn Belt as a testbed, we demonstrate that KGML can outperform conventional process-based and black-box ML models in quantifying carbon cycle dynamics. Our high-resolution approach quantitatively reveals 86% more spatial detail of soil organic carbon changes than conventional coarse-resolution approaches. Moreover, we outline a protocol for improving KGML via various paths, which can be generalized to develop hybrid models to better predict complex earth system dynamics.

Crop production systems and their interactions with the environment, known as agroecosystems, cover about one-third of the Earth's land surface. As soil constitutes the largest single carbon reservoir on land, agroecosystems play a key role in the global terrestrial carbon cycle through crop interactions with soils and atmosphere[1,2]. Globally, agriculture is a significant source of greenhouse gasses (GHGs); yet, carbon uptake by crops also removes large amounts of carbon dioxide ($CO_2$) from the atmosphere, some of which can be stabilized in soil[3]. Because most intensively cultivated soils are carbon-unsaturated, practices that increase soil organic carbon (SOC) represent a low-cost, large-scale strategy for reducing atmospheric GHG concentrations[4–6]. Thus, it is essential to accurately quantify carbon fluxes and changes in SOC in

[1]Department of Bioproducts and Biosystems Engineering, University of Minnesota, St. Paul, MN 55108, USA. [2]Agroecosystem Sustainability Center, Institute for Sustainability, Energy, and Environment, University of Illinois at Urbana-Champaign, Urbana, IL 61801, USA. [3]Department of Natural Resources and Environmental Sciences, College of Agricultural, Consumer and Environmental Sciences, University of Illinois at Urbana-Champaign, Urbana, IL 61801, USA. [4]Department of Computer Science, University of Illinois at Urbana-Champaign, Urbana, IL 61801, USA. [5]National Center for Supercomputing Applications, University of Illinois at Urbana-Champaign, Urbana, IL 61801, USA. [6]Department of Computer Science and Engineering, University of Minnesota, Minneapolis, MN 55455, USA. [7]Earth and Environmental Sciences Area, Lawrence Berkeley National Laboratory, Berkeley, CA 94720, USA. [8]Department of Computer Science, University of Pittsburgh, Pittsburgh, PA 15260, USA. [9]Department of Agroecology, Aarhus University, 4200 Slagelse, Denmark. [10]Humphrey School of Public Affairs, University of Minnesota, Twin Cities, MN 55455, USA. [11]Department of Renewable Resources, University of Alberta, Edmonton, AB T6G2E3, Canada. [12]Environmental Knowledge and Prediction Branch, Alberta Environment and Protected Areas, Edmonton, AB T5K 2J6, Canada. [13]These authors contributed equally: Licheng Liu, Wang Zhou. ✉e-mail: kaiyug@illinois.edu; jinzn@umn.edu

agroecosystems so that appropriate and effective conservation practices can be identified for any given location.

Increasing agricultural carbon sequestration is a key strategy for mitigating climate change. Significant efforts and investments have been made in the U.S. and across the globe to implement programs that incentivize SOC enrichment[7,8]. In light of these initiatives, it is important to develop robust and scalable methods for reliably quantifying field-level carbon sequestration, both to assess the climate mitigation effect and to ensure that mitigation actions by individual farmers are compensated fairly and accurately. Traditional carbon quantification methods that rely on soil sampling, emission factors, and process-based (PB) modeling entail inherent barriers to achieving the required levels of accuracy, scalability, and cost-effectiveness[9–11]. In particular, high spatial heterogeneity and seasonality due to variations in environmental conditions, crop types, and management practices present challenges for accurately quantifying carbon budgets[12]. While PB modeling approaches incorporate scientific knowledge, large uncertainties arise in PB models if local- and crop-specific parameters are not calibrated properly or if the underlying mechanisms are oversimplified or incompletely represented[12,13]. Additionally, PB models with detailed representations of existing scientific principles can be computationally prohibitive when applied to large regions at high spatial-temporal resolution (e.g., 250 m daily). On the other hand, data-driven machine learning (ML) approaches have the potential for high computational efficiency and accuracy[14–17] but suffer from out-of-sample prediction failure in the absence of large training datasets, which are unavailable for most agricultural applications. Moreover, the results of ML models are often uninterpretable due to their black-box nature[18]. Therefore, new methods[19] are needed to overcome the limitations of PB and ML models, enabling cost-effective, accurate, and interpretable measurement and monetization of carbon outcomes at the individual field level. This will reduce errors in aggregated quantifications and promote more sustainable land management practices[12,20].

The growing field of knowledge-guided machine learning (KGML)[21,22] provides a promising methodology that combines the advantages of PB models, ML models, and multi-source datasets (e.g., in-situ and remote sensing data). Existing KGMLs can successfully model certain Earth systems for which dynamic processes are well-represented by established governing equations, such as hydrology and atmospheric sciences[14,22–26]. However, complex and crucial ecosystem processes such as biogeochemical cycling are mathematically non-linear and substantially more complicated. Furthermore, unlike surficial systems, soil interactions in agroecosystems largely cannot be directly observed by remote sensing, whereas in-situ direct measurements are often expensive and limited. Therefore new KGML approaches must be developed to incorporate sufficient biogeochemical knowledge and effectively assimilate indirect measurements (e.g., remote sensing and survey data) to capture terrestrial processes that are less directly observable[27,28].

To address the existing gaps in carbon budget modeling capabilities, we developed a novel KGML framework that combines prior biogeochemical knowledge of carbon dynamics with a deep learning model to generate reliable predictions of agricultural carbon fluxes, crop yields, and changes in soil carbon stocks (KGML-ag-Carbon, Fig. 1). In-situ eddy covariance (EC) flux tower data, regional survey yield data, remotely-sensed gross primary production (GPP) data, and synthetic data generated by a PB model were assimilated into KGML-ag-Carbon. The model effectiveness is demonstrated here for corn and soybean production in the U.S. Midwest (Fig. S1), with highly accurate outputs for carbon fluxes, crop yields, and changes in soil carbon at high spatial (250 m) and temporal (daily) resolution, providing usable data for land managers. We also analyzed the improvement resulting from each KGML component and certain biogeochemical responses. The model design presented here exemplifies a solution to challenges in simulating dynamic heterogeneous systems, which will help advance broader applications of KGML for understanding Earth processes.

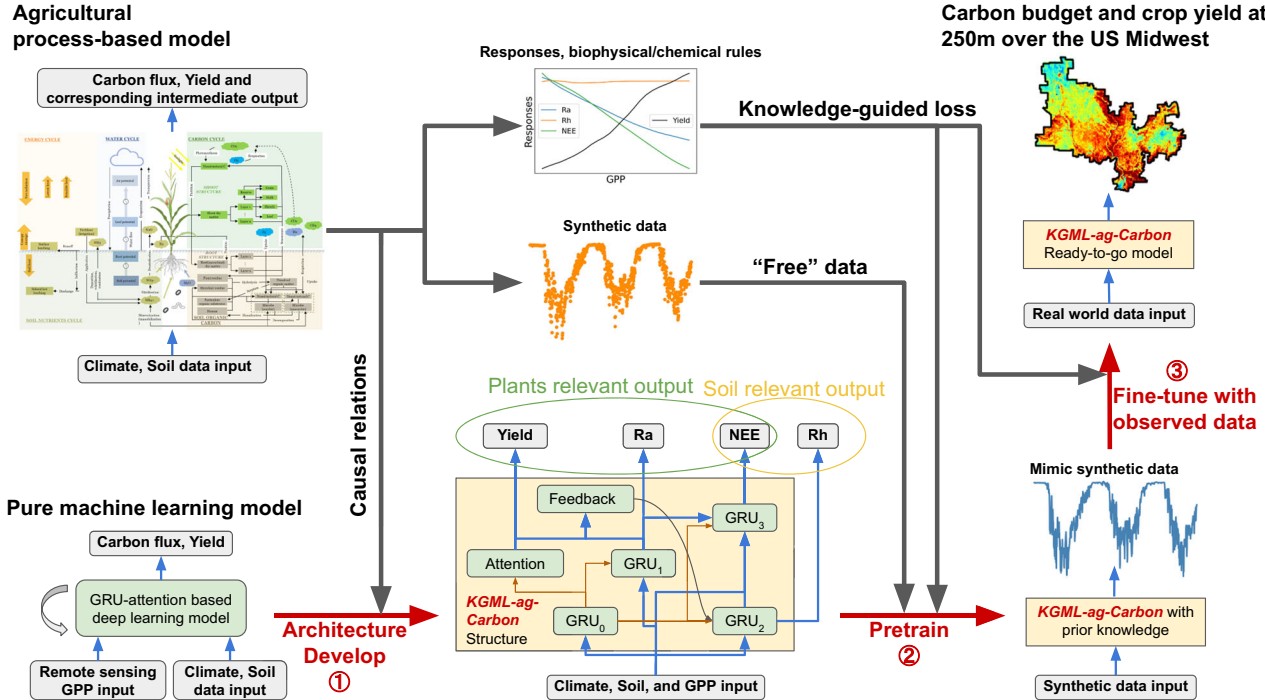

**Fig. 1 | Overview of the method and framework used for KGML-ag-Carbon development.** The development of KGML-ag-Carbon has three main steps: (1) Developing the architecture of the machine learning model based on the causal relations derived from an agricultural process-based model; (2) pre-training the KGML-ag-Carbon using synthetic data generated by a process-based model; and (3) fine-tuning KGML-ag-Carbon using observed low-resolution crop yield data and carbon fluxes from sparsely distributed eddy-covariance sites. The knowledge-guided losses were designed based on the process-based model to further constrain the response of target variables to input variables during both model pre-training and fine-tuning processes.

## Results

### Overview of the KGML-ag-Carbon framework

KGML-ag-Carbon is a novel framework combining process-based understanding and advanced AI approaches for simulating complex biogeochemical cycles under intensive management practices for agroecosystems. KGML-ag-Carbon distinguishes itself from previous KGML applications in other disciplines by its use of a well-validated PB model, *ecosys*[29], and its ability to directly assimilate remote sensing data. The deep learning model based on the gated recurrent unit (GRU) mechanism[30] was used to develop KGML-ag-Carbon's architechture (Fig. 1). The ecosystem theory of carbon allocation is the basis for the *ecosys* model, which was used to design the hierarchical structure of KGML-ag-Carbon, including submodules related to plants, soil, atmospheric carbon exchange, and feedback of carbon from plants to soil (Fig. S2). Importantly, outputs from the *ecosys* model provided synthetic data on ecosystem carbon allocation, associated fluxes, and environmental responses that were used to pre-train the KGML-ag-Carbon model. This pre-training step confers a major advantage to the model by improving the predictive ability with a minimal amount of labeled samples and accelerating the convergence in model tuning using labeled samples. GPP data, which represents the dominant carbon input to agroecosystems, was retrieved from remote sensing observations and assimilated into KGML-ag-Carbon as spatial constraints.

KGML-ag-Carbon resolves the major carbon budget components, including autotrophic respiration (Ra), heterotrophic respiration (Rh), total ecosystem respiration (Reco, Ra + Rh), and net ecosystem carbon exchange (NEE) on a daily scale, and yield on an annual scale. As in natural ecosystems, changes in agroecosystem soil carbon storage are determined by the mass balance of input and output carbon fluxes[31,32]. Ecosystem carbon inputs originate from plant photosynthesis, i.e., gross primary production (GPP), while soil carbon inputs include both aboveground and belowground litter and root exudates. Carbon outputs occur through respiration, including Ra from plant shoots and roots and Rh from SOC decomposition by microbes and fungi. Disturbances such as harvesting also remove carbon from the ecosystem periodically. Based on the carbon fluxes and yield estimated from KGML-ag-Carbon, annual changes in SOC can be determined using the mass balance equation $\Delta SOC = -NEE - Yield$[12,13,33].

We systematically explored multiple paths for improving the prediction performance of KGML-ag-Carbon, including pre-training the model with synthetic data and incorporating knowledge-guided (KG) loss functions, which addresses broader issues about reducing uncertainty for hybrid modeling. Over 14 million synthetic data and various KG loss functions were used to pre-train KGML-ag-Carbon to learn the prior knowledge from the PB model (Fig. 1). Using synthetic data generated by a PB model is several orders of magnitude cheaper than the cost of collecting real-world observations. The KG loss functions include biogeochemical/physical constraints such as mass balance (Ra+Rh-GPP = NEE), prediction thresholds (e.g., 0<yield<GPP), and responses of outputs to inputs (e.g., Rh should monotonically increase with SOC content under other fixed conditions).

A subset of the observed yield data from USDA (320 out of 630 counties) along with Reco (Ra + Rh) and NEE data from 11 cropland EC flux tower sites were then used to fine-tune the pre-trained KGML-ag-Carbon model to improve its prediction ability for real-world carbon budgets (Fig. 1). KG loss functions with biogeochemical/physical constraints similar to those used in pre-training were included in the fine-tuning. However, to preserve sufficient pre-training knowledge, the synthetic data were merged with observational data during fine-tuning, while extra constraints were added to the KG loss functions to maintain the responses of outputs to inputs (e.g., changes in Ra responses to the environment remain within 10%). Details on the structural development, datasets utilized, and training strategies for the KGML-ag-Carbon model are provided in the Methods section.

### Model performance in crop yield and carbon flux predictions

We evaluated the performance of KGML-ag-Carbon both before and after fine-tuning along with the sensitivity of model performance to the real-world training sample size (Figs. 2, S4, S6, S8). As an initial check on pre-training effectiveness, the pre-trained KGML-ag-Carbon model results for the test set of synthetic data (two years out of the 18-year period) were compared with *ecosys* simulations and found to be highly consistent, with $R^2$ values of 0.99, 0.99, 0.97, and 0.97 for yield, Ra, Rh, and NEE, respectively (Fig. S4). $R^2$ values of the *ecosys* model for corn and soybean yield predictions were 0.49 and 0.42, respectively, as benchmarked with observed county-scale crop yields, while values for daily Reco and NEE predictions were 0.67–0.89 and 0.59–0.88, respectively, compared with measurements from EC sites (green stars/boxes in Fig. 2, derived from Zhou et al. [13]). After fine-tuning, the $R^2$ of KGML-ag-Carbon for corn and soybean yield predictions on a test set of 210 counties were 0.91 and 0.88, while values for daily Reco and NEE predictions tested on 2 years of out-of-sample data from 11 EC flux tower sites were 0.94 and 0.96, respectively (Fig. S8).

The robustness test (Fig. 2) reveals that compared with a GRU-based pure ML model using the same inputs, KGML-ag-Carbon both consistently outperforms the pure ML model and has much lower sensitivity to the number of real-world training samples. In summary, the major differences between KGML-ag-Carbon and pure ML lie in the additional pre-training process and the customized model structure and loss functions guided by known scientific knowledge. These advancements enhance the optimization process of the ML model and allow reliable predictions to be made with fewer labeled samples. The pure ML model performance approached that of KGML-ag-Carbon at large sample sizes but performed poorly with small training sets, particularly for crop yield (Fig. 2a, b). The reduced need for training samples is a central advantage of KGML-ag-Carbon for real-world crop yield and carbon flux estimates because available training data are usually limited and collecting data from physical sampling is costly. The improvements in carbon flux predictions with increasing sample size mostly arise from capturing the interannual carbon dynamics, which can be more easily learned from seasonal patterns of GPP and climate (Fig. 2c, d). Even without fine-tuning (a training sample size of 0), the KGML-ag-Carbon by assimilating the GPP data as input, can outperform both *ecosys* model and pure ML model trained with small training samples (Fig. 2).

We conducted several additional experiments to evaluate the performance of KGML-ag-Carbon under different training, validation, and testing dataset splits, including out-of-sample performance of yield predictions in the spatial and temporal domains (Fig. S9a-f). For example, we used data from Illinois for testing and data from other states for training and validation and used several continuous years of data for training and validation with other years for testing. We also examined the effect of using extreme years with exceptionally high or low yields for testing and other years for training and validation. We note that the KGML-ag-Carbon model outperforms pure ML and process-based models in predicting yield in extreme years (Fig. S9e, f) primarily because it is constrained by both observations and synthetic data generated from the PB model. Similarly, out-of-sample performance for Reco and NEE predictions was investigated on both annual and daily temporal scales and specifically examined for sites (EC flux towers US-NE 1–3) with longer-term observations covering 2001–2019 (Fig. S9g–l). The results demonstrate consistently better performance of KGML-ag-Carbon compared to pure ML in all tested situations. The details of robustness tests on the KGML-ag-Carbon performance can be found in the Methods section.

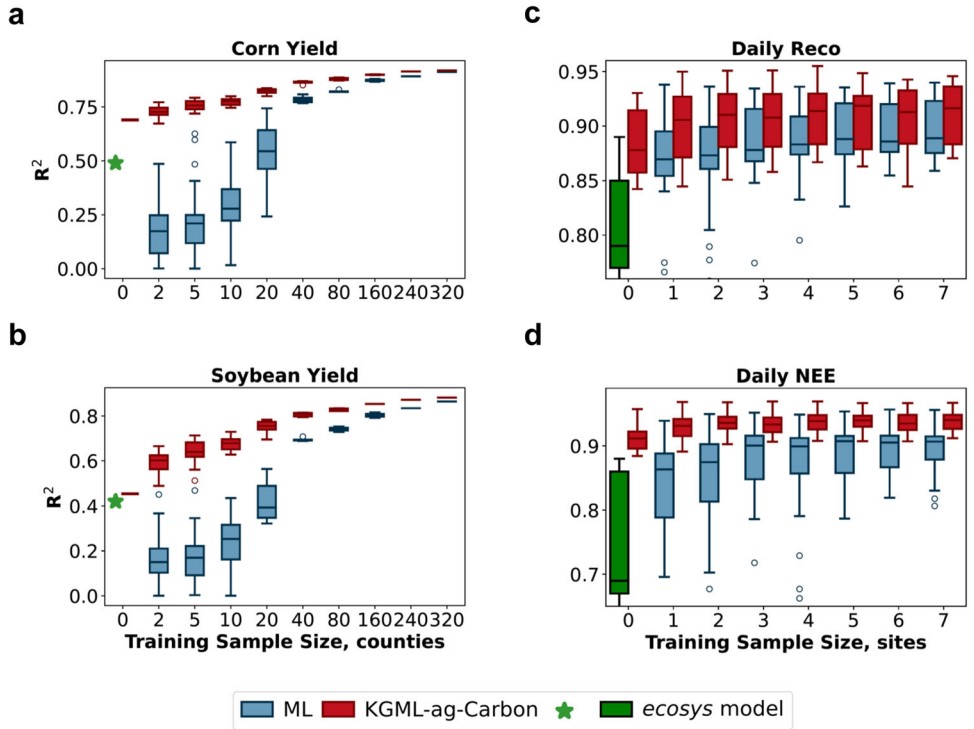

**Fig. 2 | The comparative performance of the pure ML model (blue boxes) and KGML-ag-Carbon (red boxes) when using different sizes of observed data samples for model training. a, b** The yield prediction performance over 210 counties. $n = 1, 50, 20, 10, 5, 4, 2, 1$, and 1 independent experiments for model ensembles with a training sample size of 0, 2, 5, 10, 20, 40, 80, 160, 240, and 320, respectively. Each training sample has a 21-year period of annual yield observations in one county. **c, d** The Reco and NEE prediction performance across 11 EC flux towers. n = 48 independent experiments for training sample size from 1 to 7. $n = 7$ and 6 independent experiments for the *ecosys* model and KGML-ag-Carbon model ensembles with a training sample size of 0, respectively. Each training sample has daily observations during the observation period in one site (varying by site, ranging from 5 to 19 years). Each box plot illustrates the first and third quartiles (lower and upper box edges), median (central line), and minimum and maximum (lower and upper whiskers), with outliers as round circles. The green stars represent the performance of *ecosys* in crop yield simulations across the U.S. states of Illinois, Iowa, and Indiana constrained with remotely sensed GPP and observed yield, and the green boxes represent the performance of *ecosys* in carbon flux simulations at 7 EC flux tower sites across the U.S. Midwest from Zhou et al.[13], which is a subset of the dataset used in this study. Only out-of-sample test results from cross-validation ensembles are depicted here. Details of the experiments can be found in the "Methods" section. Source data are provided as a Source Data file.

## Pathways to reduce KGML-ag-Carbon uncertainty

To understand the contribution of different strategies to improvements in the performance of KGML-ag-Carbon, we conducted full-factorial tests to include or exclude different model components and selected five representative models to use in interpreting the results (Fig. 3). The results reveal that using GPP data as an input and pre-training KGML-ag-Carbon with synthetic data contribute most to improving the performance of KGML-ag-Carbon relative to other strategies. When using larger real-world observations for model fine-tuning, the GPP data has the largest contribution to improving KGML-ag-Carbon performance; while pre-training with synthetic data is more important when using smaller real-world observation sets for model fine-tuning (Fig. 3a, b). This indicates that under data-limited situations, pre-training based on datasets generated by process models with sufficiently well-represented mechanisms can provide prior knowledge to significantly help improve the performance of ML. In contrast, when good-quality observational datasets are available, the ML model can learn complex relationships directly from the data so pre-training is less important.

The improvements to KGML-ag-Carbon provided by the hierarchical structure and KG loss functions are relatively small compared to those from GPP inputs and pre-training processes. One potential reason is that the model performance metrics ($R^2$ here) were already very high after adding GPP inputs and pre-training (Fig. 3a, b). However, including the hierarchical structure and KG loss functions significantly increases the ability of KGML-ag-Carbon to capture complex carbon flux dynamics and the interpretability of the predictions.

The pre-training process significantly reduces the residual mass balance of carbon fluxes (i.e., GPP-Reco-NEE), while the hierarchical structure and KG loss functions further reduce the mass balance residual to near-zero (Fig. 3c), indicating that the inclusion of hierarchical structure and KG loss functions constrain the model to follow physical rules. Although KGML-ag-Carbon has an overall performance similar to the ML + GPP and ML + GPP + pretrain models, significant improvements in NEE predictions are achieved in winter and summer, especially over periods with complex dynamics (Fig. 3d). The advantage of KGML-ag-Carbon over other models for various time periods is mainly attributed to the incorporated knowledge of distinct temporal patterns in Ra and Rh, such as Rh equals Reco in winter when plant growth is absent, which can be utilized to separately improve the performance of the Ra and Rh submodels in KGML-ag-Carbon. Other results from full-factorial tests and mass balance tests can be found in Fig. S11 and Fig. S12, respectively. The details outlining how the contributions of KGML-ag-Carbon components were identified can be found in the "Methods" section.

## High-resolution carbon flux estimates across the U.S. Midwest

Using the fine-tuned KGML-ag-Carbon, we predicted regional daily carbon fluxes and annual crop yields across the U.S. Midwest at a 250-m spatial resolution (smaller than a typical U.S. Midwest field, Fig. 4a−c). Inputs that drive KGML-ag-Carbon include daily climate data from NLDAS-2, topsoil properties from gSSURGO, remotely sensed daily GPP from the SLOPE product[34], and crop rotation information. The high spatial resolution is facilitated by high-resolution soil

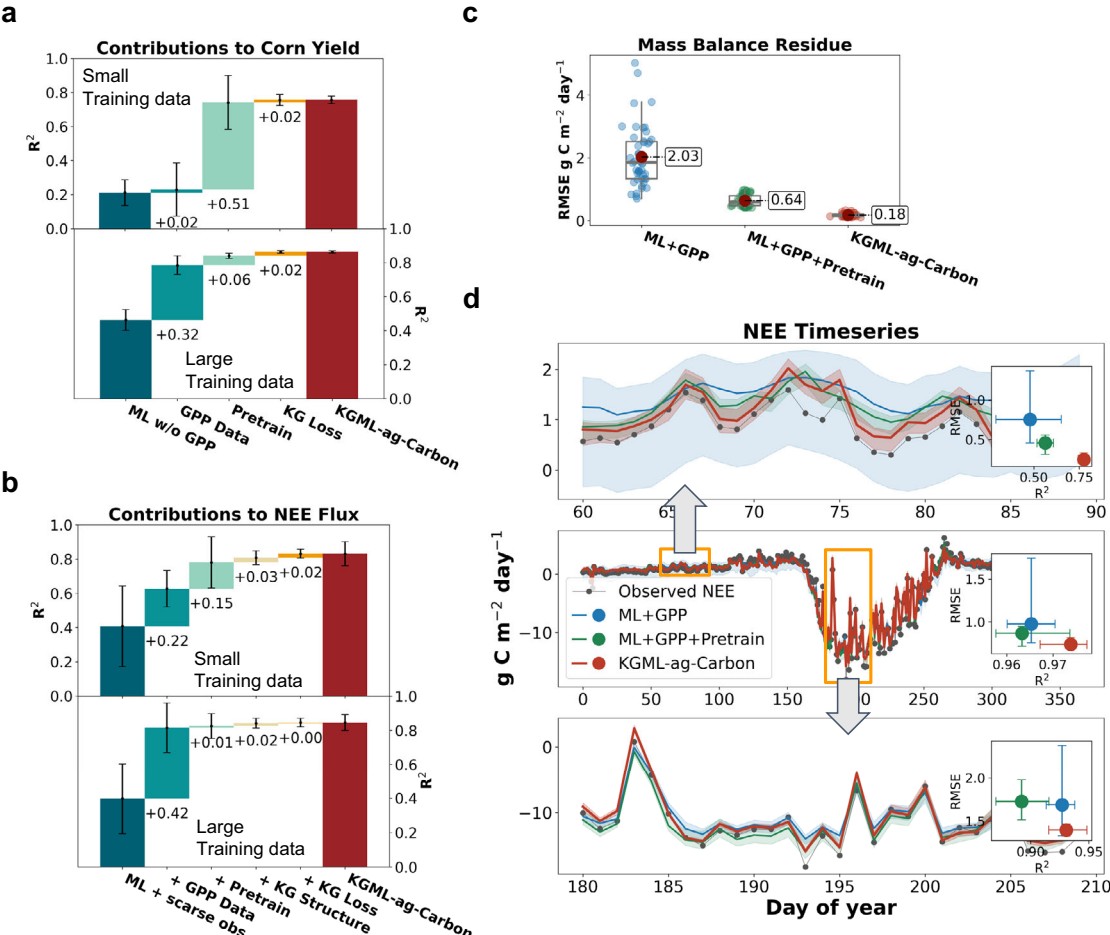

**Fig. 3 | The contributions of model structure and training strategies to improving KGML-ag-Carbon performance. a** The contributions from different components of KGML-ag-Carbon in improving the annual corn yield prediction accuracy by sequentially adding one component to the pure ML model. $n = 20$ and 5 independent experiments for model ensembles with small and large training data, respectively. **b** The contributions to improving the annual cumulative NEE flux prediction accuracy. $n = 48$ independent experiments for model ensembles with both small and large training data. Data in **a** and **b** are presented as mean values ± standard deviation. Values below each bar represent the mean performance increase from the previous step. ML w/o GPP indicates the pure ML model without GPP input; GPP Data indicates the ML model with GPP inputs; Pretrain indicates the GPP Data model pre-trained using the synthetic data generated by the process-based model; Structure indicates the model that contains hierarchical architecture, is pre-trained with synthetic data, and includes GPP inputs; KG Loss indicates the Structure model that considers knowledge-guided loss terms; KGML-

ag-Carbon indicates the final model that considers both knowledge-guided architecture and loss terms, contains GPP inputs and is pre-trained using synthetic data. **c** The benefits of knowledge-guided components (pretrain, structure, and KG Loss) on reducing the residual mass balance (GPP-Reco-NEE). $n = 48$ independent experiments. Each box plot illustrates the first and third quartiles (lower and upper box edges), median (central line), mean (solid red dot), and minimum and maximum (lower and upper whiskers). **d** An example (2016 of US-NE1) of predicted NEE fluxes from models with different knowledge-guided components. $n = 8$ independent experiments for model ensembles at this site. Shaded areas represent the region within the max and min of the simulation ensembles, while the solid lines represent the mean values. Data in inset plots are presented as mean values (solid dots) and minimums and maximums (whiskers) of $R^2$/RMSE. ML + GPP indicates the pure ML model with GPP inputs; ML + GPP + pretrain indicates the pure ML model pre-trained with synthetic data and with GPP inputs; KGML-ag-Carbon indicates the final model. Source data are provided as a Source Data file.

information, crop rotation maps, and the GPP product. The high temporal resolution comes from the climate and GPP product data, which provide daily information on environmental conditions and ecosystem carbon inputs. The procedures for generating high-resolution predictions across the U.S. Midwest are given in the "Methods" section.

The multi-year-averaged SLOPE GPP data and the carbon fluxes generated by KGML-ag-Carbon are closer to the EC flux tower observations (same dataset used in Fig. 2) than estimates from Trendy[35], a widely used carbon flux ensemble product generated by a suite of dynamic global vegetation models (Fig. 4). Although methods for evaluating KGML-ag-Carbon are somewhat limited at the regional scale, EC flux tower data and Trendy are suitable datasets for comparison in the absence of ideal benchmarks for this large region.

The distributions of GPP values are similar across SLOPE, EC flux tower observations, and the Trendy ensemble (Fig. 4d). The distributions of Reco and NEE estimated by KGML-ag-Carbon are similar to those of EC flux tower observations, but the ensemble of Trendy models over-estimated both Reco and NEE in the U.S. Midwest compared with flux tower observations (Fig. 4e, f). In addition, the estimated distributions of GPP, Reco, and NEE vary widely among individual Trendy models, which may arise from differences in structure and parameters among models. This reflects the large uncertainties remaining in current state-of-the-art PB models for carbon budgets, especially for regional-scale estimates. To summarize these comparisons, the carbon fluxes estimated by KGML-ag-Carbon demonstrate high spatial-temporal resolution and accuracy, providing a novel product for precise regional-scale carbon budget quantification down to a single field.

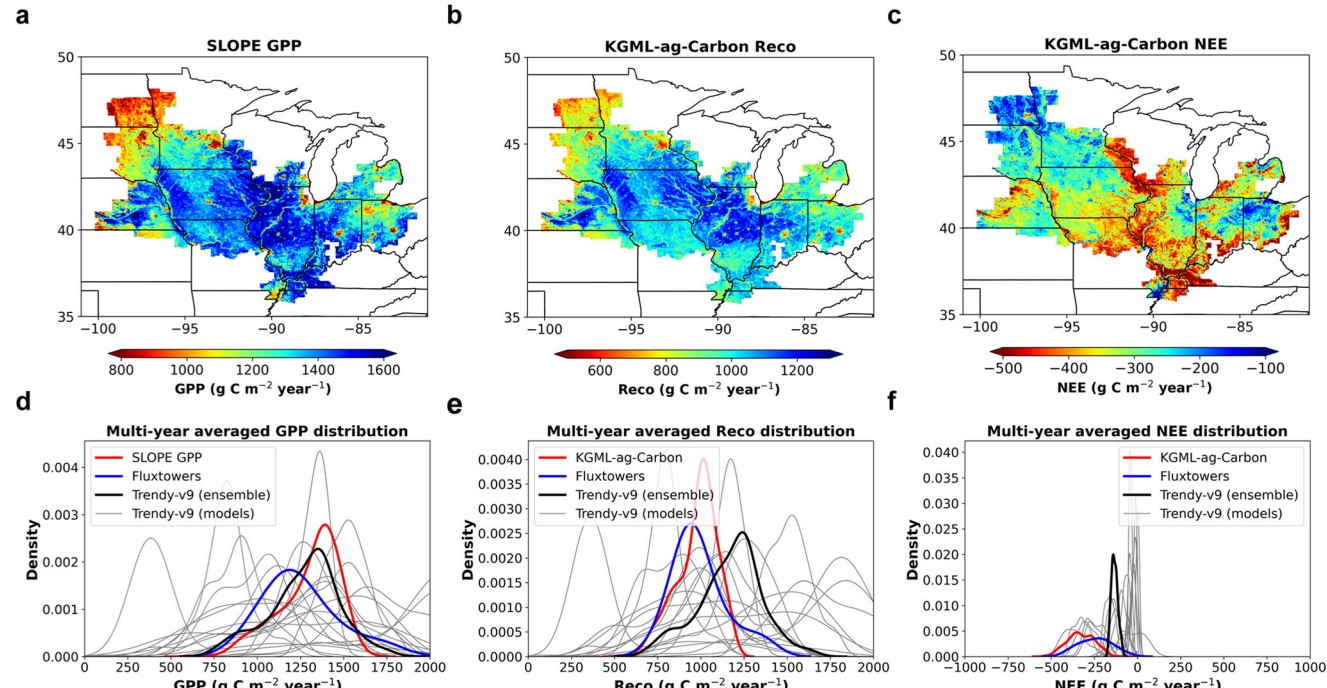

**Fig. 4 | The spatial pattern of averaged (2000–2019) annual accumulated carbon fluxes from KGML-ag-Carbon and their distributions compared with the Trendy-v9 product. a** The multi-year averaged remotely sensed GPP product based on MODIS near-infrared reflectance (SLOPE GPP), which is one of the KGML-ag-Carbon inputs. **b** The multi-year averaged annual accumulated Ra and Rh fluxes predicted by KGML-ag-Carbon. **c** The multi-year averaged annual accumulated NEE predicted by KGML-ag-Carbon. **d**–**f** The distributions of annual accumulated GPP, Reco, and NEE, respectively, from KGML-ag-Carbon predictions and Trendy-v9 during 2000–2019 and selected cropland eddy-covariance sites in the U.S. Midwest. The Trendy product used in this comparison is an ensemble product from multiple process-based models simulated carbon budget (a single gray line in **d**–**f** represents one model in Trendy, and black lines are the average outputs of all the models). The SLOPE GPP and KGML-ag-Carbon fluxes were averaged from 250 m to 0.05° in **a**–**c** for display. Source data are provided as a Source Data file.

## Discussion
### The benefits of high-resolution carbon budgets

The field-level quantification of carbon budgets, crop yields, and ΔSOC produced using KGML-ag-Carbon (as demonstrated for the U.S. Midwest) provides an accurate, cost-effective, and high-resolution product for potentially improving carbon sequestration assessments. To underscore the necessity of a high-resolution carbon budget and crop yield quantification, we generated 0.0025-degree and 0.5-degree resolution ΔSOC estimates from 2000 to 2020 using the mass balance approach with KGML-ag-Carbon. We created a high-resolution product using 250-m-resolution NEE and crop yield data predicted by KGML-ag-Carbon (Fig. 5a–c). These results were compared with ΔSOC estimates using a similar approach by implementing KGML-ag-Carbon at a 0.5-degree resolution (Fig. 5d–f). The high-resolution ΔSOC estimates reveal that the majority of changes fall within the range of -0.5% to 0.5% C/year (86%), which aligns well with the ranges observed in experimental studies[36–41] (Fig. 5c). Notable patterns include a decline in SOC across southern Minnesota, northern Iowa, and northeastern Illinois, as well as an increase in the southern U.S. Midwest. These patterns are primarily influenced by soil factors (explaining 43% variance) and climate factors (explaining 11% variance). Relatively colder, drier conditions, fewer carbon inputs into the soil, and higher SOC stock levels (larger Rh) contribute to greater carbon losses in northern regions (Fig. S14). A more detailed assessment of the ΔSOC patterns is given in the supplementary discussion. A comparison of the coarse and high-resolution ΔSOC estimations reveals notable differences (overall NRMSE = 86%) due to loss of detail (e.g., hot/cold spots) and relatively stronger mixed pixel effects in the 0.5-degree pixels (Fig. 5d, e). The histogram distribution (Fig. 5f) indicates a difference ranging from −0.1 (10% quantiles) to 0.9 (90% quantile) %/year between coarse- and high-resolution estimates. This difference cannot be neglected when compared with the high-resolution ΔSOC histogram distributions (Fig. 5c).

More detailed results regarding the differences between high-resolution and coarse-resolution GPP inputs, as well as Ra, Rh, NEE, and crop yield qualifications are provided in Fig. S15.

KGML-ag-Carbon employs a mass balance approach to estimate ΔSOC from NEE and yield, which are estimated by integrating all available data, including weather forcing, soil properties (which include static SOC), crop type, and remotely sensed GPP. These inputs and predicted NEE and yield are well-validated by observations. This approach allows us to make the best use of existing data to estimate the regional ΔSOC at low cost and high resolution, even in the absence of field-level measurements. We have undertaken validation efforts, focusing on sites within the U.S. Midwest with SOC measurements in multiple years post-2000 (Fig. S16, Table S1). These validations demonstrate that our model's ΔSOC estimates fall within observed ranges in most cases. However, performance is constrained by four key factors: (1) while all ΔSOC data was collected at the plot level (-10-m scale), the absence of localized forcing data required us to employ field-level inputs, namely 250-m GPP and weather data, to drive the model; (2) our estimated ΔSOC represents a combination of crop residue and humus, while the majority of measurements typically focus on humus content; (3) variations in management practices between each plot, such as tillage, fertilizer application, and crop rotation, further complicate field-level estimation, and (4) uncertainty in field-level SOC arises from lab measurement errors (up to 12%), spatial sampling errors (up to 50%), and resampling errors (up to 45%)[42,43], and can be exacerbated over extended time periods. Despite these challenges, our approach is valuable for mitigating carbon budget quantification errors, driven by its high resolution (250 m) and accuracy (Figs. 2–4), all while maintaining a low computational cost. It is also worth noting that while the NEE, Reco, and crop yield values in KGML-ag-Carbon are well-constrained, intermediate variables such as Ra, Rh, and crop residue still contain high uncertainties due to a lack of

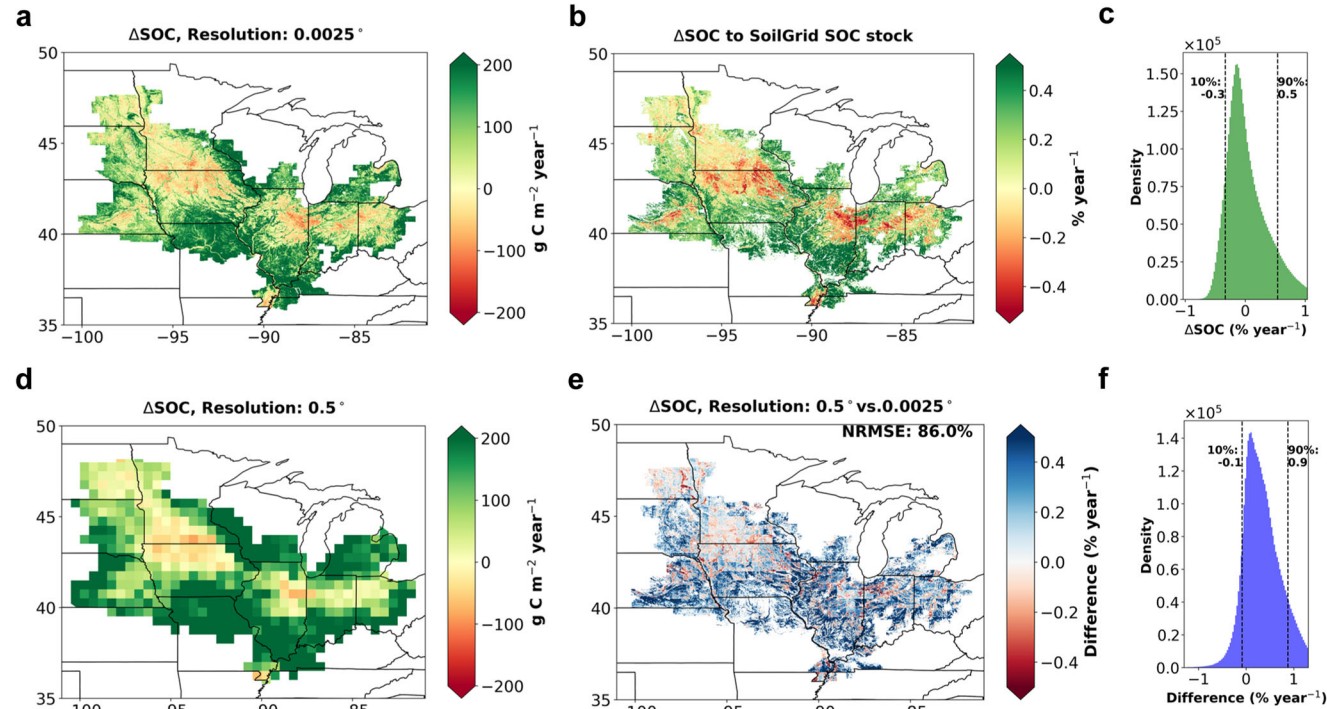

**Fig. 5 | The distribution of estimated ΔSOC during 2000–2020 and the demonstration of the impact of coarse resolution on ΔSOC. a** The ΔSOC estimation derived from the mass balance approach using KGML-ag-Carbon with 0.0025-degree-resolution carbon budgets. **b** The percentage fraction of the estimated ΔSOC in (**a**) compared to the SoilGrids SOC stock, limited to regions with over 50% corn or soybean planting; **c** The histogram distribution of percentage fractions in (**b**). **d** The ΔSOC estimation derived from the mass balance approach using KGML-ag-Carbon with 0.5-degree-resolution carbon budgets. **e** The spatial distribution of differences between coarse-resolution (0.5°) and fine-resolution (0.0025 degrees) ΔSOC estimations, relative to the SoilGrids SOC stock and limited to regions with over 50% corn or soybean planting; **f** shows the histogram distribution of the differences in **e**. Source data are provided as a Source Data file.

direct observational data constraints. These variables, however, are fundamental to understanding the underlying mechanisms. Therefore, this study also highlights the need for accurate field-level ΔSOC measurements to improve the reliability of ΔSOC quantification and the need for accurate measurements of Ra, Rh, and crop residues to constrain the underlying processes.

**Insights gained from the development of KGML-ag-Carbon**

Choosing a proper PB model as the scientific foundation for KGML development is critical. Although a large number of PB models exist for ecosystem carbon cycle modeling, models that incorporate sufficiently explicit representations of processes and are well-validated have more potential to benefit AI models, especially where no or few real-world samples are available to train the models. The PB model used in this study, *ecosys*, contains comprehensive first-principles descriptions of carbon transformation and translocation processes in plants and soil, and has been well-validated for different crop types and regions[13,44–47]. It provides valuable basic knowledge to guide the structural design and training of the KGML model. The benefits of *ecosys* in improving KGML-ag-Carbon's crop yield and carbon flux predictions were reflected in contribution tests as increased prediction accuracy (Fig. 3a, b), and reduced mass balance residuals (Fig. 3c). Future work may involve testing different PB ecosystem models (e.g., well-validated models in Asseng et al.[44] and Sitch et al.[35]) to explore the uncertainties arising from model selection for pre-training. However, this would require a significant collaborative effort.

KGML provides a promising way to use limited observations properly and efficiently by integrating them with other sources of data. In this study, we used three types of data from different sources and scales to train KGML-ag-Carbon. (1) The synthetic data generated by *ecosys* are much cheaper than real-world observations and can be used

for KGML model pre-training and designing KGML loss functions. Our results indicate that the prior knowledge learned from the synthetic data strongly contributes to improving the performance of KGML-ag-Carbon, especially in data-sparse situations (Figs. 2 and 4). (2) In-situ observations (e.g., EC flux towers, chambers) may include some important intermediate variables and can be temporally dense (long-term, frequent observations), but are often spatially sparse due to installation and labor expenses. They can be used to fine-tune the KGML model to capture temporal dynamics and intermediate processes, but it is necessary to control the responses to certain temporally static but spatially diverse factors (such as soil properties) learned from the PB model (Fig. S7). (3) Regional-scale observations at coarse resolution (e.g., county-level crop yield survey data) may have scale mismatches with the KGML input/output variables. Simply using those data to train the KGML by upscaling (or averaging) the model outputs to a coarse scale to calculate loss may force predictions of the fine-scale model to the average status of the coarse-scale observations. To overcome these shortcomings, the responses of target variables to diverse spatial and temporal factors must be guided by domain knowledge while using observations at coarse resolution to constrain the model (Fig. S5).

**Potential paths to improve agricultural GHG estimations by KGML**

Developing a KGML model with acceptable performance for GHG estimation is extremely challenging because emissions have large variations over space (hot spots) and time (hot moments), especially for intensively managed agroecosystems[12,13,33]. To further accurately quantify the high spatiotemporal variability of GHGs, KGML-ag-Carbon can be adapted to explore the use of internal network structures in recurrent neural networks (RNN), which take into account the

temporal correlations of states, and convolutional neural networks (CNN), which incorporate the spatial correlations of states. The multitask learning framework of KGML-ag-Carbon, along with the hierarchical structure, can be further enhanced by incorporating more representative processes and simulating key intermediate variables[28]. Since different GHGs are related to some common environmental states (e.g., soil moisture and soil temperature), one potential effective solution can be developing portable modules to predict shareable states, which can be used as inputs for different submodules. In the current KGML-ag-Carbon framework, some important management practices such as fertilization, irrigation, and tillage have not been explicitly considered in the model due to a lack of location-specific management information. It is currently assumed that the incorporation of remotely sensed GPP data in the KGML-ag-Carbon model can largely capture local variations in carbon fluxes due to management practices. Remote sensing data has shown potential for assessing local management practices such as cover cropping[48], tillage[49], and irrigation[50]. Recent advances in AI-based inverse modeling, such as Knowledge-Guided Self-Supervised Learning[51], may further improve estimates where management information is unknown. However, it should be noted that such methods are still in the early stages of development. Additionally, it is important to consider that management practices aimed at enhancing carbon storage in upland agroecosystems may inadvertently lead to an increase in other GHG emissions. For instance, while increasing the use of N fertilizers can improve carbon sequestration, it can also contribute to higher $N_2O$ emissions, partially offsetting the climate mitigation effect. Therefore, to conduct a comprehensive assessment of management impacts on GHG emissions (mostly $CO_2$ and $N_2O$) from upland agroecosystems, the N cycle needs to be incorporated into the framework due to the non-trivial impacts of $N_2O$ on the climate and the interactions between C and N[12]. However, incorporating C-N interactions is challenging because comprehensive measurements of both C- and N-related fluxes and soil states, which are needed to validate any new model, are lacking, and vital inputs such as fertilizer applications and crop windows needed for regional-scale extrapolation of the model are unavailable. Moreover, although KGML-ag-Carbon can accurately predict yield in extreme years (Fig. S9e, f), the impact of extreme weather conditions such as heat waves or flash droughts on agroecosystems remains unclear. Enriching KGML-ag-Carbon with simulations of intermediate environmental variables, such as canopy temperature and soil moisture, alongside the carbon budget quantification could potentially help dissect and elucidate the effects of extreme weather. If a reliable KGML tool was available to quantify the influences of different management practices and extreme weather on GHG emissions and productivity, it would be possible to develop reinforcement learning approaches[52,53] for optimizing management practices to maximize environmental and economic rewards.

### Transferability of the KGML-ag-Carbon to other applications

The KGML-ag-Carbon framework can be used for numerous other tasks, including predicting other target variables (e.g., N and P cycles), estimating C outcomes over larger regions (e.g., the entire U.S.), simulating carbon dynamics in different ecosystems (e.g., natural forests), and assessing impacts of management practices (e.g., cover cropping, tillage) and extreme weather (e.g., extreme heat or flash droughts). Three main aspects factor into the wide transferability of our framework. First, the *ecosys* model, which provides the scientific foundation for KGML-ag-Carbon, is a well-validated advanced agroecosystem model with detailed process representation for simulating complex interactions among carbon, nutrients, water, and energy cycles[13,29,54,55]. Various studies have demonstrated its global capability for simulating crop ecosystems[13], natural ecosystems[54], and management practice impacts[56–59]. Thus, *ecosys* can continue to generate abundant synthetic data for pre-training the model to adapt specific

pathways from input variables to target variables. Second, assimilating multi-source data can extend the framework to larger regions and more ecosystem types[15,25,27,60]. For example, the remotely sensed GPP data used in our study is available for the U.S. region, while other remotely sensed data (e.g., from MODIS, Landsat, WorldView, Legion, Sentinel-1, Sentinel-2, OCO-2, Planet Dove, SMAP satellites) may be available over larger areas and be used to estimate leaf area index, land surface temperature, evapotranspiration, soil moisture, tillage, fertility deficiencies, cover crop emergence, soil carbon sequestration, GHG emissions, and residue management practices. In addition, FLUXNET[61] has a total of 212 EC flux tower sites worldwide located in different ecosystems, providing carbon flux data and corresponding variables available for KGML model fine-tuning/validating. Third, KGML-ag-Carbon was tested to be over 1,000,000× faster than *ecosys*, completing the 21-year daily field-scale carbon budget quantification for the U.S. Midwest within 1.6 days using one GPU, while the *ecosys* model would require 5.9 years using 1000 CPUs. While process-based models can now be accelerated using GPUs, this typically requires significant code redesign and rewriting[62]. Unfortunately, *ecosys* is currently unable to run on GPUs. This high efficiency, together with the high fidelity of KGML-ag-Carbon to observational data, may facilitate the large-scale high-resolution multi-scenario assessment of management practices and spatially explicit parameter optimization, with some modifications to including the responses from carbon cycles to certain management practices or *ecosys* parameters.

## Methods

### Synthetic pre-training data for the KGML model

We used the agroecosystem model *ecosys* to generate synthetic data for crop yield, ecosystem autotrophic respiration (Ra), ecosystem heterotrophic respiration (Rh), net ecosystem exchange (NEE), and gross primary production (GPP). This synthetic data was used to pre-train the Knowledge-Guided Machine Learning for the Agricultural Carbon budget model (KGML-ag-Carbon). *Ecosys* simultaneously simulates carbon, water, and nutrient cycles within the soil and plant system based on biophysical and biochemical principles[29]. Its ability to simulate carbon fluxes and crop yields has been extensively validated across midwestern U.S. cropping systems[13]. We conducted county-level simulations using *ecosys* for 293 counties in the states of Illinois, Iowa, and Indiana using climate data from the North American Land Data Assimilation System (NLDAS-2 and soil data from the Gridded Soil Survey Geographic Database (gSSURGO). The synthetic database contains 10,335 simulations whose inputs include soil information, planting and harvest dates, crop parameters, and crop rotation information randomly selected from among predefined ranges to ensure a representative synthetic database. Within each county, the soil information was randomly selected from among the top 10 dominant cropland soil types in each county. The predefined range of planting dates is from April 15 to June 10, and the harvest date is from October 31 to November 20, which represents the general crop calendar in this region. In the database, one-third of the total simulations have corn-soybean rotations, one-third are soybean-corn rotations, and the remaining had corn and soybean planted randomly from 2001 to 2018 to represent common rotation strategies within this region.

### Datasets for fine-tuning, validation, and extrapolations

We fine-tuned and validated KGML-ag-Carbon for crop yield estimation over 637 counties and carbon fluxes estimation (i.e., Ra, Rh, NEE) at 11 cropland EC flux tower sites located within major U.S. corn and soybean production regions (Fig. S1). For fine-tuning and validation of the regional crop yield submodule, the 250-m daily GPP product derived from machine learning models based on Soil-Adjusted Near-Infrared Reflectance of vegetation (SANIRv)[34], 0-30 cm gSSURGO soil properties, NLDAS-2 climate data, and crop type information (CDL and CSDL[63] were used after and prior to 2008, respectively) were used as

**KGML-ag-Carbon inputs.** County-scale corn and soybean yields from NASS, and USDA (https://quickstats.nass.usda.gov/) were used as a benchmark. For fine-tuning and validation of the carbon flux submodules, the KGML-ag-Carbon inputs included the GPP data decomposed from observed NEE at EC flux tower sites using the ONEFlux tool[61], observed climate data from EC flux towers, gSSURGO soil information, and CDL crop type, while observed NEE and Reco from EC flux tower data were used as benchmarks. Because the daily GPP data we used is an average of GPP decomposed from NEE using different daytime and nighttime partition methods, it may not preserve the mass balance among NEE, Reco, and GPP; thus, we corrected the EC flux tower daily GPP by replacing it with observed Reco plus NEE in the following analysis. For estimating carbon fluxes at the regional scale, we used SANIRv-based GPP, NLDAS-2 climate data, and gSSURGO soil information as the model inputs.

## The structure of KGML-ag-Carbon

KGML-ag-Carbon uses a hierarchical structure[64] to incorporate the causal relations between different variables and processes with ecosystem knowledge for guidance, as presented in Fig. S2. It contains five submodules, including (1) a GRU_Ra module for daily Ra estimation, (2) a GRU_Rh module for daily Rh estimation, (3) a GRU_NEE module for daily NEE estimation, (4) an attention module for crop yield estimation, and (5) a GRU_Basis module to connect and support the other four modules. We used a type of recurrent neural network called a Gated Recurrent Unit (GRU) as the basic machine learning module to develop our model structure. GRU has been proven to perform similarly to Long short-term memory (LSTM[65]) in using cell states as internal memories to preserve historical information; however, GRU uses a simpler structure with fewer hidden states compared to LSTM and thus often remains more stable with a small number of training samples[28,30,66].

The recursive representations of GRU can be presented as:

$$\mathbf{h}_t = \mathbf{z}_t \odot \mathbf{n}_t + (1 - \mathbf{z}_t) \odot \mathbf{h}_{t-1} \tag{1}$$

$$\mathbf{z}_t = g(\mathbf{W}_{xz}\mathbf{x}_t + \mathbf{b}_{xz} + \mathbf{W}_{hz}\mathbf{h}_{t-1} + \mathbf{b}_{hz}) \tag{2}$$

$$\mathbf{n}_t = \tanh(\mathbf{W}_{xn}\mathbf{x}_t + \mathbf{b}_{xn} + \mathbf{r}_t \odot (\mathbf{W}_{hn}\mathbf{h}_{t-1} + \mathbf{b}_{hn})) \tag{3}$$

$$\mathbf{r}_t = g(\mathbf{W}_{xr}\mathbf{x}_t + \mathbf{b}_{xr} + \mathbf{W}_{hr}\mathbf{h}_{t-1} + \mathbf{b}_{hr}) \tag{4}$$

where $\mathbf{h}_t$ is the hidden state at time $t$, $\mathbf{x}_t$ is the input at time $t$, and $\mathbf{h}_{t-1}$ is the hidden state at time $t-1$ or the initial hidden state at time 0. $\mathbf{z}_t$, $\mathbf{n}_t$, and $\mathbf{r}_t$ are the update, reset, and new gates, respectively. $g$ is the sigmoid function and $\odot$ is the Hadamard product. $\mathbf{W}_{xz}$, $\mathbf{W}_{hz}$, $\mathbf{W}_{xn}$, $\mathbf{W}_{hn}$, $\mathbf{W}_{xr}$, and $\mathbf{W}_{hr}$ are learnable linear transformation matrices. $\mathbf{b}_{xz}$, $\mathbf{b}_{hz}$, $\mathbf{b}_{xn}$, $\mathbf{b}_{hn}$, $\mathbf{b}_{xr}$, and $\mathbf{b}_{hr}$ are corresponding learnable bias vectors.

Each GRU cell in KGML-ag-Carbon represents a GRU with 64 hidden units ($\mathbf{h}_t$ vector dimension = 64), and each dense cell is a linear transformation layer, which can be presented as:

$$\mathbf{y}_{flux,t} = \mathbf{W}_{hy}\mathbf{h}_t + \mathbf{b}_{hy} \tag{5}$$

where $\mathbf{y}_{flux,t}$ is the predicted flux target variables at time $t$, including Ra, Rh, and NEE. $\mathbf{W}_{hy}$ and $\mathbf{b}_{hy}$ are the learnable weight and bias, respectively. The GRU_basic, GRU_Ra, and GRU_NEE submodules have one layer of GRU cells while GRU_Rh has two layers of GRU cells. 20% of the output hidden states from GRU cells are randomly dropped by replacing them with zero values (the so-called 20% dropout) to avoid overfitting.

The attention module in KGML-ag-Carbon is a modified version of the traditional LSTM attention model[67], containing two layers: ATTN_Weight and ATTN_Densor. ATTN_weight can be represented as:

$$\boldsymbol{\alpha}_t = \frac{\exp(\mathbf{e}_t)}{\sum_{t=1}^{t=365} \exp(\mathbf{e}_t)} \tag{6}$$

$$\mathbf{e}_t = \tanh(\mathbf{W}_{he,4}\text{ReLU}(\mathbf{W}_{he,3}\text{ReLU}(\mathbf{W}_{he,2}\text{ReLU}(\mathbf{W}_{he,1}\mathbf{h}_t + \mathbf{b}_{he,1}) + \mathbf{b}_{he,2}) + \mathbf{b}_{he,3}) + \mathbf{b}_{he,4}) \tag{7}$$

where $\boldsymbol{\alpha}_t$ is the probability attention score calculated from a softmax function, representing the importance of time $t$ over the whole year. $\mathbf{e}_t$ is the weight score of $\mathbf{h}_t$ at time $t$ calculated from a 4-layer feedforward neural network (FNN) with a Rectified Linear Unit (ReLU) as the activation function for the first three layers and a hyperbolic tangent function (tanh) for the last layer. $\mathbf{W}_{he,i}$ and $\mathbf{b}_{he,i}$ are the learnable weight and bias for the $i^{\text{th}}$ layer in the FNN, respectively ($i = 1, 2, 3,$ and 4). $\boldsymbol{\alpha}_t$ and $\mathbf{h}_t$ are then multiplied in the ATTN_Densor layer to calculate the annual yield:

$$\mathbf{y}_{yield} = \mathbf{W}_{cy,4}\text{ReLU}(\mathbf{W}_{cy,3}\text{ReLU}(\mathbf{W}_{cy,2}\text{ReLU}(\mathbf{W}_{cy,1}\mathbf{c} + \mathbf{b}_{cy,1}) + \mathbf{b}_{cy,2}) + \mathbf{b}_{cy,3}) + \mathbf{b}_{cy,4} \tag{8}$$

$$\mathbf{c} = \sum_{t=1}^{365} \boldsymbol{\alpha}_t \mathbf{h}_t \tag{9}$$

where $\mathbf{y}_{yield}$ is the predicted yield for the input year, calculated from a 4-layer FNN with ReLU as the activation function for the first three layers. $\mathbf{c}$ is the self-weighted context vector, which has the same dimensions as the hidden state. $\mathbf{W}_{cy,i}$ and $\mathbf{b}_{cy,i}$ are the learnable weight and bias for the $i^{\text{th}}$ layer in the FNN, respectively ($i = 1, 2, 3,$ and 4). The attention module for yield collects simulated information of each day from the GRU_basis submodule as input and weighs the contribution of each day's information to the final yield prediction.

Crop annual residue $\mathbf{y}_{res}$ can be expressed as:

$$\mathbf{y}_{res} = \text{ReLU}\left(\sum_{t=1}^{Tx}(\mathbf{GPP}_t) - \sum_{t=1}^{Tx}(\mathbf{y}_{Ra,t}) - \mathbf{y}_{yield}\right) \tag{10}$$

where $\mathbf{GPP}_t$ and $\mathbf{y}_{Ra,t}$ are the GPP input and predicted Ra at time step $t$; $\mathbf{y}_{yield}$ is the annual predicted yield; and Tx is the number of days in the input time series (in this study Tx = 365). The ReLU function is used to prevent a situation in which the sum of predicted annual yield and Ra is bigger than the annual GPP.

The KGML-ag-Carbon inputs ($\mathbf{X}_t$) include seven daily climate variables: surface downward shortwave radiation (RADN, MJ m$^{-2}$ day$^{-1}$), maximum air temperature (TMAX_AIR, °C), the difference between the maximum and minimum air temperature (TDIF_AIR, °C), maximum humidity (HMAX_AIR, kPa), the difference between the maximum and minimum humidity (HDIF_AIR, kPa), wind speed (WIND, km day$^{-1}$), and precipitation (PRECN, mm day$^{-1}$). Additional inputs are daily GPP (g C m$^{-2}$ day$^{-1}$), year, crop type (corn/soybean), and nine soil properties averaged from 0 to 30 cm soil depth: bulk density (TBKDS, Mg m$^{-3}$), sand content (TCSAND, g kg$^{-1}$), silt content (TCSILT, g kg$^{-1}$), water content at field capacity (TFC, m$^3$ m$^{-3}$), water content at wilting point (TWP, m$^3$ m$^{-3}$), saturated hydraulic conductivity (TKSat, mm h$^{-1}$), soil organic carbon (TSOC, g C kg$^{-1}$), pH (TPH), and cation exchange capacity (TCEC, cmol$^+$ kg$^{-1}$). To increase the efficiency of the training process, we used the Z-normalization method to normalize each variable separately on synthetic data. The Z-normalization method can be expressed as:

$$\mathbf{Z} = \frac{(\mathbf{x} - \boldsymbol{\mu})}{\boldsymbol{\sigma}} \tag{11}$$

where $\mathbf{Z}$ is the normalized variable; $\mathbf{x}$ is the vector of a particular variable over all the data samples in the data set; $\boldsymbol{\mu}$ is the mean value of

$\mathbf{x}$; and $\sigma$ is the standard deviation (STD) of $\mathbf{x}$. The scaling factors derived from the *ecosys* synthetic data for each variable were used to normalize observed data into the same ranges as synthetic data. TDI-F_AIR and HDIF_AIR were used instead of absolute minimums of temperature (TMIN_AIR) and humidity (HMIN_AIR) because TMIN_AIR and HMIN_AIR follow similar trends as TMAX_AIR and HMAX_AIR, causing Z-normalization to be poorly defined numerically. Using the difference between maximum and minimum values provides clearer information about daily air temperature and humidity variations.

$\mathbf{X}_t$ are the inputs to the submodules of GRU_Basis, GRU_Ra, GRU_Rh, and GRU_NEE. Additionally, the output hidden states from GRU_Basis are inputs to GRU_Ra, GRU_Rh, and the attention module. The predicted annual yield, daily GPP, and daily Ra are then used to calculate the carbon in crop annual residue after harvest in Residue_layer. The annual residue is fed back to the soil for Rh calculation by inputting it on the 300$^{\text{th}}$ day of the year to GRU_Rh to assess the relationship between soil and plant carbon pools. Finally, the GRU_NEE takes predicted daily Ra and Rh together with $\mathbf{X}_t$ as input to predict daily NEE to assess the contribution of different carbon fluxes to NEE.

**Training strategies for KGML-ag-carbon**

We used a five-step training method to train KGML-ag-Carbon with *ecosys*-generated synthetic data and observed data, including (1) pre-training yield and Ra submodules using synthetic data, (2) pre-training Ra, Rh, and NEE submodules using synthetic data, (3) fine-tuning the yield submodule using observed data, (4) retraining Ra, Rh, and NEE submodules using synthetic data, and (5) fine-tuning Ra, Rh, and NEE submodules using observed data (Table 1). We utilized an enhanced mini-batch learning strategy[68] to effectively capture and maintain long-term temporal dependencies in the model. The best-performing submodules in the validation set at each step are saved for training in the next step.

Specifically, the KGML-ag-Carbon model was pre-trained using synthetic data to gain prior knowledge in steps 1-2, with two years of data randomly selected from the 18-year period of synthetic data for model validation, while the remaining 16 years of data were used for model training. In step 1, we trained the yield and Ra submodules together since they are crop-related variables and are used together for crop residue calculations (Eq. (10)), with the GRU_Rh and GRU_NEE submodules "frozen" by setting the learning gradient to zero. We used a mean-square-error (MSE)-based self-paced learning (SPL) method[69,70] to build our training losses to train the model from "easier" samples to "harder" samples (Note S1).

In step 2, we further pretrained the submodules for Ra, Rh, and NEE prediction together with the knowledge-based losses and responses by freezing the attention module and GRU_Basis module, considering the relationship of carbon fluxes. Besides the MSE loss, the loss function for step 2 also involves (1) the knowledge of mass balance (GPP - Ra -Rh = -NEE, considering the positive NEE direction to be from soil to atmosphere) to control the relationship between the input GPP and predicted Ra, Rh, and NEE, (2) the partial dependence plot (PDP, Fig. S3) to control the response of Rh to TSOC (Note S2). After two steps of pre-training, the KGML-ag-Carbon can successfully imitate *ecosys* for simulating yield, Ra, Rh, and NEE (Fig. S4).

In step 3, we fine-tuned the yield submodule with county-level crop yield data. The GRU_Ra, GRU_Rh, and GRU_NEE submodules were fully frozen and the GRU_Basis submodule was partially frozen by setting the learning rate to 20% of the original one. We included the knowledge-guided constraints in the loss function to control the range of yield (bigger than 0 and less than 0.5 times annual GPP) and maintain three key responses (i.e., yield responses to TSOC, GPP, and year) learned from the PB model (Note S3, Fig. S5). Detailed information on using coarse resolution (county-level) yield data to fine-tune our high-resolution model (250 m) is described in supplementary Note S3.

**Table 1 | Training strategies adopted for the KGML-ag-Carbon model**

| Training steps | Purposes and datasets | Submodules | Loss functions | Configurations |
|---|---|---|---|---|
| Step 1 | Pretrain yield and Ra with synthetic data | GRU_Basis; Attention module; GRU_Ra | Self-paced MSE (details in supplementary Note S1) | Adam optimizer; Learning rate = 0.001; Decay by 0.5 times per 100 epochs; Batch size = 500 samples; Random shuffle; 100-epoch early stop |
| Step 2 | Pretrain Ra, Rh, and NEE with synthetic data | GRU_Ra; GRU_Rh; GRU_NEE | MSE + Mass balance control + Response control (details in supplementary Note S2) | Adam optimizer; Learning rate = 0.001; Decay by 0.5 times per 20 epochs; Maximum 80 epochs; Batch size = 500 samples; Random shuffle; 10-epoch early stop lasting |
| Step 3 | Fine-tune yield with USDA NASS yield and synthetic data | GRU_Basis; Attention module | MSE + threshold control + response control (details in supplementary Note S3) | Adam optimizer; Learning rate = 0.001 for Attention module and 0.0002 for GRU_Basis; Decay by 0.5 times per 10 epochs; Maximum 40 epochs; Batch size = 21 counties; Random shuffle; 10-epoch early stop lasting |
| Step 4 | Maintain pretrained Ra, Rh, and NEE after yield finetuned with Synthetic data | GRU_Ra; GRU_Rh; GRU_NEE | MSE + Mass balance control + Response control (similar as Step 2) | Adam optimizer; Learning rate = 0.001; Decay by 0.5 times per 10 epochs; Maximum 40 epochs; Batch size = 500 samples; Random shuffle; 5-epoch early stop lasting |
| Step 5 | Finetune Ra, Rh, and NEE with EC flux tower data and synthetic data | GRU_Ra; GRU_Rh; GRU_NEE | MSE + Mass balance control + Response control (details in supplementary Note S4) | Adam optimizer; Learning rate = 0.0005, 0.0002, and 0.0005 for GRU_Ra, GRU_Rh, and GRU_NEE, respectively; Decay by 0.6 times per 30 epochs; Maximum 120 epochs; Batch size = 1 site; Random shuffle; 5-epoch early stop lasting |

Step 4 is similar to step 2 in terms of using synthetic data to train the Ra, Rh, and NEE submodules to avoid too much prior knowledge loss after fine-tuning the yield submodule. An experiment comparing Ra, Rh, and NEE prediction performance after step 4 and models with and without step 2 demonstrated the effectiveness of step 2 (Fig. S6). We attempted to remove step 2 and trained the model only in step 4 for carbon fluxes with 80 maximum epochs. The results showed a performance drop for Ra, Rh, and NEE pre-training, especially at the annual scale (Fig. S6; with step 2: annual RMSE = 13.9, 24.4, and 28.9 g C m$^{-2}$ day$^{-1}$ for Ra, Rh, and NEE, respectively; without step 2: annual RMSE = 17.0, 29.3, and 34.4 g C m$^{-2}$ day$^{-1}$).

Finally, we fine-tuned KGML-ag-Carbon using the daily observed Reco (Ra + Rh), NEE, and GPP data from 11 EC flux tower sites throughout the U.S. Midwest, with the GRU_Basis and Attention modules frozen (Fig. S2). The learning rates of the GRU_Ra, GRU_Rh, and GRU_NEE submodules were set to 50%, 20%, and 50% of the original one at the fine-tuning stage, respectively, to avoid overfitting and losing too much prior knowledge. The loss function for step 5 involves a similar mass balance constraint as step 2 but contains a different response constraint to preserve Ra and Rh responses to environmental variables learned from the processes-based model in data-sparse regions (Note S4, Fig. S7). Additionally, we introduced a method to separate the Ra and Rh during winter by assuming that most Reco during winter is from Rh since the selected EC flux tower sites were fallow during winter (Note S4). At each site, two years of data were randomly selected from the whole observed period as validation data, and the remaining data were used as training data. The final fine-tuned out-of-sample testing results are presented in Fig. S8.

## Robustness test for the performance of KGML-ag-Carbon

To investigate the robustness of KGML-ag-Carbon for yield, Reco, and NEE predictions, we conducted several experiments with different training sample sizes to compare the performance of KGML-ag-Carbon with a pure ML model under different conditions (Fig. 2, Fig. S9).

For yield predictions, we first randomly sampled 210 counties out of all 637 counties in the U.S. Midwest from NASS data for testing and kept 100 counties from the remainder for validation (Fig. 2a, b). Specifically, to conduct the yield robustness test, different sample sizes of 2, 5, 10, 20, 40, 80, 160, 240, and 320 counties were randomly selected from the remaining data, with ensemble times of 50, 20, 10, 10, 5, 4, 2, 1, and 1, respectively. The KGML-ag-Carbon model was trained following the 5-step training strategy described above, with a varying training sample size for fine-tuning in step 3. A sample size of zero for fine-tuning was also considered by skipping step 3 for the KGML-ag-Carbon training. The pure ML model for yield prediction is a 2-layer GRU model with attention, which is similar to GRU_Basis combined with the attention module presented in Fig. S2, with the same input features as KGML-ag-Carbon (including GPP). The pure ML models were trained with a similar method as KGML-ag-Carbon in step 3 with doubled maximum training epochs but without a knowledge-guided loss (only MSE loss) and pre-training (all other steps). In addition, we conducted further experiments with different training/testing split methods such as (1) using counties except for Illinois for training and Illinois for testing to detect spatial transferability, (2) training on the prior few years of data and testing on latter years to detect temporal transferability, and (3) training on the normal years and testing on extreme years (Fig. S9a–f). We used counties from Illinois (100) for testing and randomly sampled 100 counties from the remaining states for validation to test the KGML-ag-Carbon model in an independent out-of-sample testing data set (Fig. S9a, b). The training sample selection method was the same as the random sampling method. For detecting temporal transferability, we trained the model with all counties but split the 21-year data into training/validation/testing periods (Fig. S9c, d). We selected the front 2, 4, 6, 8, 10, 12, 14, 16, 18, and 20 years for training and validation, with the last 1, 1, 1, 2, 2, 2, 3, 3,

3, and 3 years of the selected periods as the validation sets, respectively. The remaining years of the 21-year period were used for testing the model. The prediction performance of each testing year was calculated separately and presented in Fig. S9c, d. To test the performance in extreme years, we trained the KGML-ag-Carbon model and pure ML model with data from all counties but excluded the selected extreme years of 2002, 2003, and 2012 (Fig. S9e, f). The extreme years were selected by detecting the outliers (outside the range of mean ± two times the STD) for each year based on a yield distribution calculated from the detrended yield for all counties and all years (Fig. S10). 2002, 2003, and 2012 have the top three numbers of outliers, with 98, 89, and 349 counties.

For Reco and NEE predictions, we divided the 11 EC flux tower sites into 6 testing groups based on the spatial distribution to detect spatial transferability of the KGML-ag-Carbon at different temporal scales (Fig. 2c, d, Fig. S9g–l). We conducted the ensemble experiments, and each time, we selected one group on which to test KGML-ag-Carbon and the pure ML model, which were trained and validated by randomly selected sites from the remaining groups. Specifically, we first divided the 11 eddy-covariance sites into 6 testing groups based on the spatial distribution, with US-Bo1 and 2 as group 1, US-Br1 and 3 as group 2, US-IB1 as group 3, US-KL1 as group 4, US-NE 1, 2 and 3 as group 5, and US-Ro1 and 5 as group 6 (Fig. S1). We selected one site as the validation data for each group and selected different sample sizes of 1, 2, 3, 4, 5, 6, and 7 sites as the training data from the remaining sites. The validation data traversed each of the remaining sites and training data of the same size would be forced to be different from each other. For example, group 1 has 2 sites for testing. If we would like to choose a training/validation sample for sample size 5, we would first select one site from the remaining 9 sites (excluding 2 test sites) and randomly sample 5 sites from the remaining 8 sites (excluding 2 test sites and 1 validation site). The 5 sampled sites would be compared with the previously selected 5-site training data and if they are the same, the sampling would be applied again. This process was conducted 9 times to cover all of the remaining sites so that the ensemble count for each sample size in group 1 was 9. Similarly, the ensemble times for each sample size in groups 2, 3, 4, 5, and 6 are 9, 10, 10, 8, and 9, respectively. The KGML-ag-Carbon model was trained following the 5-step training strategy described above with the training sample size varying for fine-tuning in step 5. A sample size of zero for fine-tuning was also examined by skipping step 5 in the KGML-ag-Carbon training. The pure ML model is a multitask 2-layer 64-unit GRU for Ra, Rh, and NEE simulation with the same input as the KGML-ag-Carbon model. The pure ML models were trained by a similar method as KGML-ag-Carbon in step 5 with doubled maximum training epochs but without a knowledge-guided loss (only MSE loss) and pre-training (all other steps). Finally, we investigated the overall performance of Reco and NEE prediction by combining results from all ensemble experiments at daily and annual scales (Fig. 2c, d, Fig. S9g, h), and investigated the performance at one representative location (the area containing US-NE1, 2, and 3 with 19-year data at each site) at daily and annual scales (Fig. S9i–l).

## Detecting the contributions of KGML-ag-Carbon components

To investigate the contributions of different KGML-ag-Carbon components to the final ready-to-go KGML-ag-Carbon performance, we conducted full-factorial tests for each component in the model and tested the model performance on an out-of-sample dataset (Fig. S11). Specifically, we included or excluded four components: (1) using GPP data as an input (GPP for short), (2) pre-training the model with synthetic data, (3) incorporating the KGML-ag-Carbon structure, and (4) implementing KG loss functions and the 5-step training strategy (if structure is applicable). In total, 16 individual models were trained. The training and testing data are similar to the robustness experiment described above. Specifically, to determine the contributions to the yield (flux) predictions, we used training sets of 5 and 40 counties

(1 and 7 sites) to train the models, referred to as small and large training sample sets, respectively. The optimized models were tested on out-of-sample data sets, which are NASS yields from 210 randomly selected counties and Reco and NEE from 6 groups of EC flux tower sites (the models tested on one group were trained and validated with data from sites chosen from other groups). We calculated the mean and STD of the prediction accuracy for all the models from ensemble experiments and detected the performance changes by comparing the models with and without each KGML-ag-Carbon component (Fig. S11). To illustrate the factors that contribute to the KGML-ag-Carbon model performance, we selected five representative models from the 16 trained models to showcase the direction of performance improvement. These models include (1) ML, (2) ML + GPP, (3) ML + GPP + pre-training, (4) ML + GPP + pre-training + KG structure, and (5) ML + GPP + pre-training + KG structure + KG loss (Fig. 3a, b). To further detect the influences of knowledge-guided components (i.e., pre-training, hierarchical structure, and KG loss functions) on improving the prediction performances, we compared three kinds of models, including an ML model with GPP data, an ML model with GPP and pre-training, and KGML-ag-Carbon, regarding the mass balance residues of predictions and the performance in capturing complex daily fluxes for a representative site-year (US-NE1-year 2016; Fig. 3c, d; Fig. S12).

## High-resolution predictions across the U.S. Midwest

After fine-tuning KGML-ag-Carbon with county-scale corn and soybean yield as well as EC flux tower observations from agroecosystem sites (Table 1, Step 5), the model was used to simulate regional annual crop yields and daily carbon fluxes (i.e., NEE, Ra, Rh, Reco) with a spatial resolution of 250 m over the main corn- and soybean-producing region of the U.S. Midwest (Fig. S1) from 2000 to 2020. To evaluate the performance of regional-scale carbon flux estimates, we compared the model results with Trendy[35], which was generated by a suite of dynamic global vegetation models at a monthly scale with spatial resolutions of 0.5° × 0.5° or coarser. The carbon flux values from this study were regridded to 0.5° by averaging the value of pixels within a 0.5° grid for comparison. The distribution of annual accumulated GPP, NEE, and Reco from these two datasets and the observations from the selected EC flux tower sites were used for the comparison (Fig. 4). The wide range of variation observed in the Trendy models ensemble can be attributed to the inclusion of diverse processes and alternative parameterizations adopted by models from different research communities, as described by Sitch et al. [35].

## Investigating the benefits of high-resolution quantification

To generate 0.0025-degree-resolution ΔSOC estimates for the U.S. Midwest (Fig. 5a–c), we employed the mass balance equation ΔSOC = -NEE - crop yield[12,13,33] over the period 2000–2020. Specifically, we regridded the 250-m-resolution NEE and crop yield estimations from KGML-ag-Carbon into 0.0025° estimations for use in the mass balance equation. To minimize the influence of undecomposed surface crop residues, which do not contribute to ΔSOC but are counted as part of our ΔSOC estimations through the mass balance approach, we selected the 21-year averaged value of ΔSOC. We then focused on regions where more than 50% of the area was planted with corn or soybean crops (Fig. S13a). The ΔSOC values were converted to percentage fractions (Fig. 5b) using ML-based SOC stocks derived from SoilGrids[71] (Fig. S13b). Specifically, we used corn and soybean fractions from CDL and CSDL data (Fig. S13a) to exclusively identify corn and soybean agroecosystems (total fraction > 0.5). This alignment with our model's current training scope helped reduce the mixed pixel effect resulting from inputing remotely sensed GPP data from other ecosystems. The SoilGrids SOC stock (Fig. S13b) was derived from organic carbon density (OCD) in each layer of the 200-cm soil depth at a 250-m resolution[71].

To attribute the spatial patterns of estimated ΔSOC, we conducted Pearson correlation analyses between the input variables (including seven climate variables and nine soil variables) and the target variables (including GPP, NEE, Ra, Rh, Yield, Reco, Residue, and ΔSOC) (Fig. S14). In our approach, each variable was temporally aggregated to a 21-year scale and Z-normalized using Eq. (11). The Residue variable was computed as GPP - Ra - Yield, representing the net carbon return from plants to the soil. While GPP served as an input to the KGML-ag-Carbon model, we included it as a target variable in the correlation assessment due to its pivotal role in the carbon cycle. In addition, we conducted a multiple linear regression to assess the total influence of climate factors and soil factors on ΔSOC. For more comprehensive explanations, please refer to the Supplementary discussion.

To demonstrate the advantages of high-resolution carbon budget quantification, we produced a 0.5-degree-resolution ΔSOC estimation (Fig. 5d) and conducted a comparative analysis with the 0.0025-degree-resolution estimation (Fig. 5e, f). Specifically, we employed KGML-ag-Carbon at a 0.5-degree-resolution and applied the mass balance approach to derive ΔSOC using 0.5-degree estimates of NEE and crop yield (Fig. S15; Fig. 5d). To achieve this, we employed a mean aggregation approach for each input variable, converting from 250-meter resolution to 0.5-degree resolution. However, the aggregation of crop types from high resolution to coarse resolution was not straightforward. To better emulate realistic crop rotations in the coarse-resolution simulation, we conducted two simulations involving corn–soybean rotations (corn in even years and soybean in odd years) and soybean-corn rotations (soybean in even years and corn in odd years). Subsequently, we used the corn/soybean fractions from CDL and CSDL data to compute weighted averages of corn and soybean estimations for each year using those two simulations. The differences between the 0.5-degree-resolution and 0.0025-degree-resolution ΔSOC estimations were then calculated (Fig. 5e, f) by subtracting the 0.0025-degree-resolution estimation from the 0.5-degree-resolution estimation. To enhance clarity, regions outside the corn/soybean agroecosystem were excluded, and the estimated differences were converted to percentage fractions using methods similar to those employed in generating Fig. 5b–c. We have adopted the normalized root mean square error (NRMSE) to describe the overall differences:

$$NRMSE = \frac{RMSE}{Q3 - Q1} \qquad (12)$$

Where RMSE is the root mean square error between 0.5-degree- and 0.0025-degree-resolution estimations, and Q1 and Q3 represent the three quantiles and one quantile of 0.0025-degree-resolution estimation, respectively. This method was chosen to avoid the denominator becoming too small. We also applied a similar approach to calculate the differences in other variables, including GPP, Ra, Rh, NEE, and Yield (Fig. S15). We note that the relative differences in Fig. S15 were calculated as (0.5-degree-resolution estimation - 0.0025-degree-resolution estimation)/0.0025-degree-resolution estimation, representing the relative differences relative to each 0.0025-degree pixel.

We conducted an extensive literature review to gather available soil organic carbon (SOC) measurements in the U.S. Midwest. This effort involved data from 18 sites, each with multiple SOC measurements at the plot level (-10 m) after 2000, facilitating ΔSOC validation for the KGML-ag-Carbon model (Fig. S16, Table S1). Observed data such as bulk density, initial SOC stock in the top 30 cm, and rotation management were integrated into the input feature when applicable. Other inputs were directly derived from our 250-m-resolution regional database, such as weather forcings and GPP based on the sites' geophysical locations. We have used an empirical equation[72] to simulate

the percentage fraction of SOC at different depths to total stock (assumed to be SOC in 0–100 cm), expressed as:

$$F_{SOC,Z} = -0.011\,Z^2 + 2.029*Z \qquad (13)$$

Where $F_{SOC,Z}$ is the estimated SOC percentage between 0 to $Z$ cm depth. This conversion factor aided in translating observed SOC values to the entire profile or to the top 30 cm in cases where depth-specific data was unavailable. It is worth noting that all of the collected ΔSOC data pertain to plot-level (-10 m) experimental measurements that primarily focus on detecting the influences of management practices. Data from those plots often lack the requisite localized forcing data needed by our model. Consequently, we resort to utilizing field-level (250 m) forcings such as remotely sensed GPP and reanalysis of NLDAS weather forcing, which poses a scale mismatch when compared to the plot-level observations. To illustrate this scale mismatch, we selected two sites from different studies[36,38] to compare the sizes of the experimental plots with the sizes of our predictions and neighboring real fields, as depicted in Fig. S16a, b.

### Development environment description
We used Pytorch 1.6.0 (https://pytorch.org/get-started/previous-versions/, last access: 21 Oct 2023) and Python 3.7.11 (https://www.python.org/downloads/release/python-3711/, last access: 21 Oct 2023) as the programming environment for model development. Statistical analysis, such as linear regression, was conducted using Statsmodels 0.14.0 (https://github.com/statsmodels/statsmodels/, last access: 21 Oct 2023) In order to use a GPU to speed-up the training process, we installed the CUDA Toolkit 10.1.243 (https://developer.nvidia.com/cuda-toolkit, last access: 21 Oct 2023). A desktop with an NVIDIA 2080 super GPU was used for code development and testing. The training processes, which required extensive time and memory space, were conducted on the Mangi and Agate clusters (https://www.msi.umn.edu/mangi, last access: 21 Oct 2023) from the High-Performance Computing facility of the Minnesota Supercomputing Institute (HPC-MSI, https://www.msi.umn.edu/content/hpc, last access: 21 Oct 2023) with two-way NVIDIA Tesla V100 GPUs.

### Reporting summary
Further information on research design is available in the Nature Portfolio Reporting Summary linked to this article.

## Data availability
All data used in this study are publicly available as detailed in the Methods. Briefly, the NLDAS-2 data used in study is available at https://ldas.gsfc.nasa.gov/nldas/nldas-2-forcing-data; gSSURGO is available at https://www.nrcs.usda.gov/resources/data-and-reports/description-of-gridded-soil-survey-geographic-gssurgo-database; the corn and soybean yield data is available at https://quickstats.nass.usda.gov/; the CDL data is available at https://croplandcros.scinet.usda.gov/; the CSDL data is available in Zenodo under accession code https://doi.org/10.5281/zenodo.4571628; the SLOPE GPP data is available at https://daac.ornl.gov/cgi-bin/dsviewer.pl?ds_id=1786; the benchmark TRENDY-v9 data is available at https://www.wdc-climate.de/ui/entry?acronym=DKRZ_LTA_891_ds00012; and the organic carbon density data used in this study is available in SoilGrids under accession code https://files.isric.org/soilgrids/latest/data/ocd/. The aggregated KGML-ag-Carbon predictions at 0.5° generated in this study are provided in the Source Data file, which has been deposited in the Zenodo database under accession code https://doi.org/10.5281/zenodo.10155516.

## Code availability
The *ecosys* process-based model is available at https://github.com/jinyun1tang/ECOSYS, and OneFLUX for EC flux tower data processing is available at https://github.com/fluxnet/ONEFlux. The source codes for data processing and an executable Python library of KGML-ag-Carbon models for running demo data are accessible through Zenodo under accession code https://doi.org/10.5281/zenodo.10155516.

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

## Acknowledgements

The authors acknowledge the support from DOE Advanced Research Projects Agency-Energy (ARPA-E) SMARTFARM programs (award No. DE-AR0001382), NASA Carbon Monitoring System Program (award No. 80NSSC18K0170), National Science Foundation Signal in the Soil program (award No. 2034385) and the Faculty CAREER Award program (award No. 1847334), USDA National Institute of Food and Agriculture (NIFA) Program (award No. 2017-67013-26253), and the Foundation for Food and Agriculture Research (award No. 602757). We also acknowledge the following AmeriFlux sites for their data records: US-Ne1, US-Ne2, US-Ne3, US-Bo1, US-Bo2, US-Br1, US-Br3, US-Ib1, US-Ro1, US-Ro5, and US-KL1.

## Author contributions

L.L., Z.J., W.Z., and K.G. conceived the study. L.L. and W.Z. jointly led the analysis, generated figures and tables, and wrote the initial paper. Z.J. and K.G. supervised the whole process. B.P., S.X., J. Tang, Q.Z., X.J., C.J., Z.Q., and V.K. contributed raw data, ecosys code and documentation, and/or machine learning methodology. All authors contributed to the interpretation of the results and edited the paper. The primary corresponding author is Z.J., whose lab maintains all shared data and code related to this paper.

## Competing interests

The authors declare no competing interests.
