## [Peer Review File · Nature Communications]

Reviewers' Comments:

Reviewer #1:

Remarks to the Author:

The manuscript "Knowledge-based artificial intelligence significantly improved agroecosystem carbon cycle quantification" uses multiple model structure and training strategies to achieve better carbon cycle quantification than purely-PB or purely-ML approaches. The authors demonstrate the relative contributions of various techniques to model performance and demonstrate the economic/policy value of high-resolution carbon flux information. In my opinion the novelty of this work is in the systematic evaluation of various forms of knowledge guidance and the connection of quantitative improvements in predictions to the carbon market.

As far as I can ascertain, the methodology and interpretation of results are sound. While the large number of training strategies can become somewhat convoluted, the authors justify each variation and make attempts to summarize clearly (e.g. Table 1).

The authors make several references to high-resolution carbon flux and crop yield predictions. It would be helpful to have the resolution contextualized earlier in the paper. I recognize this is covered to some extent in the discussion of basis risk.

I believe this application of KGML will be interesting in many application fields beyond agroecosystem studies and therefore read by non-experts. As such, I think the paper would be improved by providing more explanation of the major carbon cycle components and building more intuition about how they interact with each other.

Minor concerns:

I suggest removing language like "remarkable" (L24) and "outstanding" (L174)

Can you comment on why there is such significant spread in the ensemble of Trendy models?

Reviewer #2:

Remarks to the Author:

This work demonstrates the usefulness and applicability of combining process-based models with the data-adaptiveness of machine learning methods. The authors go an interesting route by simply pretraining on outputs from a process-based agricultural model, ecosys, to embed knowledge into a machine-learning framework. Additionally, the authors embed knowledge in constructing the machine learning model by considering the structure of their studied system by sensibly stacking stateful ML models. Furthermore, they designed so-called knowledge-guided loss functions to incorporate knowledge into the training process further.

The authors have developed, trained and tested this framework for modelling fine-scale carbon budgets and crop yields in the Midwest of the USA. They have done detailed tests and their line of argumentation is overall easy to follow. Overall, the authors make a sound case for their KGML-ag-Carbon framework.

I see four areas where I would like to see substantial improvements and clarifications:

1) Claim 'help save over 500 million US\$ per year for stakeholders'

It is unclear to me who these stakeholders are. Insurances? Would they overpay farmers because of coarse-scale carbon-budget accounting? Or are these stakeholders farmers who cannot sell their carbon sequestration services to the market because of a lack of reportability? My question is, who would lose 500 million US\$, and who would gain 500 million US\$ in your calculations? Would climate change mitigation be affected by this 500 million US\$ example because funds are not going into something cost-effective?

2) Realism of Δ SOC

The calculation of carbon credit basis risk relies on the simulation of NEE and yields since the authors define Δ SOC as $NEE - \text{yield}$. In my opinion, the readers should see histograms and maps of Δ SOC and how they compare with current SOC stocks, so we can assess the realism of these Δ SOC predictions. The authors should be able to compare their Δ SOC with repeated SOC measurements in this area, ideally at the flux towers. To me, this is paramount to avoid carbon double-accounting and ensure SOC is increasing. One point that the authors could discuss more thoroughly is the calculation of harvest residues that should remain on the field. Is it accurate to assume that harvest residues that remain on the field are $GPP - R_a - \text{yield}$? Is there uncertainty regarding where the residues go?

3) Static SOC

The KGML-ag-carbon has TSOC as a static input feature (chapter 5.3). This is problematic since that way the KGML-ag-carbon model is not dynamic, i.e. the essential variable the model predicts, Δ SOC, does not feedback on SOC. Instead, the initial SOC content is treated as a static, not changing variable. First, this is a missed opportunity which the authors could easily remedy in KGML-ag-carbon by introducing a recurrent cell which updates as $SOC[t+1] = SOC[t] + \Delta SOC[t]$. Second, by explicitly outputting SOC concentrations, it would be easy to compare the outputs of the model with the ground truth, i.e. if SOC contents are indeed increasing in these fields.

4) The trade-off with climate costs of N fertilization and N₂O

With the claim that stakeholders can save 500 million \$ in the climate mitigation market, the authors neglect the nitrogen cost for permanently storing carbon as soil organic carbon. Harvest residues are generally depleted in nitrogen compared to the harvest product of interest, especially soil microbes, which decompose harvest residues to SOC with a carbon use efficiency that depends on the stoichiometric imbalance between harvest residues and soil microbes. Nitrogen inputs are needed for a high carbon use efficiency. The energy costs of these nitrogen fertilizers would need to be considered for these carbon credits. Similarly, KGML-ag-carbon would reward management practices that enhance carbon storage but would not consider a potential trade-off of high fertilization with high N₂O emissions. The authors seem to have worked on N₂O separately in 'KGML-ag: A Modeling Framework of Knowledge-Guided Machine Learning to Simulate Agroecosystems: A Case Study of Estimating N₂O Emission using Data from Mesocosm Experiments' in GMD. In my opinion, it is very important to avoid incremental publishing, especially in high-impact journals such as Nature Communications and with claims like the 500 million \$ costs saved for stakeholders, which may come from false accounting when neglecting N₂O. The climate benefits of management practices have to be reported here, not financial implications for one group of stakeholders.

Miscellaneous issues and questions

- Why is an attention layer introduced for yield predictions? Can you explain why GRU modules were not used for this task?
- Is mass conservation encoded into the model? Is $GPP = \text{Yield} + R_a + R_h + NEE$ encoded or learned?
- In Figure 2, the authors show how the number of counties and flux towers influences the performance of the different models – it would be good also to have the number of observations associated with each sample, i.e. the number of annual or daily measurements per training set.
- For Figure 3, it would be better to have a full-factorial test of the contributions of GPP Data, Pretrain data from ecosys, and KG Loss. Maybe the authors did this already, and I misunderstood. For example, how well would a model perform if we combined ML with a knowledge-guided loss function? It seems the authors only added them in the last step.
- The authors compare KGML-ag-carbon with potential runtimes of ecosys. However, they compare GPU and CPU runtimes. Nowadays, process-based models can be run in parallel on GPUs – the comparison may be unfair.
- The GPP product SLOPE is not a pure remote-sensing product (line 241) but has machine learning models included – it would be good to note that down for the reader

- While the methods are described in great detail in the supplement it would be better to provide the code associated with this study to ensure reproducibility. There is also no mention of which framework was used to code KGML-ag-carbon and what kind of GPU was used for training.

Some typos

Line 110: cheapercompared – space is missing

Line 656: Fig. SX – number is missing

Reviewer #3:

Remarks to the Author:

In this manuscript, the authors design an AI framework, Knowledge Guided Machine Learning (KGML), to calculate carbon capture in agriculture sites using data from an ecosystem model, ecosys, to pretrain the KGML and observations of yield and eddy covariance to fine tune the model. They perform a variety of experiments to examine the performance sensitivity to the size of the training data and compared the results with a simple ML model. They also performed additional experiments to explore alternative training, out-of-sample and extreme year data, and different data-splitting to understand how to reduce uncertainty. The KGML method outperformed the ecosystem model and pure ML model for capturing yields and EC fluxes. Finally, the authors use this approach to estimate carbon capture in agriculture fields. I have a few suggestions and questions for the authors to address, but overall, I find this paper to be an interesting new application of machine learning that can have useful approaches for a variety of biogeochemical modeling.

I'm not an expert in AI so I can't comment much on that element. I did get a little lost in some of the details of the methods since I am not familiar with KGML, but I was able to understand overall what the authors did. I liked how the authors essentially did a sensitivity study of the approach, comparing with pure ML, and against the ecosystem model itself. One additional suggestion would be to look at how the method performs with pretraining from a different ecosystem model. I think it outside the scope of this manuscript, and the authors discuss the importance of ecosystem model choice in Section 4.2. I wonder how well the ecosys model works outside the U.S. or with other crop types, such as wheat or rice. This discussion might be expanded more, especially since the ecosys model doesn't perform great compared with observations.

I'm not convinced that the model would be accurate enough to estimate carbon capture in an agriculture field given the lack of validation data. Discussion section 4.1 seems out of place and unnecessary.

The field sites where data is collected (particularly the flux sites) don't have much information about them in the manuscript. All those plots have different crops and management practices (e.g., rotation, tillage, fertilizer application, irrigation, etc.). The reader doesn't find out until section 4.3 that these aren't taken into account in the model. While I like that the authors suggested a future application could be to quantify management, it should be emphasized that management practices are strong drivers of carbon inputs. There should be some data at these sites that can be used as model input or at least include a comment on how that will influence the results.

Another consideration for this model is that ML can only learn from the past and extreme events are often not included but they can be important for future estimates of carbon fluxes. For example, carbon fertilization is generally not included in ecosystem models, nor is crop response to extreme heat or flash droughts. Including these in ML models could improve our prediction of crop models.

The methods section indicates that most of the data is used for pre-training and training purposes. Do the authors worry about overfitting the model? Also, what is the difference between "easier" and "harder" samples (Line 623)

In the abstract, line 23, do the authors mean remote data instead of satellite?

Line 110: cheapercompared should be two words

Reviewer #4:

Remarks to the Author:

In this work, the authors proposed a Knowledge-Guided Machine Learning (KGML) framework and demonstrated that KGML can outperform existing process-based or ML models in quantifying carbon cycles. In general, none of the key parts in KGML framework (i.e., knowledge-guided loss, synthetic data pre-training, the use of indirect data) makes a novel contribution to existing machine learning techniques. Although the combined use of all those parts hasn't been explored by existing works yet, this manuscript presents an overall performance of KGML without clearly studying the contribution of each part (e.g., whether the sequence of including different parts would impact the contributions in Figs. 3ab, why knowledge-guided loss always has a very small contribution in Fig. 3). In addition, the descriptions of some key parts (e.g., knowledge-guided losses, and the use of indirect data) are not clear. In terms of quantifying carbon cycles, this manuscript does not provide a clear description of the case studies and the general motivation for studying them. The quality of the figures in the manuscript should be improved as well. Therefore, the reviewer cannot support the publication of this manuscript in Nature Communications.

Response Letter

Please be aware of the formatting of all responses:

1. Reviewer comment in **black**, response in **blue** and quotation from the main text in **red**;
2. The referred sections are based on the clean version of the revised manuscript, not the old or track change version.

Response to Reviewer #1

The manuscript "Knowledge-based artificial intelligence significantly improved agroecosystem carbon cycle quantification" uses multiple model structure and training strategies to achieve better carbon cycle quantification than purely-PB or purely-ML approaches. The authors demonstrate the relative contributions of various techniques to model performance and demonstrate the economic/policy value of high-resolution carbon flux information. In my opinion the novelty of this work is in the systematic evaluation of various forms of knowledge guidance and the connection of quantitative improvements in predictions to the carbon market.

As far as I can ascertain, the methodology and interpretation of results are sound. While the large number of training strategies can become somewhat convoluted, the authors justify each variation and make attempts to summarize clearly (e.g. Table 1).

Response: We greatly appreciate these positive comments on our manuscript, particularly the recognition that our systematic evaluation of various forms of knowledge guidance is novel, as well as the economic/policy value of high-resolution carbon flux information.

The authors make several references to high-resolution carbon flux and crop yield predictions. It would be helpful to have the resolution contextualized earlier in the paper. I recognize this is covered to some extent in the discussion of basis risk.

Response: Thank you for this suggestion, which we fully agree with. We have revised the second paragraph of the Introduction to emphasize the need for accurate, cost-effective, and high-resolution carbon quantification methods, particularly at the field level. These revisions should help better contextualize the resolution of our method in the paper. The revised paragraph is below:

“Increasing agricultural carbon sequestration is a key strategy for mitigating climate change, and significant efforts and investments have been made in the US and across the globe to implement programs that incentivize SOC enrichment^{7,8}. It is thus more important than ever to develop robust and scalable methods for reliably quantifying field-level carbon sequestration, both to assess the climate mitigation effect and to ensure that mitigation actions by individual farmers are compensated fairly and accurately. However, traditional carbon quantification methods, which rely on soil sampling, emission factors, and process-based (PB) modeling, encounter inherent challenges that hinder their ability to achieve the required levels of accuracy, scalability, and cost-effectiveness⁹⁻¹¹. ... Therefore, new methods are needed to overcome the limitations of PB and ML models, enabling cost-effective, accurate, and interpretable measurement and monetization of carbon outcomes at the individual field level, to reduce errors in aggregated quantifications and promote more sustainable land management practices^{12,19}.”

I believe this application of KGML will be interesting in many application fields beyond agroecosystem studies and therefore read by non-experts. As such, I think the paper would be improved by providing more explanation of the major carbon cycle components and building more intuition about how they interact with each other.

Response: Thank you for this suggestion. We agree that it's important to provide more context and elaborate on carbon cycle components and their interactions. Thus, we have added more description to section 2, paragraph 2, covering carbon inputs/outputs of agroecosystems, respiration types, and

disturbance impacts to enhance the accessibility to non-experts and clarify the significance of KGML-ag-Carbon. The revised content is below:

“KGML-ag-Carbon resolves the major carbon budget components, including autotrophic respiration (R_a), heterotrophic respiration (R_h), total ecosystem respiration (Reco, $R_a + R_h$) and net ecosystem carbon exchange (NEE) on a daily scale, and yield on an annual scale. As in natural ecosystems, changes in agroecosystem soil carbon storage are determined by the mass balance of input and output carbon fluxes^{30,31}. Carbon inputs originate from plant photosynthesis, i.e., gross primary production (GPP), while carbon inputs to the soil include both aboveground and belowground litter and root exudates. Carbon outputs occur through respiration, including R_a from plant shoots and roots, and R_h from SOC decomposition by microbes and fungi. Disturbances such as harvesting may also remove carbon from the ecosystem periodically. Based on the carbon fluxes and yield estimated from KGML-ag-Carbon, annual changes in SOC can be determined using the mass balance equation $\Delta SOC = -NEE - Yield$ ^{12,13,32}. Detailed information about the structural development of KGML-ag-Carbon is given in Methods 5.3.”

Minor concerns:

I suggest removing language like "remarkable" (L24) and "outstanding" (L174)

Response: Following this suggestion, we have removed such expressions from the revised manuscript.

Can you comment on why there is such significant spread in the ensemble of Trendy models?

Response: The significant spread in the Trendy models ensemble is mainly due to the inclusion of diverse processes and alternative parameterizations in the models from different research communities, as described by Sitch et al. (2015). More details about the parameterizations and processes of the Trendy models are described in Table S1 of the supplementary material for that paper: <https://bg.copernicus.org/articles/12/653/2015/bg-12-653-2015-supplement.pdf>. We have also added a brief explanation in section 5.7:

“... The wide range of variation observed in the Trendy models ensemble can be attributed to the inclusion of diverse processes and alternative parameterizations adopted by models from different research communities, as described by Sitch et al. (2015)¹⁵.”

Reference:

Sitch, S., Friedlingstein, P., Gruber, N., Jones, S. D., Murray-Tortarolo, G., Ahlström, A., Doney, S. C., Graven, H., Heinze, C., Huntingford, C., Levis, S., Levy, P. E., Lomas, M., Poulter, B., Viogy, N., Zaehle, S., Zeng, N., Arneth, A., Bonan, G., ... Myneni, R. (2015). Recent trends and drivers of regional sources and sinks of carbon dioxide. *Biogeosciences*, 12(3), 653–679.

Response to Reviewer #2

This work demonstrates the usefulness and applicability of combining process-based models with the data-adaptiveness of machine learning methods. The authors go an interesting route by simply pretraining on outputs from a process-based agricultural model, ecosys, to embed knowledge into a machine-learning framework. Additionally, the authors embed knowledge in constructing the machine learning model by considering the structure of their studied system by sensibly stacking stateful ML models. Furthermore, they designed so-called knowledge-guided loss functions to incorporate knowledge into the training process further.

The authors have developed, trained and tested this framework for modelling fine-scale carbon budgets and crop yields in the Midwest of the USA. They have done detailed tests and their line of argumentation is overall easy to follow. Overall, the authors make a sound case for their KGML-ag-Carbon framework.

I see four areas where I would like to see substantial improvements and clarifications:

1) Claim ‘help save over 500 million US\$ per year for stakeholders’

It is unclear to me who these stakeholders are. Insurances? Would they overpay farmers because of coarse-scale carbon-budget accounting? Or are these stakeholders farmers who cannot sell their carbon sequestration services to the market because of a lack of reportability? My question is, who would lose 500 million US\$, and who would gain 500 million US\$ in your calculations? Would climate change mitigation be affected by this 500 million US\$ example because funds are not going into something cost-effective?

Response: The stakeholders we refer to are participants in the carbon market, which include farmers who adopt new practices, carbon project developers (e.g. companies or governments) that pay farmers to make changes and collect generated carbon credits, and the entities that purchase carbon credits. Overpayments to farmers represent a loss of money that could have funded more carbon removal, whereas underpayments to farmers risk farmers not being able to recover the cost of implementing carbon sequestration practices and being discouraged from continuing them. We calculated the total loss to both carbon credit buyers and farmers, which represents misallocated funds.

Moreover, our main point in mentioning the potential loss is to emphasize the importance of accurate carbon quantification in making cost-effective decisions for climate mitigation. The overall goal of climate mitigation is to reduce greenhouse gas emissions and support carbon sinks, which requires a multi-faceted approach and careful allocation of limited resources. Inaccurate carbon quantification can lead to an inefficient carbon market, which may result in wasted funds that could be better invested in more cost-effective solutions. Therefore, we believe that improving the accuracy and reliability of carbon quantification is crucial for optimizing investments and achieving our collective climate goals. We have added new text to emphasize this point in the second paragraph of the introduction:

“Increasing agricultural carbon sequestration is a key strategy for mitigating climate change, and significant efforts and investments have been made in the US and across the globe to implement programs that incentivize SOC enrichment^{7,8}. It is thus more important than ever to develop robust and scalable methods for reliably quantifying field-level carbon sequestration, both to assess the climate mitigation effect and to ensure that mitigation actions by individual farmers are compensated fairly and accurately. However, traditional carbon quantification methods, which rely on soil sampling, emission factors, and

process-based (PB) modeling, encounter inherent challenges that hinder their ability to achieve the required levels of accuracy, scalability, and cost-effectiveness⁹⁻¹¹. ... Therefore, new methods are needed to overcome the limitations of PB and ML models, enabling cost-effective, accurate, and interpretable measurement and monetization of carbon outcomes at the individual field level, to reduce errors in aggregated quantifications and promote more sustainable land management practices^{12,19}.”

We have removed the statement regarding the 500 million US\$ savings to acknowledge the concern that this finding is based on scenario simulations with many uncertainties. Instead, we have revised the abstract and added discussion in section 4.1 to emphasize the uncertainties in both observation and simulation of soil organic carbon:

“The prediction improvement indicates that over 70% of the error in NEE and Δ SOC estimations can be reduced.”

2) Realism of Δ SOC

The calculation of carbon credit basis risk relies on the simulation of NEE and yields since the authors define Δ SOC as NEE – yield. In my opinion, the readers should see histograms and maps of Δ SOC and how they compare with current SOC stocks, so we can assess the realism of these Δ SOC predictions. The authors should be able to compare their Δ SOC with repeated SOC measurements in this area, ideally at the flux towers. To me, this is paramount to avoid carbon double-accounting and ensure SOC is increasing. One point that the authors could discuss more thoroughly is the calculation of harvest residues that should remain on the field. Is it accurate to assume that harvest residues that remain on the field are $GPP - R_a - yield$? Is there uncertainty regarding where the residues go?

Response: We fully agree that validation of Δ SOC is crucial to ensure the use of this model for carbon accounting. The validation suggested by the reviewer would be an ideal approach, but one that is not currently feasible due to limitations in existing SOC data. An exhaustive literature search revealed three main reasons for this:

- (i) SOC measurements have high uncertainty from various sources, including spatial heterogeneity, sampling design, variability in bulk density, and variation in soil processing methods and laboratory assays (Goidts et al., 2009; Stanley et al., 2023), which together could easily lead to uncertainty exceeding 50% in field-level SOC stock estimation, even over multi-year timescales. The uncertainty of annual Δ SOC is even larger because the magnitude of change is relatively small. This is likely the reason why the most accessible SOC data in the US are static measurements (Batjes et al., 2020).
- (ii) Scale mismatch between point sampling and field-level estimation. Our literature search also found that most Δ SOC measurements are from plot-level experiments (Al-Kaisi & Kwaw-Mensah, 2020; Ibrahim et al., 2018; Jin et al., 2015; Khan et al., 2007; Olson et al., 2014; Poffenbarger et al., 2017; Schmer et al., 2014; Varvel, 2006; Venterea et al., 2006), which are not representative of the field-level values needed for Δ SOC validation. There're some dense sampling efforts for field-level SOC benchmarking by private companies, but unfortunately, data are not available for public research. A discussion of soil sampling methods that can be used for benchmarking field-level SOC change is given by Guan et al. (2022).
- (iii) Other key variables needed to verify model performance are also missing. We have attempted to downscale the KGML-ag-Carbon model from field scale to plot scale to validate the Δ SOC using plot-level data. However, such experiments often do not report precise latitude and longitude

coordinates that are needed to extract remotely sensed GPP data to constrain the KGML-ag-Carbon model, or have too small plot size so that even the 30m resolution SLOPE-GPP data is too coarse to distinguish between plots.

For the KGML-ag-Carbon model, we therefore adopted a mass balance approach to indirectly estimate Δ SOC from NEE and yield, which have been well-validated against available individual components. The Δ SOC estimates integrate the best available data, including static SOC, weather forcing, soil properties, crop type and remotely sensed GPP. However, we acknowledge that this approach has limitations: Ra and Rh cannot be separately validated, as they are seldom partitioned in flux tower sites, and the influences of different management practices are not directly considered, which could further affect the accuracy of the calculated Δ SOC. This type of situation is a common challenge for the modeling community; thus, we emphasize the need for more comprehensive measurements of different components of the carbon cycle.

Similarly, we acknowledge that predicting residue fates using GPP-Ra-Yield also has limitations due to the lack of direct observations of residues and Ra, and the absence of management data regarding residue removal from the field. The crop residue estimated in the Residue_layer of KGML-ag-Carbon was only used as a prior information input into the GRU_Rh submodule. After pretraining using *ecosys*-generated synthetic data, KGML-ag-Carbon is able to determine the relationship between GPP-Ra-Yield and actual residue when there are disturbances like removal. The Rh was further constrained together with Ra by Reco, which is derived from EC flux-tower measurements. While partitioned Rh data from Reco would be a more ideal constraint, such partitioned data is often limited and available only at coarse temporal resolution (seasonally ~ annually) (Jian et al., 2021).

Because we don't have direct Δ SOC validation, as pointed out by the reviewer, we have removed the basis risk calculation using Δ SOC that involves policy-sensitive US\$ values. Instead, we have added discussion on the need for accurate SOC measurements and focus on the benefits of high-resolution quantification of NEE (which is well-validated in this study) and Δ SOC. We have also added a discussion of the uncertainty of the model's intermediate variables and the need for measurements to constrain them. We have updated Section 4.1, Fig. 5 (shown here as Fig. R3), Fig. S13 (with the removal of panel c that displays basis risk), and Methods 5.8 to reflect these changes. While we recognize the importance of accurate field-level SOC measurements for validating carbon accounting and the need for measurements of Ra, Rh, and crop residue to constrain the underlying processes, we believe that our study provides valuable insights into the potential of the extensible KGML-ag-Carbon framework for reducing carbon budget quantification errors, which can aid future carbon accounting and climate mitigation efforts.

The revised Fig. 5 (here as R3) is:

Fig. R3. The impact of spatial resolution on NEE and Δ SOC quantification. (a-b) NEE absolute errors caused by using coarse resolution for carbon budget estimations. (c-d) Δ SOC absolute errors caused by using coarse resolution. 0.0025-degree resolution is treated as “field-level” ground truth.

Revised discussion section 4.1 is listed below:

“The field-level quantification of carbon budgets, crop yields, and Δ SOC produced using KGML-ag-Carbon (as demonstrated for the US Midwest) provides an accurate, cost-effective, and high-resolution product for potentially improving carbon sequestration assessments. We used the term absolute error to quantify the advantages of high-resolution carbon budgets generated by KGML-ag-Carbon. Previous approaches often assess the carbon budget using factors that are averaged over a certain region, and absolute error describes the potential consequences resulting from mismatches between local-scale factors and the regional baseline. As used here, absolute error represents the difference of NEE or Δ SOC that can be attributed to individual fields using a high-resolution carbon budget product compared to larger-scale estimates based on lower-resolution products (Methods 5.8). Generally, the absolute error decreases quickly with increasing resolution of the carbon budget product for the US Midwest (Fig. 5 and Fig. S13). For example, when using a 1x1°-resolution product, the uncertainties in estimated NEE and Δ SOC of individual fields are up to 90 gC/m²/year and 101 gC/m²/year (75% quantile), respectively (Fig. 5). Based on this calculation, those absolute errors will lead to a total 5.6 and 6.2 GgC/year (1 Gg = 10⁹ g) errors for NEE and Δ SOC quantification, respectively, in the US Midwest. However, if using a product with a relatively high resolution of 0.005°, maximum

estimate uncertainties decrease to 27 gC/m²/year for both NEE and ΔSOC (75% quantile). The total errors would be reduced by 70% and 73% purely by changing the resolution of NEE and ΔSOC quantifications, respectively (Fig. 5). More detailed results about the benefits of high-resolution NEE, crop yield, and ΔSOC quantifications are given in Fig. S13. To be noted, the KGML-ag-Carbon employs a mass balance approach to estimate ΔSOC from NEE and yield, which are estimated by integrating all available data, including weather forcing, soil properties (which include static SOC), crop type, and remotely sensed GPP, and are well-validated by observations. This approach allows us to make the best use of existing data to estimate the regional ΔSOC at low cost and high resolution, even in the absence of field-level measurements. However, it is important to acknowledge that the lack of validation using field-level measurements of ΔSOC limits the certainty of our estimates, as such measurements are often scarce and associated with large uncertainties. For instance, at 250-m scales, uncertainty in SOC arises from lab measurement error (up to 12%), spatial sampling error (up to 50%), and resampling error (up to 45%)^{35,36}. The uncertainty of ΔSOC over a long period of time will be even larger. Including measurements with such uncertainties in the KGML-ag-Carbon training data may decrease the reliability of other components such as NEE and yield, which have relatively lower measurement errors. It's also worth noting that while the NEE, Reco, and crop yield in the KGML-ag-Carbon are well-constrained, the intermediate variables like Ra, Rh, and crop residue may still contain high uncertainty due to a lack of direct observational data. These variables, however, are crucial for a better understanding of the underlying mechanisms. Therefore, while the KGML-ag-Carbon offers a feasible method for reducing carbon budget quantification errors due to its high resolution (250 m), high accuracy (Fig. 2-4), and low computational cost, it also highlights the need for accurate field-level ΔSOC measurements to improve the reliability of ΔSOC quantification, and the need for accurate measurements of Ra, Rh, and crop residue to constrain the underlying processes.”

In addition, we have also revised Methods section 5.8:

“5.8 Investigating the benefits of high-resolution carbon budget quantification

We used the absolute error to quantify the uncertainties in NEE and crop yield quantification for individual fields in coarse-resolution carbon budget estimations (Fig. 5, Fig. S13). Specifically, we first regridded the 250-m-resolution products from KGML-ag-Carbon into a 0.0025-degree product to be used as a baseline for field-level carbon credit quantification. Then we aggregated the 0.0025-degree products to various coarse resolutions, including 0.005, 0.01, 0.02, 0.05, 0.1, 0.2, 0.5, and 1 degrees. The coarse-resolution products were compared with the 0.0025-degree product to calculate the NEE and yield differences as absolute errors. ”

3) Static SOC

The KGML-ag-carbon has TSOC as a static input feature (chapter 5.3). This is problematic since that way the KGML-ag-carbon model is not dynamic, i.e. the essential variable the model predicts, ΔSOC, does not feedback on SOC. Instead, the initial SOC content is treated as a static, not changing variable. First, this is a missed opportunity which the authors could easily remedy in KGML-ag-carbon by introducing a recurrent cell which updates as $SOC[t+1] = SOC[t] + \Delta SOC[t]$. Second, by explicitly outputting SOC concentrations, it would be easy to compare the outputs of the model with the ground truth, i.e. if SOC contents are indeed increasing in these fields.

Response: As suggested, we conducted extra simulation experiments using a recurrent cell to update $SOC[t+1]$ as $SOC[t] + \Delta SOC[t]$ in the KGML-ag-Carbon model and tested this new approach with both

synthetic and observed data. However, the results indicate that including a predicted variable with large uncertainty as an input in the model reduces the prediction performance of target variables in synthetic data tests, especially those that are most relevant, such as Rh (Fig. R4). Additionally, adding dynamic SOC either has no significant influence on performance in real observations or slightly reduces the performance of Reco and NEE (Fig. R5).

Fig. R4. Performance comparison between KGML-ag-Carbon models using synthetic data with static SOC input from the gSSURGO database and dynamic SOC input from updating static SOC with Δ SOC predicted from the last time step.

Fig. R5. Performance comparison between KGML-ag-Carbon models using observed data with static SOC input from the gSSURGO database and dynamic SOC input from updating static SOC with Δ SOC predicted from the last time step. Models were trained using either 5 or 40 counties (1 or 7 sites) for yield (flux) optimization and tested using out-of-sample yield data for 210 counties from NASS (the out-of-sample group of sites).

Introducing a recurrent cell that updates $SOC[t+1] = SOC[t] + \Delta SOC[t]$ is an example of the teacher forcing (TF) approach in ML. However, this method has limitations in modeling time series, as it primarily focuses on learning $\Delta SOC[t]$, hindering the model's ability to accurately accumulate SOC changes over time. Additionally, due to temporal dependencies, $\Delta SOC[t]$ is often small, causing TF-trained models to overly rely on $SOC[t]$ for predicting $SOC[t+1]$. In the testing phase, errors in predicting $SOC[t]$ cannot be corrected, leading to error accumulation and poor SOC predictions. More detailed explanation of the limitations of TF can be found in section 7.8 of Xu et al. (2023). Moreover, while using a recurrent cell to update $SOC[t+1]$ is an ideal solution when sufficient accurate SOC measurements are available as constraints, we cannot currently do so at the landscape level due to the lack of reliable measurements of SOC change. Therefore, we choose to retain SOC as a static input feature in this study.

4) The trade-off with climate costs of N fertilization and N₂O

With the claim that stakeholders can save 500 million \$ in the climate mitigation market, the authors neglect the nitrogen cost for permanently storing carbon as soil organic carbon. Harvest residues are generally depleted in nitrogen compared to the harvest product of interest, especially soil microbes, which decompose harvest residues to SOC with a carbon use efficiency that depends on the stoichiometric imbalance between harvest residues and soil microbes. Nitrogen inputs are needed for a high carbon use efficiency. The energy costs of these nitrogen fertilizers would need to be considered for these carbon credits. Similarly, KGML-ag-carbon would reward management practices that enhance carbon storage but would not consider a potential trade-off of high fertilization with high N₂O emissions. The authors seem to have worked on N₂O separately in 'KGML-ag: A Modeling Framework of Knowledge-Guided Machine Learning to Simulate Agroecosystems: A Case Study of Estimating N₂O Emission using Data from Mesocosm Experiments' in GMD. In my opinion, it is very important to avoid incremental publishing, especially in high-impact journals such as Nature Communications and with claims like the 500 million \$ costs saved for stakeholders, which may come from false accounting when neglecting N₂O. The climate benefits of management practices have to be reported here, not financial implications for one group of stakeholders.

Response: Thank you for the valuable discussion. We have revised the discussion in section 4.1 to focus on the NEE and ΔSOC regional results, and removed the less well-validated basis risk calculation related to ΔSOC . Additionally, we have added new content in section 4.3 to discuss the limitations of the current model and potential solutions for integrating C-N interactions in future versions of KGML-ag-Carbon.

We acknowledge that management practices that aim to enhance carbon storage may unintentionally increase N fertilizer use, especially when new practices reduce yields, thereby increasing N₂O emissions and partially offsetting climate mitigation effects. In practice, this is less of a concern because most agricultural carbon programs impose strict requirements on “additionality”, so they require farmers to report their fertilizer use and will discount the carbon credit if farmers increase fertilizer rate.

To attract more farmers who have concerns about potential yield reduction when adopting conservation practices without increasing fertilizer use, carbon programs indeed should have better quantification approaches that can evaluate the comprehensive GHG outcomes, especially the tradeoff between SOC sequestration and N₂O emissions. However, most of current quantification approaches entail costly direct measurements like soil sampling to determine SOC change (Wendt & Hauser, 2013) or eddy-covariance flux sensors for GHG emissions measurement (Baldocchi et al., 1988), or they use

simple emission factor estimates to approximate 'carbon outcomes' based on various management practices (IPCC, 2019).

In light of these considerations, the role of KGML-ag-Carbon will become pivotal in the existing carbon program due to its high accuracy, efficiency, and cost-effectiveness. The KGML-ag-carbon model does not incentivize any particular management practices. Instead, it provides a novel means of quantifying the carbon outcomes of varying practices. In the US, this might mean the implementation of cover crops and reduced tillage without an increase in fertilizer.

Currently, we do not consider C-N interactions in KGML-ag-Carbon for the following reasons:

(i) Although N₂O has a higher global warming potential (273±118 times) than CO₂ (Forster et al., 2021), it accounts for only 6% (CO₂eq) of all U.S. GHGs from human activities in 2021, which is much smaller than that of CO₂, which accounts for 79% (EPA, 2023).

(ii) The agroecosystem C cycle, N cycle and their interactions are extremely complex and have underlying processes that remain unclear, so our use-inspired research focuses solely on the carbon cycle.

(iii) Modeling C-N interactions using KGML requires detailed intermediate variables or co-incident measurements of CO₂, N₂O, and soil inorganic nitrogen, along with many other physical variables that are currently not available.

(iv) Our KGML-ag model for N₂O is an early attempt at hybrid modeling for predicting biogeochemistry at the plot-level and is not directly applicable to the regional level due to the lack of regional-scale data on certain key drivers, such as explicit fertilizer use and crop planting dates, as well as the lack of regional observations to constraint the nitrification and denitrification processes at scale.

In the future, creating a coupled C-N KGML framework will require an improved understanding of the mechanisms involved, sufficient data from super sites (i.e., containing CO₂, N₂O, and soil state data) to validate the processes, reliable information on regional management practices (e.g., fertilizer, tillage, irrigation), and ideally regional constraints on the N cycle to scale up the model as well.

The revision for section 4.3 is:

“In the current KGML-ag-Carbon framework, some important management practices such as fertilizing, irrigation, and tillage, have not been explicitly considered in the model due to a lack of location-specific management information. It is currently assumed that the incorporation of remotely sensed GPP data in the KGML-ag-Carbon model can largely capture local variations in carbon fluxes due to management practices. Remote sensing data has shown potential for assessing local management practices such as cover cropping⁴¹, tillage⁴², and irrigation⁴³. Recent advances in AI-based inverse modeling, such as Knowledge-Guided Self-Supervised Learning⁴⁴, may further improve estimates where management information is unknown. However, it should be noted that such methods are still in the early stages of development. Additionally, it is important to consider that management practices aimed at enhancing carbon storage in upland agroecosystems may inadvertently lead to an increase in other GHG emissions. For instance, while increasing the use of N fertilizers can improve carbon sequestration, it can also contribute to higher N₂O emissions, partially offsetting the climate mitigation effect. Therefore, to conduct a comprehensive assessment of management impacts on GHG emissions (mostly CO₂ and N₂O) from upland agroecosystems, the N cycle needs to be incorporated into the framework due to the non-trivial impacts of N₂O on the climate and the interactions between C and N¹². However, incorporating C-N interactions is complicated because the underlying processes are unclear and comprehensive measurements of both C- and N-related fluxes and soil states needed to validate any new model are lacking.”

Miscellaneous issues and questions

- Why is an attention layer introduced for yield predictions? Can you explain why GRU modules were not used for this task?

Response: The reason for introducing an attention model for yield prediction is that yield is an annual variable, while other components, such as Ra, Rh, and NEE, are reported on a daily scale. The attention mechanism was introduced to selectively weight daily GRU outputs and improve annual yield prediction by focusing on the most pertinent temporal period. The attention mechanism has been well documented in the literature for its effectiveness in sequence-to-sequence learning tasks.

To clarify the role of the attention mechanism in yield predictions, we have added a description in Methods 5.3, which provides more details on how the attention model collects information from GRU and weighs the contribution of data from each day to the final yield prediction. Details of the revision in Methods 5.3:

“The Attention module in KGML-ag-Carbon is a modified version of the traditional LSTM attention model¹¹, containing two layers: ATTN_Weight and ATTN_Densor. ATTN_weight can be represented as:

$$\alpha_t = \exp(e_t) / \sum_{t=1}^{t=365} \exp(e_t) \quad (\text{Eq. 6})$$

$$e_t = \tanh(W_{4,he} \text{ReLU}(W_{3,he} \text{ReLU}(W_{2,he} \text{ReLU}(W_{1,he} h_t + b_{1,he}) + b_{2,he}) + b_{3,he}) + b_{4,he}) \quad (\text{Eq. 7})$$

where α_t is the probability attention score calculated from a softmax function, representing the importance of time t over the whole year. e_t is the weight score of h_t at time t calculated from a 4-layer feedforward neural network (FNN) with a Rectified Linear Unit (*ReLU*) as the activation function for the first three layers and a hyperbolic tangent function (*tanh*) for the last layer. $W_{i,he}$ and $b_{i,he}$ are the learnable weight and bias for the i^{th} layer in the FNN, respectively ($i = 1, 2, 3,$ and 4). α_t and h_t are then multiplied in the ATTN_Densor layer to calculate the annual yield:

$$y_{\text{yield}} = W_{4,cy} \text{ReLU}(W_{3,cy} \text{ReLU}(W_{2,cy} \text{ReLU}(W_{1,cy} c + b_{1,cy}) + b_{2,cy}) + b_{3,cy}) + b_{4,cy} \quad (\text{Eq. 8})$$

$$c = \sum_{t=1}^{365} \alpha_t h_t \quad (\text{Eq. 9})$$

where y_{yield} is the predicted yield for the input year, calculated from a 4-layer FNN with ReLU as the activation function for the first three layers. c is the self-weighted context vector, which has the same dimensions as the hidden state. $W_{i,cy}$ and $b_{i,cy}$ are the learnable weight and bias for the i^{th} layer in the FNN, respectively ($i = 1, 2, 3,$ and 4). The attention module for yield collects simulated information of each day from the GRU_basis as input and weighs the contribution of each day's information to the final yield prediction.”

- Is mass conservation encoded into the model? Is $GPP = Yield + Ra + Rh + NEE$ encoded or learned?

Response: The short-term (~daily) mass balance, which is $GPP = Ra + Rh - NEE$ ($NEE = -NEP$ and $NEP = GPP - Ra - Rh$) (Stuart Chapin et al., 2011), is learned by the KGML-ag-Carbon model through the loss function with a tolerance. This way of enforcing mass conservation is a soft constraint, which we believe is better than hard-coding it into the model structure, since the data used to train the model also has an uncertainty that may not always follow the mass balance. We did not consider the long-term (~annual) mass balance due to the large uncertainty in disturbances and accumulated carbon leakage.

Details can be found in Text S2 in supplemental material: “...the second term constrains the model predictions with carbon mass balance ($GPP - Ra - Rh = -NEE$) and can be expressed as:

$$MBLoss(y'_{Ra}, y'_{Rh}, y'_{NEE}, GPP) = \sum_{i=1}^N \sum_{t=1}^{Tx} ReLU(|GPP_{t,i} + y'_{Ra,t,i} + y'_{Rh,t,i} + y'_{NEE,t,i}| - \beta_{MB} |y'_{Ra,t,i} + y'_{Rh,t,i}|) / Tx / N \quad (Eq. 17)$$

where $\beta_{MB} |y'_{Ra,t,i} + y'_{Rh,t,i}|$ is the maximum mass balance tolerance that varies with the predicted Ra and Rh, and β_{MB} is the tolerance fraction, set to be 0.01 in this study. Note that the predicted values of Ra and Rh are negative in order to be consistent with a positive direction from the atmosphere to the soil.”

- In Fig. 2, the authors show how the number of counties and flux towers influences the performance of the different models – it would be good also to have the number of observations associated with each sample, i.e. the number of annual or daily measurements per training set.

Potential solution: We have added this information to the Fig. 2 caption: “For yield training data, each county has a 20-year period of annual yield observations. For Reco and NEE training data, each site has daily observations during the observation period (varying by site, ranging from 5 to 19 years).”

- For Fig. 3, it would be better to have a full-factorial test of the contributions of GPP Data, Pretrain data from ecosys, and KG Loss. Maybe the authors did this already, and I misunderstood. For example, how well would a model perform if we combined ML with a knowledge-guided loss function? It seems the authors only added them in the last step.

Response: Thank you for the suggestion. In this revision, we have conducted full-factorial tests to investigate the impacts of different components of KGML-ag-Carbon on model performance, as shown in Fig. R6 (Fig. S11 in the supplementary materials). We have also revised Methods 5.6 and Section 3.2 to describe these additional tests. The results are consistent with our previous findings that the strategies of using GPP data as input and pretraining have a greater contribution to improving the performance of KGML-ag-Carbon than the other strategies, and their contributions are related to sample size. In the main text, we retain Fig. 3 to show the performance of five representative models including (1) ML, (2) ML + GPP, (3) ML + GPP + pretraining, (4) ML + GPP + pretraining + KG structure, and (5) ML + GPP + pretraining + KG structure + KG loss. This sequence of models best illustrates the direction of performance improvements. The main findings presented in Section 3.2 remain unchanged. Readers who are interested in the full-factorial results can refer to supplementary Fig. S11 for details.

Fig. R6. Full-factorial tests for KGML-ag-Carbon. (a-b) Impacts of different components of the model on annual corn and soybean yield prediction accuracy with models trained using different training sample sizes (either 5 or 40 counties out of 637 counties); (c-d) Impacts on daily Reco and NEE flux prediction accuracy after training with either 1 or 7 sites out of 11 sites. ML represents machine learning, PRE represents pretraining using synthetic data, KGL represents knowledge-guided loss, and STRU represents the knowledge-guided structure.

Details of revised section 3.2: “To understand the contribution of different strategies to improvements in the performance of KGML-ag-Carbon, we conducted full-factorial tests to include or exclude different KGML-ag-Carbon components, and selected five representative models to use in interpreting the results (Fig. 3, Method 5.6).” and “Other results of full-factorial tests and mass balance tests can be found in Fig. S11 and Fig. S12, respectively.”

Details of revised Methods 5.6: “To investigate the contributions of different KGML-ag-Carbon components to the final ready-to-go KGML-ag-Carbon performance, we conducted full-factorial tests for each component in the model, and tested the model performance on an out-of-sample dataset (Fig. S11). Specifically, we included or excluded four components:(1) using GPP data as input (GPP for short), (2) pretraining the model with synthetic data, (3) incorporating the KGML-ag-Carbon structure, and (4) implementing KG loss functions and the 5-step training strategies (if a structure is applicable). In total, 16 individual models were trained. The training and testing data are similar to the robustness experiment described above. Specifically, to determine the contributions to the yield (flux) predictions, we used training sets of 5 and 40 counties (1 and 7 sites) to train the models, referred to as small and large training sample sets, respectively. The optimized models were tested on out-of-sample data sets, which are NASS yields from 210 randomly selected counties, and Reco and NEE from 6 groups of EC flux tower sites (the models tested on one group were trained and validated with data from sites chosen from other groups). We calculated the mean and STD of the prediction accuracy for all the models from ensemble experiments and detected the performance changes by comparing the models with and without each KGML-ag-Carbon component (Fig. S11). To illustrate the factors that contribute to the KGML-ag-Carbon model performance, we selected five representative models from the 16 trained models to showcase the direction of performance improvement. These models include (1) ML, (2) ML + GPP, (3) ML + GPP + pretraining, (4) ML + GPP + pretraining + KG structure, and (5) ML + GPP + pretraining + KG structure+ KG loss (Fig. 3a-b).”

- The authors compare KGML-ag-carbon with potential runtimes of *ecosys*. However, they compare GPU and CPU runtimes. Nowadays, process-based models can be run in parallel on GPUs – the comparison may be unfair.

Response: While running process-based models on GPUs can significantly reduce the runtime, it requires intensive re-engineering of the code and adaptation to I/O and memory for the code to the GPU environment., Based on previous attempts to transfer other Earth system models from CPU to GPU (Bauer et al., 2021), this is not a trivial task. While models with explicit representations of biogeochemical processes, like *ecosys*, are an ideal tool to guide the development of models such as KGML-ag-Carbon, no similar models have been transferred to GPU. We have addressed the reviewer’s comments by discussing the potential of process-based models using GPUs in section 4.4: “While process-based models can now be accelerated using GPUs, this typically requires significant code redesign and rewriting⁵⁵. Unfortunately, *ecosys* is currently unable to run on GPUs.”

- The GPP product SLOPE is not a pure remote-sensing product (line 241) but has machine learning models included – it would be good to note that down for the reader

Response: We have revised the detailed description for GPP in Methods 5.2. “... the 250-m daily GPP product derived from machine learning models based on Soil-Adjusted Near-Infrared Reflectance of vegetation (SANIRv)³ ...”

- While the methods are described in great detail in the supplement it would be better to provide the code associated with this study to ensure reproducibility. There is also no mention of which framework was used to code KGML-ag-carbon and what kind of GPU was used for training.

Response: Thank you for pointing this out. We believe in ensuring the reproducibility of our work. The code for KGML-ag-carbon will be made publicly available upon publication. The PyTorch framework was used to code KGML-ag-carbon, and the model training was conducted on a system with two NVIDIA Tesla V100 GPUs. We have added a new section 5.9 for describing the code availability and GPU configuration information:

“5.9 Development environment description

We used Pytorch 1.6.0 (<https://pytorch.org/get-started/previous-versions/>, last access: 14 May 2023) and Python 3.7.11 (<https://www.python.org/downloads/release/python-3711/>, last access: 14 May 2023) as the programming environment for model development. In order to use a GPU to speed-up the training process, we installed the CUDA Toolkit 10.1.243 (<https://developer.nvidia.com/cuda-toolkit>, last access: 14 May 2023). A desktop with an NVIDIA 2080 super GPU was used for code development and testing. The training processes, which required a long time and big memory space, were conducted on the Mangi cluster (<https://www.msi.umn.edu/mangi>, last access: 14 May 2023) from the High-Performance Computing facility of the Minnesota Supercomputing Institute (HPC-MSI, <https://www.msi.umn.edu/content/hpc>, last access: 14 May 2023) with a two-way NVIDIA Tesla V100 GPU.”

Some typos:

Line 110: cheapercompared – space is missing

Response: We have corrected this.

Line 656: Fig. SX – number is missing

Response: It should be Fig. S9g-l. This has been fixed.

Reference:

- Al-Kaisi, M. M., & Kwaw-Mensah, D. (2020). Quantifying soil carbon change in a long-term tillage and crop rotation study across Iowa landscapes. *Soil Science Society of America Journal*. *Soil Science Society of America*, 84(1), 182–202.
- Batjes, N. H., Ribeiro, E., & van Oostrum, A. (2020). Standardised soil profile data to support global mapping and modelling (WoSIS snapshot 2019). *Earth System Science Data*, 12(1), 299–320.
- Bauer, P., Dueben, P. D., Hoefler, T., Quintino, T., Schulthess, T. C., & Wedi, N. P. (2021). The digital revolution of Earth-system science. *Nature Computational Science*, 1(2), 104–113.
- EPA. (2023). *Inventory of U.S. Greenhouse Gas Emissions and Sinks: 1990-2021*. U.S. Environmental Protection Agency, EPA 430-R-23-002.
<https://www.epa.gov/ghgemissions/inventory-us-greenhouse-gas-emissions-and-sinks-1990-2021>
- Forster, P., Storelvmo, T., Armour, K., Collins, W., Dufresne, J.-L., Frame, D., Lunt, D., Mauritsen, T., Palmer, M., Watanabe, M., Wild, M., & Zhang, H. (2021). *Chapter 7: The Earth's energy budget, climate feedbacks, and climate sensitivity*. Open Access Victoria University of Wellington | Te Herenga Waka. <https://doi.org/10.25455/WGTN.16869671.V1>
- Goidts, E., Van Wesemael, B., & Crucifix, M. (2009). Magnitude and sources of uncertainties in soil organic carbon (SOC) stock assessments at various scales. *European Journal of Soil Science*, 60(5),

723–739.

- Guan, K., Jin, Z., DeLucia, E. H., West, P., Peng, B., Tang, J., Jiang, C., Wang, S., Kim, T., Zhou, W., Griffis, T., Liu, L., Qin, Z., Margenot, A. J., Kumar, V., Bernacchi, C. J., Yang, W. H., Lee, D., Coppess, J. W., ... Yang, S.-J. (2022). *A framework for scalably quantifying field-level agricultural carbon outcomes*. <http://eartharxiv.org/repository/view/2905/>
- Ibrahim, M. A., Chua-Ona, T., Liebman, M., & Thompson, M. L. (2018). Soil organic carbon storage under biofuel cropping systems in a humid, continental climate. *Agronomy Journal*, *110*(5), 1748–1753.
- IPCC. (2019). *2019 Refinement to the 2006 IPCC Guidelines for National Greenhouse Gas Inventories* (E. Calvo Buendia, K. Tanabe, A. Kranjc, J. Baasansuren, M. Fukuda, N. S., A. Osako, Y. Pyrozhenko, P. Shermanau, & S. Federici (eds.)). IPCC.
- Jian, J., Vargas, R., Anderson-Teixeira, K., Stell, E., Herrmann, V., Horn, M., Kholod, N., Manzon, J., Marchesi, R., Paredes, D., & Bond-Lamberty, B. (2021). A restructured and updated global soil respiration database (SRDB-V5). *Earth System Science Data*, *13*(2), 255–267.
- Jin, V. L., Schmer, M. R., Wienhold, B. J., Stewart, C. E., Varvel, G. E., Sindelar, A. J., Follett, R. F., Mitchell, R. B., & Vogel, K. P. (2015). Twelve years of Stover removal increases soil erosion potential without impacting yield. *Soil Science Society of America Journal. Soil Science Society of America*, *79*(4), 1169–1178.
- Khan, S. A., Mulvaney, R. L., Ellsworth, T. R., & Boast, C. W. (2007). The myth of nitrogen fertilization for soil carbon sequestration. *Journal of Environmental Quality*, *36*(6), 1821–1832.
- Olson, K., Ebelhar, S. A., & Lang, J. M. (2014). Long-term effects of cover crops on crop yields, soil organic carbon stocks and sequestration. *Open Journal of Soil Science*, *04*(08), 284–292.
- Poffenbarger, H. J., Barker, D. W., Helmers, M. J., Miguez, F. E., Olk, D. C., Sawyer, J. E., Six, J., & Castellano, M. J. (2017). Maximum soil organic carbon storage in Midwest U.S. cropping systems when crops are optimally nitrogen-fertilized. *PloS One*, *12*(3), e0172293.
- Schmer, M. R., Jin, V. L., Wienhold, B. J., Varvel, G. E., & Follett, R. F. (2014). Tillage and residue management effects on soil carbon and nitrogen under irrigated continuous corn. *Soil Science Society of America Journal. Soil Science Society of America*, *78*(6), 1987–1996.
- Stanley, P., Spertus, J., Chiartas, J., Stark, P. B., & Bowles, T. (2023). Valid inferences about soil carbon in heterogeneous landscapes. *Geoderma*, *430*(116323), 116323.
- Stuart Chapin, F., III, Matson, P. A., & Mooney, H. A. (2011). *Principles of Terrestrial Ecosystem Ecology*. Springer Science & Business Media.
- Varvel, G. E. (2006). Soil organic carbon changes in diversified rotations of the western corn belt. *Soil Science Society of America Journal. Soil Science Society of America*, *70*(2), 426–433.
- Venterea, R. T., Baker, J. M., Dolan, M. S., & Spokas, K. A. (2006). Carbon and nitrogen storage are greater under biennial tillage in a Minnesota corn–soybean rotation. *Soil Science Society of America Journal. Soil Science Society of America*, *70*(5), 1752.
- Xu, S., Khandelwal, A., Li, X., Jia, X., Liu, L., Willard, J., Ghosh, R., Cutler, K., Steinbach, M., Duffy, C., Nieber, J., & Kumar, V. (2023). Mini-Batch Learning Strategies for modeling long term temporal dependencies: A study in environmental applications. In *Proceedings of the 2023 SIAM International Conference on Data Mining (SDM)* (pp. 649–657). Society for Industrial and Applied Mathematics.

Response to Reviewer #3

In this manuscript, the authors design an AI framework, Knowledge Guided Machine Learning (KGML), to calculate carbon capture in agriculture sites using data from an ecosystem model, *ecosys*, to pretrain the KGML and observations of yield and eddy covariance to fine tune the model. They perform a variety of experiments to examine the performance sensitivity to the size of the training data and compared the results with a simple ML model. They also performed additional experiments to explore alternative training, out-of-sample and extreme year data, and different data-splitting to understand how to reduce uncertainty. The KGML method outperformed the ecosystem model and pure ML model for capturing yields and EC fluxes. Finally, the authors use this approach to estimate carbon capture in agriculture fields. I have a few suggestions and questions for the authors to address, but overall, I find this paper to be an interesting new application of machine learning that can have useful approaches for a variety of biogeochemical modeling.

I'm not an expert in AI so I can't comment much on that element. I did get a little lost in some of the details of the methods since I am not familiar with KGML, but I was able to understand overall what the authors did. I liked how the authors essentially did a sensitivity study of the approach, comparing with pure ML, and against the ecosystem model itself. One additional suggestion would be to look at how the method performs with pretraining from a different ecosystem model. I think it outside the scope of this manuscript, and the authors discuss the importance of ecosystem model choice in Section 4.2. I wonder how well the *ecosys* model works outside the U.S. or with other crop types, such as wheat or rice. This discussion might be expanded more, especially since the *ecosys* model doesn't perform great compared with observations.

Response: Thank you for this suggestion. In theory, any process-based model can be used to provide synthetic data for pretraining, but certain models are preferred, such as those that simulate various environmental and management practices based on first-principle-based mechanistic approaches instead of empirical methods, and those that have been well-validated for diverse ecosystems and locations. We chose the *ecosys* model because it is one of the few models that explicitly represent processes based on biogeochemical and physical principles and has been well-validated for various crop types across the globe (Asseng et al., 2013; Grant et al., 2011; Zhou et al., 2021). We believe other models such as APSIM, DSSAT, EPIC, and CLM could be used to provide synthetic data for pretraining KGML-ag-Carbon as well, and the difference will likely be small if a good observational dataset is available for finetuning. Model intercomparison within the KGML framework is an interesting idea, but would require a significant collaborative effort and is therefore beyond the scope of this study.

To address the reviewer's comment, we have extended the discussion in Section 4.2 to include information on the performance of *ecosys* for other crop types and regions outside the US, and the potential to use other process-based models. We hope this will provide a more comprehensive understanding of the applicability of different PB models and their role in guiding KGML-ag-Carbon. The revised text is:

“Choosing a proper PB model as the scientific foundation for KGML development is critical. Although a huge number of PB models exist for ecosystem carbon cycle modeling, models that incorporate sufficiently explicit representations of processes and are well-validated have more potential to benefit the AI models, especially in situations with no or few real-world samples available to train the AI models. The PB model used in this study, *ecosys*, contains comprehensive first-principles descriptions of carbon

transformation and translocation processes in plants and soil, and has been well-validated for different crop types and regions^{13,37-40}. It provides valuable basic knowledge to guide the structure design and training of the KGML model. The benefits of *ecosys* in improving KGML-ag-Carbon's crop yield and carbon flux predictions were reflected in contribution tests as increased prediction accuracy (Fig. 3a-b), and reduced mass balance residue (Fig. 3c). Future work may involve testing different PB ecosystem models (e.g. well-validated models in Asseng et al. (2013)³⁷ and Sitch et al. (2015)³⁴) to explore the uncertainties arising from model selection for pretraining. However, this would require a significant collaborative effort.”

I'm not convinced that the model would be accurate enough to estimate carbon capture in an agriculture field given the lack of validation data. Discussion section 4.1 seems out of place and unnecessary.

Response: Thank you for your comment. To address these concerns, we have removed the basis risk calculation using Δ SOC that involves policy-sensitive US\$ values. Instead, we have added discussion on the need for accurate SOC measurements and focus on the benefits of high-resolution quantification of NEE (which is well-validated in this study) and Δ SOC. We have also added a discussion on the uncertainty of the intermediate model variables and the need for measurements to constrain them. We have updated section 4.1, Fig. 5 (shown here as Fig. R7), Fig. S13 (removed panel c about basis risk), and Methods 5.8 to reflect these changes.

We agree that the validation of Δ SOC is critical. However, the current accuracy of SOC measurements is limited by various sources of uncertainties, including spatial heterogeneity, sampling design, variability in bulk density, and variation in soil processing methods and laboratory assays (Goidts et al., 2009; Stanley et al., 2023) which lead to uncertainties exceeding 50% in SOC changes over a certain period of time. In addition, after an intensive literature search for existing studies across the US, we found that most accessible SOC data are static data (Batjes et al., 2020). We found some plot-level experimental data from individual studies (Al-Kaisi & Kwaw-Mensah, 2020; Ibrahim et al., 2018; Jin et al., 2015; Khan et al., 2007; Olson et al., 2014; Poffenbarger et al., 2017; Schmer et al., 2014; Varvel, 2006; Venterea et al., 2006), but they are not suitable for field-level Δ SOC validation, because of their relatively small spatial scale (~10 m), which does not match the field scale (~100 m). Furthermore, we have attempted to downscale the KGML-ag-Carbon model from field scale to plot scale to validate the Δ SOC using plot-level data. However, the downscaling would require explicit geographical location and weather conditions, which are rarely reported in the literature.

The revised Fig. 5 (shown here as R7):

Fig. R7. The impact of spatial resolution on NEE and Δ SOC quantification. (a-b) NEE absolute errors caused by using coarse resolution for carbon budget estimations. (c-d) Δ SOC absolute errors caused by using coarse resolution. 0.0025-degree resolution is treated as “field-level” ground truth.

Revised discussion section 4.1 is listed below:

“The field-level quantification of carbon budgets, crop yields, and Δ SOC produced using KGML-ag-Carbon (as demonstrated for the US Midwest) provides an accurate, cost-effective, and high-resolution product for potentially improving carbon sequestration assessments. We used the term absolute error to quantify the advantages of high-resolution carbon budgets generated by KGML-ag-Carbon. Previous approaches often assess the carbon budget using factors that are averaged over a certain region, and absolute error describes the potential consequences resulting from mismatches between local-scale factors and the regional baseline. As used here, absolute error represents the difference of NEE or Δ SOC that can be attributed to individual fields using a high-resolution carbon budget product compared to larger-scale estimates based on lower-resolution products (Methods 5.8). Generally, the absolute error decreases quickly with increasing resolution of the carbon budget product for the US Midwest (Fig. 5 and Fig. S13). For example, when using a 1x1°-resolution product, the uncertainties in estimated NEE and Δ SOC of individual fields are up to 90 gC/m²/year and 101 gC/m²/year (75% quantile), respectively (Fig. 5). Based on this calculation, those absolute errors will lead to a total 5.6 and 6.2 GgC/year (1 Gg = 10⁹ g) errors for NEE and Δ SOC quantification, respectively, in the US Midwest. However, if using a product with a relatively high resolution of 0.005°, maximum estimate uncertainties decrease to 27 gC/m²/year for both NEE and Δ SOC (75% quantile). The total errors

would be reduced by 70% and 73% purely by changing the resolution of NEE and Δ SOC quantifications, respectively (Fig. 5). More detailed results about the benefits of high-resolution NEE, crop yield, and Δ SOC quantifications are given in Fig. S13. To be noted, the KGML-ag-Carbon employs a mass balance approach to estimate Δ SOC from NEE and yield, which are estimated by integrating all available data, including weather forcing, soil properties (which include static SOC), crop type, and remotely sensed GPP, and are well-validated by observations. This approach allows us to make the best use of existing data to estimate the regional Δ SOC at low cost and high resolution, even in the absence of field-level measurements. However, it is important to acknowledge that the lack of validation using field-level measurements of Δ SOC limits the certainty of our estimates, as such measurements are often scarce and associated with large uncertainties. For instance, at 250-m scales, uncertainty in SOC arises from lab measurement error (up to 12%), spatial sampling error (up to 50%), and resampling error (up to 45%)^{35,36}. The uncertainty of Δ SOC over a long period of time will be even larger. Including measurements with such uncertainties in the KGML-ag-Carbon training data may decrease the reliability of other components such as NEE and yield, which have relatively lower measurement errors. It's also worth noting that while the NEE, Reco, and crop yield in the KGML-ag-Carbon are well-constrained, the intermediate variables like Ra, Rh, and crop residue may still contain high uncertainty due to a lack of direct observational data. These variables, however, are crucial for a better understanding of the underlying mechanisms. Therefore, while the KGML-ag-Carbon offers a feasible method for reducing carbon budget quantification errors due to its high resolution (250 m), high accuracy (Fig. 2-4), and low computational cost, it also highlights the need for accurate field-level Δ SOC measurements to improve the reliability of Δ SOC quantification, and the need for accurate measurements of Ra, Rh, and crop residue to constrain the underlying processes.”

The field sites where data is collected (particularly the flux sites) don't have much information about them in the manuscript. All those plots have different crops and management practices (e.g., rotation, tillage, fertilizer application, irrigation, etc.). The reader doesn't find out until section 4.3 that these aren't taken into account in the model. While I like that the authors suggested a future application could be to quantify management, it should be emphasized that management practices are strong drivers of carbon inputs. There should be some data at these sites that can be used as model input or at least include a comment on how that will influence the results.

Response: Thank you for your valuable comments. We agree that management practices are strong drivers of carbon inputs, and we acknowledge that their inclusion could improve the accuracy of the model. However, at the regional scale, management practices data are often unavailable in a spatially explicit form, which is why we chose to rely on remotely sensed GPP as a constraint to indirectly account for the effects of management practices. While GPP does not provide direct information on management practices, it can still reveal vegetation status changes that result from different practices. However, we agree that future studies should aim to include more detailed management practice data to improve the model. We have added a detailed discussion in section 4.3. The revised part of section 4.3:

“In the current KGML-ag-Carbon framework, some important management practices such as fertilizing, irrigation, and tillage, have not been explicitly considered in the model due to a lack of location-specific management information. It is currently assumed that the incorporation of remotely sensed GPP data in the KGML-ag-Carbon model can largely capture local variations in carbon fluxes due to management practices. Remote sensing data has shown potential for assessing local management practices such as

cover cropping⁴¹, tillage⁴², and irrigation⁴³. Recent advances in AI-based inverse modeling, such as Knowledge-Guided Self-Supervised Learning⁴⁴, may further improve estimates where management information is unknown. However, it should be noted that such methods are still in the early stages of development. Additionally, it is important to consider that management practices aimed at enhancing carbon storage in upland agroecosystems may inadvertently lead to an increase in other GHG emissions. For instance, while increasing the use of N fertilizers can improve carbon sequestration, it can also contribute to higher N₂O emissions, partially offsetting the climate mitigation effect. Therefore, to conduct a comprehensive assessment of management impacts on GHG emissions (mostly CO₂ and N₂O) from upland agroecosystems, the N cycle needs to be incorporated into the framework due to the non-trivial impacts of N₂O on the climate and the interactions between C and N¹². However, incorporating C-N interactions is complicated because the underlying processes are unclear and comprehensive measurements of both C- and N-related fluxes and soil states needed to validate any new model are lacking.”

Another consideration for this model is that ML can only learn from the past and extreme events are often not included but they can be important for future estimates of carbon fluxes. For example, carbon fertilization is generally not included in ecosystem models, nor is crop response to extreme heat or flash droughts. Including these in ML models could improve our prediction of crop models.

Response: Thank you for your valuable comments and suggestions. We acknowledge that ML models have limitations for extrapolation, but argue that out-of-sample prediction is a strength of the KGML framework. In fact, we found that the KGML-ag-Carbon model captures yields better than pure ML and PB models in historical extreme events, such as those that occurred in 2002, 2003, and 2012 (Fig. R8, taken from Fig. S9 in the supplementary material). The better performance by KGML-ag-Carbon is mainly because pretraining using *ecosys*-generated synthetic data helps the model learn certain mechanisms from extreme year scenarios.

Fig. R8. A comparison of robust test results from a pure ML model (blue), the KGML-ag-Carbon model before finetuning (green, performing the same as *ecosys* model), and the KGML-ag-Carbon model (red) for corn and soybean yield predictions in extreme years. All models were constrained by remotely sensed GPP.

In section 3.1, we have added a sentence to describe the model's ability to capture extreme events: “We note that the KGML-ag-Carbon can outperform pure ML and process-based models in predicting yield in extreme years (Fig. S9e-f), primarily because it is constrained by both observations and synthetic data generated from the PB model.”

We agree that fully capturing extreme events is an important future direction to explore, and have added discussion in sections 4.3 and 4.4. Details of revised section 4.3:

“... Moreover, while the KGML-ag-Carbon can accurately predict yield in extreme years (Fig. S9e-f), the impact of extreme weather conditions like heatwaves or flash droughts on the agroecosystem remains unclear. Enriching KGML-ag-Carbon with simulations of intermediate environmental variables, such as canopy temperature and soil moisture, alongside the carbon budget quantification, could potentially help dissect and elucidate the effects of extreme weather. If a reliable KGML tool was available to quantify the influences of different management practices and extreme weather on GHG emissions and productivity, it would be possible to develop reinforcement learning approaches^{45,46} for optimizing management practices to maximize environmental and economic rewards.”

We added discussion on other potential applications of KGML-ag-Carbon to section 4.4:

“KGML-ag-Carbon can be used for numerous other tasks including: predicting other target variables (e.g. N and P cycles), estimating C outcomes over larger regions (e.g. the entire U.S.), simulating carbon dynamics in different ecosystems (e.g. natural forests), and assessing impacts of management practices (e.g. cover cropping, tillage) and extreme weather (e.g. extreme heat or flash droughts).”

The methods section indicates that most of the data is used for pre-training and training purposes. Do the authors worry about overfitting the model? Also, what is the difference between “easier” and “harder” samples (Line 623)

Response: Thank you for the comments. To mitigate the potential for overfitting, we applied dropout regularization in the KGML-ag-Carbon model, and described this in the section Methods 5.3:

“20% of the output hidden states from GRU cells are randomly dropped by replacing them with zero values (the so-called 20% dropout) to avoid overfitting.”

Additionally, we used knowledge-guided (KG) loss functions to constrain model predictions based on prior knowledge, which reduces the chance of overfitting by limiting the model's ability to fit the training data too closely. The KG losses for each training step have been described in detail in Methods 5.4 and supplemental material.

When training with the self-paced learning (SPL) method, we calculate the mean square error (MSE) between the model prediction and the observed value for each sample in every epoch. "Easier" samples are those whose MSE is below a predefined threshold, indicating that they are easier to fit accurately with the current model than the "harder" samples, whose MSE is above the threshold. SPL selects easier samples first during the early stages of training to facilitate faster convergence to a better local minimum, while gradually adding harder samples as training progresses to further refine the model. In this revision, we have added references in Method 5.4 to explain the SPL method: “We used a mean-square-error (MSE)-based self-paced learning (SPL) method^{13,14} to build our training losses to train the model from “easier” samples to “harder” samples (Text S1).”

More detailed information about our implementation of SPL and the concept of easier and harder samples can be found in Text S1 in supplemental material:

“The loss function for step 1 with mean-square-error (MSE) based self-paced learning (SPL) method^{1,2} can be expressed as:

$$Loss_{step1} = \sum_{i=1}^N v_{Ra,i} \sum_{t=1}^{Tx} (y_{Ra,t,i} - y'_{Ra,t,i})^2 / Tx / N + \sum_{i=1}^N v_{yield,i} (y_{yield,i} - y'_{yield,i})^2 / N \quad (Eq. 12)$$

$$v_{Ra,i} = \begin{cases} 1 & \text{when } \sum_{t=1}^{Tx} (y_{Ra,t,i} - y'_{Ra,t,i})^2 / Tx < \lambda \\ 0 & \text{when } \sum_{t=1}^{Tx} (y_{Ra,t,i} - y'_{Ra,t,i})^2 / Tx > \lambda \end{cases} \quad (Eq. 13)$$

$$v_{yield,i} = \begin{cases} 1 & \text{when } (y_{yield,i} - y'_{yield,i})^2 < \lambda \\ 0 & \text{when } (y_{yield,i} - y'_{yield,i})^2 > \lambda \end{cases} \quad (Eq. 14)$$

$$\lambda = \lambda_0 \prod_1^{epoch} K_{growth} \quad (Eq. 15)$$

where the $Loss_{step1}$ is the loss function used in step 1; $y_{Ra,t,i}$ and $y'_{Ra,t,i}$ are the synthetic and predicted Ra flux of the i^{th} sample at time t , respectively; $y_{yield,i}$ and $y'_{yield,i}$ are the synthetic and predicted annual yield of the i^{th} sample, respectively. $v_{Ra,i}$ and $v_{yield,i}$ are the integer multipliers for Ra and yield to decide whether we use the i th sample to calculate the loss; N is the total number of samples for a certain simulation; λ is the threshold with an initial value of λ_0 (0.05) and a growth rate of K_{growth} (1.02). The SPL-MSE method can start training the model with “easier” samples (those with a distance between the synthetic data and the prediction below the threshold λ). As the training epoch increases, the λ will increase, and more “difficult” samples will be involved. ”

In the abstract, line 23, do the authors mean remote data instead of satellite?

Response: Yes, we meant "remote sensing" instead of "satellite". We have revised the abstract accordingly.

Line 110: cheapercompared should be two words

Response: We have corrected this.

Reference:

- Al-Kaisi, M. M., & Kwaw-Mensah, D. (2020). Quantifying soil carbon change in a long-term tillage and crop rotation study across Iowa landscapes. *Soil Science Society of America Journal. Soil Science Society of America*, 84(1), 182–202.
- Asseng, S., Ewert, F., Rosenzweig, C., Jones, J. W., Hatfield, J. L., Ruane, A. C., Boote, K. J., Thorburn, P. J., Rötter, R. P., Cammarano, D., Brisson, N., Basso, B., Martre, P., Aggarwal, P. K., Angulo, C.,

- Bertuzzi, P., Biernath, C., Challinor, A. J., Doltra, J., ... Wolf, J. (2013). Uncertainty in simulating wheat yields under climate change. *Nature Climate Change*, 3(9), 827–832.
- Batjes, N. H., Ribeiro, E., & van Oostrum, A. (2020). Standardised soil profile data to support global mapping and modelling (WoSIS snapshot 2019). *Earth System Science Data*, 12(1), 299–320.
- Goidts, E., Van Wesemael, B., & Crucifix, M. (2009). Magnitude and sources of uncertainties in soil organic carbon (SOC) stock assessments at various scales. *European Journal of Soil Science*, 60(5), 723–739.
- Grant, R. F., Kimball, B. A., Conley, M. M., White, J. W., Wall, G. W., & Ottman, M. J. (2011). Controlled Warming Effects on Wheat Growth and Yield: Field Measurements and Modeling. *Agronomy Journal*, 103(6), 1742–1754.
- Ibrahim, M. A., Chua-Ona, T., Liebman, M., & Thompson, M. L. (2018). Soil organic carbon storage under biofuel cropping systems in a humid, continental climate. *Agronomy Journal*, 110(5), 1748–1753.
- Jin, V. L., Schmer, M. R., Wienhold, B. J., Stewart, C. E., Varvel, G. E., Sindelar, A. J., Follett, R. F., Mitchell, R. B., & Vogel, K. P. (2015). Twelve years of Stover removal increases soil erosion potential without impacting yield. *Soil Science Society of America Journal. Soil Science Society of America*, 79(4), 1169–1178.
- Khan, S. A., Mulvaney, R. L., Ellsworth, T. R., & Boast, C. W. (2007). The myth of nitrogen fertilization for soil carbon sequestration. *Journal of Environmental Quality*, 36(6), 1821–1832.
- Olson, K., Ebelhar, S. A., & Lang, J. M. (2014). Long-term effects of cover crops on crop yields, soil organic carbon stocks and sequestration. *Open Journal of Soil Science*, 04(08), 284–292.
- Poffenbarger, H. J., Barker, D. W., Helmers, M. J., Miguez, F. E., Olk, D. C., Sawyer, J. E., Six, J., & Castellano, M. J. (2017). Maximum soil organic carbon storage in Midwest U.S. cropping systems when crops are optimally nitrogen-fertilized. *PloS One*, 12(3), e0172293.
- Schmer, M. R., Jin, V. L., Wienhold, B. J., Varvel, G. E., & Follett, R. F. (2014). Tillage and residue management effects on soil carbon and nitrogen under irrigated continuous corn. *Soil Science Society of America Journal. Soil Science Society of America*, 78(6), 1987–1996.
- Stanley, P., Spertus, J., Chiartas, J., Stark, P. B., & Bowles, T. (2023). Valid inferences about soil carbon in heterogeneous landscapes. *Geoderma*, 430(116323), 116323.
- Varvel, G. E. (2006). Soil organic carbon changes in diversified rotations of the western corn belt. *Soil Science Society of America Journal. Soil Science Society of America*, 70(2), 426–433.
- Venterea, R. T., Baker, J. M., Dolan, M. S., & Spokas, K. A. (2006). Carbon and nitrogen storage are greater under biennial tillage in a Minnesota corn–soybean rotation. *Soil Science Society of America Journal. Soil Science Society of America*, 70(5), 1752.
- Zhou, W., Guan, K., Peng, B., Tang, J., Jin, Z., Jiang, C., Grant, R., & Mezbahuddin, S. (2021). Quantifying carbon budget, crop yields and their responses to environmental variability using the ecosys model for U.S. Midwestern agroecosystems. *Agricultural and Forest Meteorology*, 307(108521), 108521.

Response to Reviewer #4

In this work, the authors proposed a Knowledge-Guided Machine Learning (KGML) framework and demonstrated that KGML can outperform existing process-based or ML models in quantifying carbon cycles. In general, none of the key parts in KGML framework (i.e., knowledge-guided loss, synthetic data pre-training, the use of indirect data) makes a novel contribution to existing machine learning techniques. Although the combined use of all those parts hasn't been explored by existing works yet, this manuscript presents an overall performance of KGML without clearly studying the contribution of each part (e.g., whether the sequence of including different parts would impact the contributions in Figs. 3ab, why knowledge-guided loss always has a very small contribution in Fig. 3). In addition, the descriptions of some key parts (e.g., knowledge-guided losses, and the use of indirect data) are not clear. In terms of quantifying carbon cycles, this manuscript does not provide a clear description of the case studies and the general motivation for studying them. The quality of the figures in the manuscript should be improved as well. Therefore, the reviewer cannot support the publication of this manuscript in Nature Communications.

Response: We appreciate your feedback and have carefully considered these concerns. Here are our responses to address your points:

While some of the individual components of the KGML framework may not be novel in the computer science domain, our study presents the first systematic method to deeply integrate AI with a process-based model and multi-source data to quantify the agroecosystem carbon cycle with very high spatial and temporal resolutions. This topic has broader implications for both technical advances and sustainable development. Hybrid modeling, which integrates AI with process-based models, is predicted to dominate Earth system modeling research in the next 5-10 years (AI4ESP, 2021; Reichstein et al., 2019). However, efforts to develop hybrid models are still in a nascent stage and existing case studies often have a limited scope on relatively tractable systems with well-known physical processes (Irrgang et al, 2021). Developing hybrid modeling for agroecosystems, which involves many poorly-known biophysical and biogeochemical processes, poses challenges at a much tougher level. Our KGML framework successfully demonstrates a protocol for developing a performance-guaranteed hybrid model that integrates AI, process-based modeling, and remote sensing and in-situ observations. Additionally, we have provided a high-accuracy, cost-effective, and high-resolution method for quantifying and predicting the agroecosystem carbon budget. Such a method has broader impacts because it quantifies carbon outcomes in response to management at decision-relevant scales, thereby informing spatially-explicit solutions for climate change mitigation while ensuring food production.

Moreover, we have made innovative contributions to existing machine learning methodologies in this study by developing a neural network architecture that recognizes the causality among physical and biogeochemical variables. Current research often designs knowledge-guided (KG) loss functions based on observed variables and final predictions, rooted in established physical laws such as the conservation of energy and mass (Willard et al., 2022). However, numerous physical and biogeochemical relationships involve intermediate variables that are unobserved or partially observed, like R_a , R_h , and NEE in the mass conservation law employed in our work. Our method enables the representation and estimation of these unobserved or partially observed variables from intermediate neural network layers. It also allows for the creation of KG loss functions that utilize both final predicted and intermediate variables. The use of synthetic data generated from the PB model during the training phase is crucial for maintaining these causal relationships within the architecture.

In short, our work represents significant progress towards the next generation of Earth System Models using the hybrid modeling concept.

The novelty and importance of our study have also been acknowledged by all other reviewers, such as:

Reviewer #1: “In my opinion the novelty of this work is in the systematic evaluation of various forms of knowledge guidance and the connection of quantitative improvements in predictions to the carbon market.”

Reviewer #2: “The authors have developed, trained and tested this framework for modelling fine-scale carbon budgets and crop yields in the Midwest of the USA. They have done detailed tests and their line of argumentation is overall easy to follow. Overall, the authors make a sound case for their KGML-ag-Carbon framework.”

Reviewer #3: “...overall, I find this paper to be an interesting new application of machine learning that can have useful approaches for a variety of biogeochemical modeling.”

Regardless of our argument above, we have implemented the suggestion to conduct full-factorial tests to investigate the contributions of different parts of the KGML-ag-Carbon framework (Fig. R9, on the next page), including assimilation of GPP, pretraining, KG structure, and KG loss. To reflect the changes, we have revised Fig. S11 (shown here as Fig. R9), section 3.2, and Section 5.6 accordingly. The results are consistent with our previous findings that GPP and pretraining make a greater contribution to improving KGML-ag-Carbon performance than the other strategies, and all contributions are related to sample size. Therefore, we retain Fig. 3 in the main text to show the performance of five representative models including (1) ML, (2) ML + GPP, (3) ML + GPP + pretraining, (4) ML + GPP + pretraining + KG structure, and (5) ML + GPP + pretraining + KG structure + KG loss. This sequence of models best illustrates the direction of performance improvements. In addition, we found that KG loss greatly enhances performance when the model lacks certain components (e.g. ML+GPP vs ML+GPP+KGL in Fig. R9a and b), but only when used together with GPP. KGL is closely interconnected with other components, implying that their collective use is necessary for robust reduction of mass balance residue and to capture the complex temporal dynamics of carbon fluxes, as shown in Fig. 3c and d.

In addition, we have extended the Methods section to include all model development processes, data processing, and experiment design details to provide a clearer and more comprehensive description of our study. We have also revised the introduction to provide a clearer description of the case studies and the motivation for studying them. Our motivation is to develop a high-resolution product from KGML-ag-Carbon that can accurately estimate carbon fluxes at a regional scale and provide important information for decision-makers in agriculture and climate mitigation.

In summary, we believe that our study is a significant contribution to the field of earth system carbon cycle modeling and the application of AI techniques for agricultural carbon budget analysis.

Fig. R9. Full-factorial tests for KGML-ag-Carbon. (a-b) Impacts of different components of the model on annual corn and soybean yield prediction accuracy with models trained using different training sample sizes (either 5 or 40 counties out of 637 counties); (c-d) Impacts on daily Reco and NEE flux prediction accuracy after training with either 1 or 7 sites out of 11 sites. ML represents machine learning, PRE represents pretraining using synthetic data, KGL represents knowledge-guided loss, and STRU represents the knowledge-guided structure.

Reference:

- AI4ESP. (2021). *Artificial Intelligence for Earth System Predictability*. AI4ESP. <https://www.ai4esp.org/>,
Last accessed: May 14, 2023
- Reichstein, M., Camps-Valls, G., Stevens, B., Jung, M., Denzler, J., Carvalhais, N., & Prabhat. (2019).
Deep learning and process understanding for data-driven Earth system science. *Nature*, 566(7743),
195–204.
- Willard, J., Jia, X., Xu, S., Steinbach, M., & Kumar, V. (2022). Integrating Scientific Knowledge with
Machine Learning for Engineering and Environmental Systems. *ACM Computing Surveys*.
<https://doi.org/10.1145/3514228>

Reviewers' Comments:

Reviewer #1:

Remarks to the Author:

The authors have responded in detail to my comments and those of the other reviewers and edited the manuscript accordingly. In my opinion the paper is suitable for publication.

Reviewer #2:

Remarks to the Author:

I reviewed the revised manuscript and the authors' responses to my remarks and criticisms. While I appreciate that the authors removed the hard-to-follow statements on 500\$ million dollars savings, some of my other comments were not adequately addressed. I will highlight them again and explain why these issues should be addressed better.

Realism of Δ SOC

This is my most important concern. The authors have not taken up my suggestion to plot Δ SOC as a map and histogram.

This should be done as it is the main outcome of this study. The readers and reviewers should see what KGML-ag-Carbon predicts as carbon sequestration in these agroecosystems (in g C m⁻² yr⁻¹).

The authors also do not contrast their estimate of Δ SOC with machine learning-based estimates of SOC, e.g. from SoilGrids https://www.soilgrids.org/#!/?layer=TAXNWRB_250m&vector=1.

What is the percentage increase in SOC that KGML-ag-Carbon is predicting per year? Just use SoilGrids as a reference.

This is needed for me to trust the outcome of this model.

Other concerns

I also have concerns regarding the openness of code associated with the paper upon publication. There is substantial mismatch between what the authors write in the code and software submission checklist and the main text:

Why is the code 'Proprietary and confidential'? From your code and software submission checklist.

Why is it prohibited to copy/distribute/modify the files? From your code and software submission checklist.

In the main text you write: 'We will provide source code of data processing and executable python library of KGML-ag Carbon framework in the final published version for running demo data through open archive.' What is the open archive you are planning to use?

Software availability:

Will you plan to provide software that allows forward runs? If so, please change the respective documents in the submission system.

Why do you only want to provide predictions at county level (L 1095)? SoilGrids.org shows that is possible to provide 250m resolution data at a global scale.

I (reviewer #2) was asked to evaluate the responses of the authors to the concerns of reviewer #3:

Point 2 – carbon capture in agriculture

The response of the authors mainly consists of removing the monetary estimates of the effects resolution on DeltaSOC estimates. The authors should show the direction of DeltaSOC estimates and how much soil carbon is increasing or decreasing compared to initial stocks (from SoilGrids.org for example). I came across this study with the ecosys model (<https://www.sciencedirect.com/science/article/pii/S0016706122005614>) which should be used for comparison of DeltaSOC estimates.

The line of argumentation of the authors that SOC measurements will be too inaccurate to assess DeltaSOC, is insufficient. They should at least compare their rates of carbon sequestration or loss with literature values to show the realism of their DeltaSOC values. In Figure 4 of the above-mentioned study, DeltaSOC values are reported. How do they compare with the KGML approach? Figure 4d of Georgiou et al., 2022 (Georgiou K, Jackson RB, Vindušková O et al. (2022) Global stocks and capacity of mineral-associated soil organic carbon. Nature Communications, 13.) also report some carbon accrual rates. How do these compare with DeltaSOC from KGML? A literature survey should give further data on DeltaSOC values.

In section 5.8, the authors still use the carbon credit wording. Furthermore, it is not clear how the authors did the aggregation from 0.0025 degrees up to 1 degrees to quantify the error from coarse resolution. At first it was my impression that the error comes from aggregation in the covariates but from section 5.8 it seems like the error just comes from the way the 250m maps are aggregated. It should be possible to do the aggregation and conserving mass.

Point 3 - management impacts

In their new text, the authors argue that "incorporating C-N interactions is complicated because the underlying processes are unclear and comprehensive measurements of both C- and N-related fluxes and soil states needed to validate any new model are lacking". I think this is somewhat inaccurate since ecosys already has an N cycle and there are KGML variants on N2O.

Point 4 – past extreme events in the estimates

The authors' argumentation makes sense. I would ask for more clarification on the test setup for the extreme years. Is this with training on corn yield at all counties except 2002, 2003, 2012? Does the training see the NEE in these years? How does KGML perform with regard to NEE in extreme years?

Point 5- overfitting the model

Here, I would ask the authors to clarify how they prevented leakage of the pre-training covariates to the test set? In other words, in a clean setup, you should pretrain on ecosys in a period that will not be used for testing. As far as I see this has not been done and the periods of pretraining and testing are overlapping.

Furthermore, in Figure S9 g and S9 h, it seems like the test was on all EC flux towers - how did the authors perform the tests? You cannot include the training site into the test set. Here more clarification is needed if the authors prevented this. Similarly, it is not clear if Fig. 2 potentially has a similar problem.

Reviewer #3:
None

Reviewer #4:
None

Response Letter

Please be aware of the formatting of all responses:

1. Reviewer comment in **black**, response in **blue** and quotation from the main text in **red**;
2. The referred sections are based on the clean version of the revised manuscript, not the old or track change version.

Response to Reviewer #2

Comment: I reviewed the revised manuscript and the authors' responses to my remarks and criticisms. While I appreciate that the authors removed the hard-to-follow statements on 500\$ million dollars savings, some of my other comments were not adequately addressed. I will highlight them again and explain why these issues should be addressed better.

Response: We greatly appreciate your feedback and input. In this revision, we have diligently addressed these concerns and believe the scientific quality and clarity of the manuscript is significantly improved. Specifically, we have added a Δ SOC map, histogram, Soilgrids comparison, and evaluation with SOC measurements. We have also shared relevant code and data. Please see our detailed point-by-point response below.

Comment: Realism of Δ SOC

This is my most important concern. The authors have not taken up my suggestion to plot Δ SOC as a map and histogram.

This should be done as it is the main outcome of this study. The readers and reviewers should see what KGML-ag-Carbon predicts as carbon sequestration in these agroecosystems (in g C m⁻² yr⁻¹).

The authors also do not contrast their estimate of Δ SOC with machine learning-based estimates of SOC, e.g. from SoilGrids https://www.soilgrids.org/#!/layer=TAXNWRB_250m&vector=1.

What is the percentage increase in SOC that KGML-ag-Carbon is predicting per year? Just use SoilGrids as a reference.

This is needed for me to trust the outcome of this model.

Response: Thank you for clarifying these points. In this revision, we have included a map showing the spatial distribution of annually averaged Δ SOC from 2000 to 2020, along with the percentage fraction of total SOC stock from SoilGrids, as depicted in Fig. S14 (as Fig. R1 below). We utilized the corn and soybean fraction to exclusively focus on regions where corn and soybean cultivation accounts for more than 50% of the area. This selection also aligns with the training scope of our model and reduces the mixed pixel effect of remotely sensed GPP input from other ecosystems, such as forests, grasslands, and other agroecosystems. Fig. R1c-d shows that the majority of changes fall within the -0.5% to 0.5% C/year range (86%, mostly in line with other experimental studies), with notable patterns, including a decline in SOC across southern Minnesota, northern Iowa, and northern eastern Illinois, as well as an increase in the southern US Midwest. However, the latter increase in southern US Midwest SOC might be overestimated due to three potential factors: 1) Lower corn and soybean fractions (<60%, Fig. S1, as Fig. R2 below) cause stronger mixed pixel effects when utilizing remotely sensed GPP as an input; 2) The absence of EC flux tower data (indicated by blue stars in Fig. R2) limits the constraining of NEE predictions, so that the relatively higher temperatures in the south may contribute to the observed overestimation. and 3) The mass balance approach does not exclude the crop residues in the estimation of Δ SOC, and so far there's no good data to separate crop residues. To help readers better understand our intention, we have also enhanced the discussion on Δ SOC estimation and its uncertainties in discussion section 4.1 and added an experiment description in Method 5.8.

We wish to emphasize that while the Δ SOC distribution presented here provides valuable insights, the primary focus of the paper is on high-resolution carbon budget estimation benefits, demonstrated through the NEE and Δ SOC absolute error estimation in Fig. R2. It's crucial to note that the field-level (~250 m) Δ SOC measurements available to support effective training or validation for KGML-ag-Carbon Δ SOC

estimations are extremely limited within the US Midwest agroecosystems; despite this, we have conducted a comprehensive literature review and validation using nearly all available Δ SOC measurements in the US Midwest following your insightful suggestions (details and references given in responses below). Specifically, the limitations of the data include: (1) lack of key environmental drivers (e.g. localized GPP, weather); (2) scale mismatch (~ 250 m vs. ~ 10 m); and (3) large uncertainty in SOC measurements (Stanley et al. 2023; Goidts et al. 2009). Therefore, we acknowledge that the current KGML-ag-Carbon Δ SOC estimation carries inherent uncertainty, but the sources cannot be clearly identified due to these data limitations. Non-trivial efforts must first be undertaken to conduct comprehensive long-term field-level measurement of Δ SOC in agroecosystems, as outlined in (Guan et al. 2023).

Figure R1. The spatial distributions of (a) the Δ SOC derived from KGML-ag-Carbon estimated carbon budgets for the whole soil profile during 2000-2020; (b) the SOC stock derived from SoilGrid organic carbon density (OCD) in each layer within 0-200 cm depth of soil (representing the whole soil profile); and (c-d) the distributions of percentage fraction of the estimated Δ SOC compared to the SoilGrids SOC stock.

Figure R2. Study region (red lines highlighted region) and the location of agroecosystem eddy-covariance sites used for the KGML-ag-Carbon model finetune and validation.

Revised discussion section 4.1 is listed below:

“Besides, we have compared Δ SOC estimations from KGML-ag-Carbon with ML-based SOC stock estimations at 250 m resolution from SoilGrids³⁵ in Fig. S14 (Methods 5.8). The results show that the majority of changes fall within the -0.5% to 0.5% C/year range (86%), which is broadly consistent with the observed ranges in other experimental studies³⁶⁻⁴¹. Notable patterns include a decline in SOC across southern Minnesota, northern Iowa, and northern eastern Illinois, as well as an increase in the southern US Midwest. However, the observed increase in southern US Midwest SOC may be overestimated due to three factors: 1) Lower corn and soybean fractions (<60%, Fig. S1) intensify mixed pixel effects when using remotely sensed GPP; 2) The lack of EC flux tower data (blue stars in Fig. S1) limits the constraining of NEE predictions; and 3) The mass balance approach includes crop residues in Δ SOC estimation and so far there’s no good data to separate crop residues.”

Revised Method 5.8 is listed below:

“We have visualized the spatial distribution of KGML-ag-Carbon estimated Δ SOC during the period of 2000-2020 (Fig. S14a) and compared them with the SOC stocks derived from the SoilGrids (Fig. S14c-d). Specifically, the SoilGrids SOC stock (Fig. S14b) has been derived from organic carbon density (OCD) in each layer of 200 cm soil depth at 250 m resolution¹⁶. We used corn and soybean fractions from CDL and CSDL data to identify regions exclusively for these agroecosystems (fraction > 0.5), which aligns with our model’s current training scope and reduces the mixed pixel effect of inputted remotely sensed GPP from other ecosystems.”

Other concerns

Comment: I also have concerns regarding the openness of code associated with the paper upon publication. There is substantial mismatch between what the authors write in the code and software submission checklist and the main text:

Why is the code ‘Proprietary and confidential’? From your code and software submission checklist.

Why is it prohibited to copy/distribute/modify the files? From your code and software submission checklist.

Response: The designation of "Proprietary and confidential" for our code/data license was intended solely to ensure the security of the materials during the review process, as the manuscript and associated data have not been published at this stage. Also, while we welcome academic research derived from our work, we do not wish for our model to be used to support commercial activities. We want to clarify that we have assembled the relevant code and sample data for all experiments presented in the manuscript, which is sufficient for academic research purposes, and they will be publicly accessible with a proper open-source license such as MIT license once the paper is published. This will facilitate test runs and facilitate openness and reproducibility. We have updated the document “Code and Software Submission Checklist” to reflect this.

Comment: In the main text you write: ‘We will provide source code of data processing and executable python library of KGML-ag Carbon framework in the final published version for running demo data through open archive.’ What is the open archive you are planning to use?

Response: We will use zenodo.org to permanently store the relevant code (i.e. packaged Python libraries for the KGML-ag-Carbon model and Jupyter Notebook files for data processing and running the model) along with demo data for all experiments presented in this study.

Software availability:

Comment: Will you plan to provide software that allows forward runs? If so, please change the respective documents in the submission system.

Response: Yes, we will provide code, a well-constrained KGML-ag-Carbon model, and demo data for forward runs and we have changed the respective documents in the submission system accordingly.

Comment: Why do you only want to provide predictions at county level (L 1095)? SoilGrids.org shows that is possible to provide 250m resolution data at a global scale.

Response: The decision to provide county-level predictions instead of 250 m resolution data is based on privacy considerations. The final maps of flux variables have a resolution high enough to distinguish individual farmers' fields, which could be used to derive indicators related to the US agricultural carbon market, and this could raise questions or disputes. It should be noted US farmers in general are very conservative about releasing information about their farms, especially when they are tied to subsidies and economic returns. SoilGrids.org provides high-resolution static soil maps, which are different from dynamic flux information from farmers' fields. We would like readers of this study to focus on the science of the carbon cycle rather than concerns related to the market or government regulations.

References:

Goidts, E., B. Van Wesemael, and M. Crucifix. 2009. “Magnitude and Sources of Uncertainties in Soil

Organic Carbon (SOC) Stock Assessments at Various Scales.” *European Journal of Soil Science* 60 (5): 723–39.

Guan, Kaiyu, Zhenong Jin, Bin Peng, Jinyun Tang, Evan H. DeLucia, Paul C. West, Chongya Jiang, et al. 2023. “A Scalable Framework for Quantifying Field-Level Agricultural Carbon Outcomes.” *Earth-Science Reviews* 243 (104462): 104462.

Stanley, Paige, Jacob Spertus, Jessica Chiartas, Philip B. Stark, and Timothy Bowles. 2023. “Valid Inferences about Soil Carbon in Heterogeneous Landscapes.” *Geoderma* 430 (116323): 116323.

I (reviewer #2) was asked to evaluate the responses of the authors to the concerns of reviewer #3:

Point 2 – carbon capture in agriculture

Comment: The response of the authors mainly consists of removing the monetary estimates of the effects resolution on DeltaSOC estimates. The authors should show the direction of DeltaSOC estimates and how much soil carbon is increasing or decreasing compared to initial stocks (from SoilGrids.org for example). I came across this study with the *ecosys* model (<https://www.sciencedirect.com/science/article/pii/S0016706122005614>) which should be used for comparison of DeltaSOC estimates.

Response: We conducted comparisons of our estimated Δ SOC with SOC stock data sourced from SOC stocks derived from organic carbon density (OCD) in SoilGrids, as illustrated in Fig. R2. Our results reveal notable patterns, including SOC decreases in southern Minnesota, northern Iowa, and northeastern Illinois (primarily $<0.5\%/year$), while a substantial region of the southern US Midwest exhibits increasing SOC (primarily $<0.5\%/year$, but extending to $1\%/year$). This pattern predicted by KGML-ag-carbon (averaged at the county level in the states of Iowa, Illinois, and Indiana during 2000-2018) generally agrees with findings from the earlier study based on the *ecosys* model (Zhou et al. 2023), as depicted in Fig. R3 (all scales were converted to match those in Zhou et al. (2023)). However, it's worth noting that our model indicates larger values of SOC increases than those reported in the above-mentioned paper in the southern part of Iowa and Illinois. This difference can be attributed to the different scales of these two studies (the 250 m pixel scale in this study vs. county-scale results in Zhou et al. 2023 using gSSURGO majority soil types), along with the fine-tuning processes we implemented using eddy-covariance carbon flux data, NASS yield data, and the assimilation of remotely sensed GPP data, none of which were included in the *ecosys* modeling study. For instance, the remotely sensed GPP, as shown in Fig. R3 b, exhibits higher values in the southern regions of Iowa and Illinois compared to the GPP simulated by *ecosys* model in Fig. 7b of Zhou et al. (2023) (as Fig. R3 d here). This difference may be attributed to stronger mixed pixel effects caused by the relatively lower corn and soybean fractions.

Figure R3. Spatial distributions in Iowa, Illinois, and Indiana during 2000-2018 of KGML-ag-Carbon simulated county-level Δ SOC (a), input SLOPE GPP (b), and *ecosys* model (Zhou et al. 2023) simulated Δ SOC (c) and GPP (d).

Comment: The line of argumentation of the authors that SOC measurements will be too inaccurate to assess DeltaSOC, is insufficient. They should at least compare their rates of carbon sequestration or loss with literature values to show the realism of their DeltaSOC values. In Figure 4 of the above-mentioned study, DeltaSOC values are reported. How do they compare with the KGML approach? Figure 4d of Georgiou et al., 2022 (Georgiou K, Jackson RB, Vinduřková O et al. (2022) Global stocks and capacity of mineral-associated soil organic carbon. Nature Communications, 13.) also report some carbon accrual rates. How do these compare with DeltaSOC from KGML? A literature survey should give further data on DeltaSOC values.

Response: We appreciate your valuable input and suggestions. In response, we have conducted an extensive literature review to gather available Δ SOC measurements, and amassed data from 18 distinct sites that have multi-year SOC measurements for agroecosystems in the US Midwest after 2000. These findings are compiled in Table S1 (shown here as Table R1), which are sourced from nine studies (Al-Kaisi and Kwaw-Mensah 2020; Venterea et al. 2006; Ibrahim et al. 2018; Poffenbarger et al. 2017; Olson, Ebelhar, and Lang 2014; Varvel 2006; Khan et al. 2007; Jin et al. 2015; Schmer et al. 2014). The availability of Δ SOC data is much scarcer than that of static SOC data, such as the WoSIS soil profile database (Batjes, Ribeiro, and van Oostrum 2020) which is used to extrapolate SoilGrids. For instance, Fig. 4 of Zhou et al. 2023 is actually a comparison between two simulation results using different soil

databases as input. Fig. 4d of (Georgiou et al. 2022) and Table 4 in its supplement include cropland accrual rates from Europe (Alcántara et al. 2016; Mazzoncini et al. 2011), USA (Dick et al. 1998), and Brazil (Babujia et al. 2010; Boddey et al. 2010; Carvalho et al. 2010; Freixo et al. 2002; Jantalia et al. 2007; Marchão et al. 2009; Roscoe 2003). However, the scopes of all of those studies differ from ours in the temporal domain (i.e. most studies reported data from before the year 2000), the spatial domain (i.e. locations outside the US Midwest), and plant types (i.e. not corn or soybean agroecosystems). More importantly, all of the collected Δ SOC data pertain to plot-level (~10 m) experimental measurements that primarily focus on detecting the influences of management practices. Data from those plots often lack the requisite localized forcing data needed by our model. Consequently, we resort to utilizing field-level (250m) forcings such as remotely sensed GPP and reanalysis NLDAS weather forcing, which poses a scale mismatch when compared to the plot-level observations. To illustrate the scale mismatch (~10 m vs. 250 m), we selected two sites from different studies to compare the sizes of the experimental plots with the sizes of our predictions and real neighboring fields, as depicted in Fig. S15 a-b (as Fig. R4 a-b below). The test plots (blue boxes) in site no. 7 range from 9.1 to 27.4 m long and 18.3 to 34.6 m wide (Al-Kaisi and Kwaw-Mensah 2020), while plots in site no. 11 range from 4.6 to 6.1 m long and 15.2 to 19.8 m wide (Poffenbarger et al. 2017). These dimensions are notably smaller than our simulation grid (red boxes) or actual neighboring agricultural fields (green boxes).

Table R1. Sites used for validating SOC estimations using KGML-ag-Carbon.

Site no.	State	Location	Latitude	Longitude	Length	Plots	N	Period ^a	Reference ^b
1	Iowa	Kanawha	42.9368	-93.7987	13	15	2002-2014	Al-Kaisi, M. M., and Kwaw-Mensah, D., 2020	
2	Iowa	Sutherland	42.9304	-95.5273	13	15	2002-2014	Al-Kaisi, M. M., and Kwaw-Mensah, D., 2020	
3	Iowa	Nashua	42.9318	-92.5727	13	15	2002-2014	Al-Kaisi, M. M., and Kwaw-Mensah, D., 2020	
4	Iowa	Armstrong	41.3052	-95.1732	13	15	2002-2014	Al-Kaisi, M. M., and Kwaw-Mensah, D., 2020	
5	Iowa	Ames	41.9928	-93.6555	13	15	2002-2014	Al-Kaisi, M. M., and Kwaw-Mensah, D., 2020	
6	Iowa	Crawfordsville	41.2171	-91.5086	13	15	2002-2014	Al-Kaisi, M. M., and Kwaw-Mensah, D., 2020	
7	Iowa	McNay	40.9725	-93.4263	13	15	2002-2014	Al-Kaisi, M. M., and Kwaw-Mensah, D., 2020	
8	Minnesota	Rosemount	44.7508	-93.0746	6	18	2000-2005	Venterea, R. T., et al., 2006	
9	Iowa	Boone	42.0070	-93.7873	8	2	2008-2015	Ibrahim, M. A., et al., 2018	
10	Iowa	Central	42.0167	-93.7833	15	10	2000-2014	Poffenbarger, H. J., et al., 2017	
11	Iowa	Northwest	42.9292	-95.5403	15	14	2000-2014	Poffenbarger, H. J., et al., 2017	
12	Iowa	South	40.9667	-93.4167	15	14	2000-2014	Poffenbarger, H. J., et al., 2017	
13	Iowa	Southeast	41.1833	-91.4833	16	14	2000-2015	Poffenbarger, H. J., et al., 2017	
14	Illinois	Dixon Springs	37.4347	-88.6678	13	6	2000-2012	Olson, K., et al., 2014	
15	Nebraska	Mead	41.2476	-96.4695	3	21	2000-2002	Varvel, G. E., et al., 2006	
16	Illinois	Urbana	40.1047	-88.2261	6	9	2000-2005	Khan, S. A., et al., 2007	
17	Nebraska	Ithaca	41.0300	-96.0600	12	3	2000-2011	Jin, V. L., et al., 2015	
18	Nebraska	Ithaca	41.162	-96.4115	10	6	2001-2010	Schmer, M. R., et al., 2014	

^aSelected data periods include only years after 2000. If years before 2000 were presented but the year 2000 was missing, a linear interpolation was used to estimate SOC for the year 2000.

Figure R4. Illustration of scale mismatch between Δ SOC measurements and model estimates (a-b) and validation of KGML-ag-Carbon estimates of Δ SOC (c-d) and. Site details can be found in Table R1. In panels (a-b), the blue boxes indicate the plot measurement areas, the red boxes indicate the KGML-ag-Carbon estimate areas that encompass the plot measurement areas, and the green boxes indicate real agricultural fields in the US Midwest. In panels (c-d), box plots depict the distributions of observations or simulations from different plots.

We have also conducted validation using the gathered plot-level data. These results, as depicted in Fig. S15 c-d (shown here as Fig. R4 c-d) demonstrate that our modeled Δ SOC estimates generally align with the observed range in 10 out of the 18 cases (including sites no. 2, 3, 6, 8, 9, 10, 11, 16, 17, and 18), with the exception of sites having strong mixed pixel effects sourced from remotely sensed GPP (e.g. site no 7 in Fig. R4 a). This observation points to a potential overestimation trend in the southern regions of the US Midwest, as elucidated in Fig. R3. It's important to acknowledge that the main factor affecting model performance is the inherent scale mismatch between our predictions and the collected measurements. Additionally, variations in management practices and measurement uncertainties contribute to the observed discrepancies. We have revised the discussion in 4.1 by incorporating the validation results and augmented the Method section 5.8 with a comprehensive description of the related experiments. The revised discussion section 4.1 is shown below:

“The KGML-ag-Carbon employs a mass balance approach to estimate Δ SOC from NEE and yield, which are estimated by integrating all available data, including weather forcing, soil properties (which include

static SOC), crop type, and remotely sensed GPP. These inputs and predicted NEE and yield are well-validated by observations. This approach allows us to make the best use of existing data to estimate the regional Δ SOC at low cost and high resolution, even in the absence of field-level measurements. Validation efforts have been undertaken, focusing on sites within the US Midwest with SOC measurements in multiple years post-2000 (Fig. S15, Table S1, Methods 5.8). These validations demonstrate that our model's Δ SOC estimates fall within observed ranges in most cases. However, performance is constrained by three key factors: (1) Despite all collected Δ SOC data being at the plot-level scale (~10m), the absence of localized forcing data led us to employ field-level inputs like 250m GPP and weather data to drive the model; (2) Variations in management practices in each plot, such as tillage, fertilizer application, and crop rotation, further complicate field-level estimation; and (3) Uncertainty in field-level SOC arises from lab measurement error (up to 12%), spatial sampling error (up to 50%), and resampling error (up to 45%)^{42,43}, and can be exacerbated over extended time periods. Despite these challenges, our approach is valuable for mitigating carbon budget quantification errors, driven by its high resolution (250m) and accuracy (Fig. 2-4), all while maintaining a low computational cost. It's also worth noting that while the NEE, Reco, and crop yield in the KGML-ag-Carbon are well-constrained, the intermediate variables like Ra, Rh, and crop residue may still contain high uncertainty due to a lack of direct observational data constraint. These variables, however, are crucial for a better understanding of the underlying mechanisms. Therefore, the KGML-ag-Carbon also highlights the need for accurate field-level Δ SOC measurements to improve the reliability of Δ SOC quantification, and the need for accurate measurements of Ra, Rh, and crop residue to constrain the underlying processes.”

Revised Method 5.8 is listed below:

“We conducted an extensive literature review to gather available soil organic carbon (SOC) measurements in the US Midwest. This effort yielded data from 18 sites, each with multiple SOC measurements at plot level (~10m) after 2000, facilitating Δ SOC validation for the KGML-ag-Carbon model (Fig. S15, Table S1). The observed data such as bulk density, initial SOC stock at the top 30 cm, and rotation management were integrated into the input feature when applicable. Other inputs were directly derived from our 250m resolution regional database, such as weather forcings and GPP based on the sites’ geophysical locations. We have used an empirical equation¹⁷ to simulate the percentage fraction of SOC at different depths to total stock (assumed to be SOC in 0-100 cm), expressed as:

$$F_{SOC,Z} = -0.011 Z^2 + 2.029 * Z \quad (\text{Eq. 13})$$

Where $F_{SOC,Z}$ is the estimated SOC percentage between 0 to Z cm depth. This conversion factor aided in translating observed SOC values to the entire profile or to the top 30 cm in cases where depth-specific data was unavailable. It is worth noting that all of the collected Δ SOC data pertain to plot-level (~10m) experimental measurements that primarily focus on detecting the influences of management practices. Data from those plots often lack the requisite localized forcing data needed by our model. Consequently, we resort to utilizing field-level (250 m) forcings such as remotely sensed GPP and reanalysis NLDAS weather forcing, which poses a scale mismatch when compared to the plot-level observations. To illustrate this scale mismatch, we selected two sites from different studies^{18,19} to compare the sizes of the experimental plots with the sizes of our predictions and neighboring real fields, as depicted in Fig. S15 a-b.”

Comment: In section 5.8, the authors still use the carbon credit wording. Furthermore, it is not clear how the authors did the aggregation from 0.0025 degrees up to 1 degrees to quantify the error from coarse resolution. At first it was my impression that the error comes from aggregation in the covariates but from section 5.8 it seems like the error just comes from the way the 250m maps are aggregated. It should be possible to do the aggregation and conserving mass.

Response: The term “carbon credit” has been removed. The concept of the “absolute error” in our study is similar to the “basis risk” concept in agricultural insurance (Benami et al. 2021), representing the potential consequences resulting from mismatches between local-scale factors and the regional baseline. For example, if ΔSOC in a 250 m x 250 m field is +100 gC/m²/year, conventional methods that average values over a 50 km x 50 km region might yield +50 gC/m²/year or +150 gC/m²/year for that field. The absolute error remains constant at 50 gC/m²/year, regardless of whether the ΔSOC is +50 gC/m²/year or +150 gC/m²/year. Both scenarios, whether underestimated or overestimated, lead to flawed estimations and unfavorable outcomes like misguided management recommendations. We recognize the confusion that arose from the term “absolute error” and have revised the Method section 5.8 to explicitly describe the calculation of absolute error to avoid confusion. The revised Method 5.8 is listed below:

“We used the absolute error to quantify the uncertainties in NEE and ΔSOC quantification for individual fields in coarse-resolution carbon budget estimations (Fig. 5, Fig. S13). Specifically, we first regridded the 250-m-resolution products from KGML-ag-Carbon into a 0.0025-degree product to be used as a baseline for field-level absolute error quantification. Then we aggregated the 0.0025-degree products to various coarse resolutions, including 0.005, 0.01, 0.02, 0.05, 0.1, 0.2, 0.5, and 1 degrees. The calculation of the absolute error can be expressed as:

$$\text{Absolute Error } (Y_{res,i}) = |Y_{0.0025,i} - Y_{res,i}| \quad (\text{Eq. 12})$$

Where the *Absolute Error* ($Y_{res,i}$) represents the absolute error for produced variables (i.e. NEE or ΔSOC) at a 0.0025-degree pixel i when using a resolution of res . The $Y_{0.0025,i}$ represents the 0.0025-degree product of Y at pixel i , while $Y_{res,i}$ is the res -degree product of Y covering i pixel.”

Point 3 - management impacts

Comment: In their new text, the authors argue that “incorporating C-N interactions is complicated because the underlying processes are unclear and comprehensive measurements of both C- and N-related fluxes and soil states needed to validate any new model are lacking”. I think this is somewhat inaccurate since ecosys already has an N cycle and there are KGML variants on N₂O.

Response: We have revised the sentence as shown below:

“However, incorporating C-N interactions is challenging because comprehensive measurements of both C- and N-related fluxes and soil states, which are needed to validate any new model, are lacking, and vital inputs such as fertilizer applications and crop windows needed by regional-scale extrapolation of the model are absent.”

Point 4 – past extreme events in the estimates

Comment: The authors' argumentation makes sense. I would ask for more clarification on the test setup for the extreme years. Is this with training on corn yield at all counties except 2002, 2003, 2012? Does the training see the NEE in these years? How does KGML perform with regard to NEE in extreme years?

Response: We appreciate the attention to these details. Yes, the model training includes corn or soybean yield data from all counties except for the years 2002, 2003, and 2012. During this training process, the

model is not exposed to NEE data from these particular years. In response to your question about NEE performance during extreme years, we evaluated it using the US-NE1, US-NE2, and US-NE3 EC flux tower sites with 19 years of data spanning 2001 to 2019 (Fig. R5). We used data from 2001-2014 (excluding extreme years) for training, 2015-2019 for validation, and extreme years, 2002, 2003, and 2012 for testing. The results indicate that KGML-ag-Carbon can still outperform both pure ML and PB models for NEE and Reco flux predictions in extreme years.

Figure R5. A comparison of robust test results from a pure ML model (blue), the KGML-ag-Carbon model before finetuning (green, performance equivalent to the *ecosys* model), and the KGML-ag-Carbon model (red) for Reco (a, c) and NEE (b, d) predictions in extreme years. All models were constrained by remotely sensed GPP.

Point 5- overfitting the model

Comment: Here, I would ask the authors to clarify how they prevented leakage of the pre-training covariates to the test set? In other words, in a clean setup, you should pretrain on *ecosys* in a period that

will not be used for testing. As far as I see this has not been done and the periods of pretraining and testing are overlapping.

Response: We appreciate this insightful observation regarding potential data leakage concerns. We acknowledge that pretraining may not be a totally clean setup for data leakage as the inputs for pretraining (2000-2018 in three states) and finetuning (2000-2020 in the US Midwest including 12 states) may share similar climate information. However, we've taken meticulous steps to mitigate the impact of input leakage to the utmost extent, by employing distinctly different outputs (synthetic data vs. observations) and utilizing input data at varying scales (county-level vs. plot-level). Furthermore, to confirm the robustness of our KGML-ag-Carbon model, we have examined its performance in full factorial tests, as depicted in Fig. S11 (shown here as Fig. R6). Our findings demonstrate that even when compared to a pure ML model with pretraining (indicated in red), the KGML-ag-Carbon model (depicted in light brown) maintains superior performance in all scenarios.

Figure R6. Full-factorial tests for KGML-ag-Carbon. (a-b) Impacts of different components of the KGML-ag-Carbon model on annual corn yield prediction accuracy for models trained with either 5 or 40 counties out of 637 counties. (c-d) Impacts on the daily NEE flux prediction accuracy for models trained with either 1 or 7 sites out of 11 sites. ML represents machine learning, PRE represents pretraining using synthetic data, KGL represents the knowledge-guided loss, and STRU represents the knowledge-guided structure.

Comment: Furthermore, in Figure S9 g and S9 h, it seems like the test was on all EC flux towers - how did the authors perform the tests? You cannot include the training site into the test set. Here more clarification is needed if the authors prevented this. Similarly, it is not clear if Fig. 2 potentially has a similar problem.

Response: The reviewer is correct that training and test sets should not be mixed, and we did not do so. Instead, we followed a strict cross-validation (Stone 1974) and an ensemble approach to ensure the robustness of our testing methodology. Specifically, we divided the 11 EC flux tower sites into 6 groups based on the spatial distribution shown in Fig. S1 (referred to as Fig. R7 here with the grouping illustrated). For example, in an experiment with a training sample size of 7, we randomly selected 7 sites (repeatedly for ensemble purposes) from groups [1-5]/[1-4,6]/[1-3,5-6]/[1-2,4-6]/[1,3-6]/[2-6] for training, with the remaining group [6]/[5]/[4]/[3]/[2]/[1] being used for testing, respectively. The out-of-sample test outcomes from various models on different groups were consolidated in Fig. S9g and h and Fig. 2 c and d.

Figure R7. Study region (area highlighted in red lines) and the location of agroecosystem eddy-covariance sites used for the KGML-ag-Carbon model finetuning and validation (blue stars). The dark blue boxes represent the grouping methods used in cross-validation.

Detailed procedures are outlined in Method section 5.5. To eliminate any ambiguity, we have revised the captions for both Fig. 2 and Fig. S9. The updated caption for Fig. 2 is:

“**Figure 2.** ... Only out-of-sample test results from cross-validation ensembles are depicted here. Further details can be found in Method 5.5.”

The updated caption for Fig. S9 is:

“Figure S9. ... A cross-validation method together with ensemble methods are used for robustness tests for fluxes using 11 flux tower sites (g-l). Details can be found in Method 5.5.”

References:

- Alcántara, Viridiana, Axel Don, Reinhard Well, and Rolf Nieder. 2016. “Deep Ploughing Increases Agricultural Soil Organic Matter Stocks.” *Global Change Biology* 22 (8): 2939–56.
- Al-Kaisi, Mahdi M., and David Kwaw-Mensah. 2020. “Quantifying Soil Carbon Change in a Long-term Tillage and Crop Rotation Study across Iowa Landscapes.” *Soil Science Society of America Journal. Soil Science Society of America* 84 (1): 182–202.
- Babujia, L. C., M. Hungria, J. C. Franchini, and P. C. Brookes. 2010. “Microbial Biomass and Activity at Various Soil Depths in a Brazilian Oxisol after Two Decades of No-Tillage and Conventional Tillage.” *Soil Biology & Biochemistry* 42 (12): 2174–81.
- Batjes, Niels H., Eloi Ribeiro, and Ad van Oostrum. 2020. “Standardised Soil Profile Data to Support Global Mapping and Modelling (WoSIS Snapshot 2019).” *Earth System Science Data* 12 (1): 299–320.
- Benami, Elinor, Zhenong Jin, Michael R. Carter, Aniruddha Ghosh, Robert J. Hijmans, Andrew Hobbs, Benson Kenduyiwo, and David B. Lobell. 2021. “Uniting Remote Sensing, Crop Modelling and Economics for Agricultural Risk Management.” *Nature Reviews. Earth & Environment* 2 (2): 140–59.
- Boddey, Robert M., Claudia P. Jantalia, Paulo C. Conceição, Josileia A. Zanatta, Cimílio Bayer, João Mielniczuk, Jeferson Dieckow, et al. 2010. “Carbon Accumulation at Depth in Ferralsols under Zero-till Subtropical Agriculture.” *Global Change Biology* 16 (2): 784–95.
- Cao, Q., J. Li, G. Wang, D. Wang, Z. Xin, H. Xiao, and K. Zhang. 2021. “On the Spatial Variability and Influencing Factors of Soil Organic Carbon and Total Nitrogen Stocks in a Desert Oasis Ecotone of Northwestern China.” *Catena* 206 (November): 105533.
- Carvalho, João Luís Nunes, Guilherme Siva Raucci, Carlos Eduardo P. Cerri, Martial Bernoux, Brigitte Josefina Feigl, Flávio Jesus Wruck, and Carlos Clemente Cerri. 2010. “Impact of Pasture, Agriculture and Crop-Livestock Systems on Soil C Stocks in Brazil.” *Soil and Tillage Research* 110 (1): 175–86.
- Dick, W. A., R. L. Blevins, W. W. Frye, S. E. Peters, D. R. Christenson, F. J. Pierce, and M. L. Vitosh. 1998. “Impacts of Agricultural Management Practices on C Sequestration in Forest-Derived Soils of the Eastern Corn Belt.” *Soil and Tillage Research* 47 (3-4): 235–44.
- Freixo, Alessandra A., Pedro Luiz O. de A. Machado, Henrique P. dos Santos, Carlos A. Silva, and Francisco de S. Fadigas. 2002. “Soil Organic Carbon and Fractions of a Rhodic Ferralsol under the Influence of Tillage and Crop Rotation Systems in Southern Brazil.” *Soil and Tillage Research* 64 (3-4): 221–30.
- Georgiou, Katerina, Robert B. Jackson, Olga Vindušková, Rose Z. Abramoff, Anders Ahlström, Wenting Feng, Jennifer W. Harden, et al. 2022. “Global Stocks and Capacity of Mineral-Associated Soil Organic Carbon.” *Nature Communications* 13 (1): 3797.
- Ibrahim, Mostafa A., Teresita Chua-Ona, Matt Liebman, and Michael L. Thompson. 2018. “Soil Organic Carbon Storage under Biofuel Cropping Systems in a Humid, Continental Climate.” *Agronomy Journal* 110 (5): 1748–53.
- Jantalia, Cláudia P., Dimas V. S. Resck, Bruno J. R. Alves, Lincoln Zotarelli, Segundo Urquiaga, and Robert M. Boddey. 2007. “Tillage Effect on C Stocks of a Clayey Oxisol under a Soybean-Based Crop Rotation in the Brazilian Cerrado Region.” *Soil and Tillage Research* 95 (1-2): 97–109.
- Jin, Virginia L., Marty R. Schmer, Brian J. Wienhold, Catherine E. Stewart, Gary E. Varvel, Aaron J. Sindelar, Ronald F. Follett, Robert B. Mitchell, and Kenneth P. Vogel. 2015. “Twelve Years of Stover Removal Increases Soil Erosion Potential without Impacting Yield.” *Soil Science Society of America Journal. Soil Science Society of America* 79 (4): 1169–78.

- Khan, S. A., R. L. Mulvaney, T. R. Ellsworth, and C. W. Boast. 2007. "The Myth of Nitrogen Fertilization for Soil Carbon Sequestration." *Journal of Environmental Quality* 36 (6): 1821–32.
- Marchão, Robélio Leandro, Thierry Becquer, Didier Brunet, Luiz Carlos Balbino, Lourival Vilela, and Michel Brossard. 2009. "Carbon and Nitrogen Stocks in a Brazilian Clayey Oxisol: 13-Year Effects of Integrated Crop–livestock Management Systems." *Soil and Tillage Research* 103 (2): 442–50.
- Mazzoncini, Marco, Tek Bahadur Sapkota, Paolo Barberi, Daniele Antichi, and Rosalba Risaliti. 2011. "Long-Term Effect of Tillage, Nitrogen Fertilization and Cover Crops on Soil Organic Carbon and Total Nitrogen Content." *Soil and Tillage Research* 114 (2): 165–74.
- Olson, Kenneth, Stephen A. Ebelhar, and James M. Lang. 2014. "Long-Term Effects of Cover Crops on Crop Yields, Soil Organic Carbon Stocks and Sequestration." *Open Journal of Soil Science* 04 (08): 284–92.
- Poffenbarger, Hanna J., Daniel W. Barker, Matthew J. Helmers, Fernando E. Miguez, Daniel C. Olk, John E. Sawyer, Johan Six, and Michael J. Castellano. 2017. "Maximum Soil Organic Carbon Storage in Midwest U.S. Cropping Systems When Crops Are Optimally Nitrogen-Fertilized." *PloS One* 12 (3): e0172293.
- Roscoe, R. 2003. "Tillage Effects on Soil Organic Matter in Density Fractions of a Cerrado Oxisol." *Soil and Tillage Research* 70 (2): 107–19.
- Schmer, M. R., V. L. Jin, B. J. Wienhold, G. E. Varvel, and R. F. Follett. 2014. "Tillage and Residue Management Effects on Soil Carbon and Nitrogen under Irrigated Continuous Corn." *Soil Science Society of America Journal. Soil Science Society of America* 78 (6): 1987–96.
- Stone, M. 1974. "Cross-Validatory Choice and Assessment of Statistical Predictions." *Journal of the Royal Statistical Society* 36 (2): 111–33.
- Varvel, G. E. 2006. "Soil Organic Carbon Changes in Diversified Rotations of the Western Corn Belt." *Soil Science Society of America Journal. Soil Science Society of America* 70 (2): 426–33.
- Venterea, Rodney T., John M. Baker, Michael S. Dolan, and Kurt A. Spokas. 2006. "Carbon and Nitrogen Storage Are Greater under Biennial Tillage in a Minnesota Corn–soybean Rotation." *Soil Science Society of America Journal. Soil Science Society of America* 70 (5): 1752.
- Zhou, Wang, Kaiyu Guan, Bin Peng, Andrew Margenot, Dokyoung Lee, Jinyun Tang, Zhenong Jin, et al. 2023. "How Does Uncertainty of Soil Organic Carbon Stock Affect the Calculation of Carbon Budgets and Soil Carbon Credits for Croplands in the U.S. Midwest?" *Geoderma* 429 (116254): 116254.

Reviewers' Comments:

Reviewer #1:

Remarks to the Author:

My concerns have been addressed and I support the manuscript for publications.

Reviewer #2:

Remarks to the Author:

I would like to thank the authors for addressing most of my comments very well and comprehensively. I only note down in the following where I still see weaknesses – some have to be decided by the editors ultimately.

Comment: Realism of Δ SOC

I very much appreciate the effort and diligence that the authors put into preparing the response to my comment. The authors added a discussion on increases and declines of SOC and the realism of this. I have a couple critical points that arise from this:

“The lack of EC flux tower data (blue stars in Fig. S1) limits the constraining of NEE predictions”

- The argument for KGML was to combine ML and process-based models for better results in regions where data is scarce. In this argument, you say we would need flux towers there to get trustworthy estimates and the ML-version of ecosys is not enough.

“The mass balance approach includes crop residues in Δ SOC estimation and so far there’s no good data to separate crop residues”

- I was asking about this point earlier. It is not clear what you mean by “separat(ing) crop residues”. Separate into which fractions and how would this affect the KGML estimates?

- Is it guaranteed that crop residues such as straw stay on the field? Is this what you assumed for KGML, or is there some data on how much crop residue stays on the field?

While these added discussions are valuable and open, the reader would want to know why KGML predicts increases in some regions and losses in others. Is it management, climate or other factors?

It is strange to present the DeltaSOC only in Figure S14, deep in the supplement. You argue that the main point of the manuscript is the aggregation errors from high resolution to coarse resolution. However, these are merely a result of the patterns in high-resolution DeltaSOC in Figure S14. The spatial aggregation finding is only a secondary result in my mind. First, you have to discuss the fine-resolution DeltaSOC product; then, you can discuss spatial aggregation. This should be done with figures and text in the main manuscript.

The spatial aggregation analysis is somewhat trivial. You have high-resolution data and could do a mass-conserving aggregation instead of conventional averaging aggregation (which was done?). I only see the point of mismatch if the forcing data and covariates for upscaling were coarse and you wanted to quantify this error. The way you present Figure 5, is somewhat of an artificial experiment since there is not much reason to do this aggregation. Instead, I would like you to focus on why certain regions gain or lose carbon (first step). Then it makes sense to run KGML with coarse resolution forcing to see what we lose in detail (second step) and compare it to conventional aggregation (third step, Figure 5).

You still stick with the basis risk concept from agricultural insurance: ‘The concept of the “absolute error” in our study is similar to the “basis risk” concept in agricultural insurance (Benami et al. 2021), representing the potential consequences resulting from mismatches between local-scale factors and the regional baseline’

As described above, I think simply aggregating from fine to coarse resolution is a misleading way to look at this. Fine-scale information is available in your case, so there is no true coarse resolution comparison. Second, you still stick with insurance and financial logic – it is ultimately the task of the editors if this is within the scope of Nature Communications. I prefer insights into why KGML predicts carbon losses and gains in different regions.

Comment: openness of code

Thanks for the clarification. Here, I see the responsibility of the editors to ensure that this indeed is happening as described at the same time when the paper is published. Ideally, there would be a way for reviewers to judge this now.

Comment: Point 5- overfitting the model

Thanks for these detailed responses. You may consider incorporating parts of this response into the material and methods or supplement to ensure that other readers directly find the answers to these questions in the manuscript.

Response Letter

Please be aware of the formatting of all responses:

1. Reviewer comment in **black**, response in **blue** and quotation from the main text in **red**;
2. The referred sections are based on the clean version of the revised manuscript, not the old or track-change version.

Response to Reviewer #2

Comment: I would like to thank the authors for addressing most of my comments very well and comprehensively. I only note down in the following where I still see weaknesses – some have to be decided by the editors ultimately.

Response: We greatly appreciate your feedback and suggestions. In this revision, we have taken your suggestions, removed the simple aggregation-based “absolute error” experiments, and added more rigorous analysis to (1) attribute the KGML-ag-Carbon predicted carbon increase/decrease to climate and soil factors in the US Midwest, and (2) demonstrate the high-resolution quantification benefit by comparing KGML-ag-Carbon simulations at 0.0025-degree vs 0.5-degree resolutions. We have addressed your comments to the best we can and hope you find more satisfaction with this version. Please see our detailed point-by-point response below.

Comment: Realism of Δ SOC

I very much appreciate the effort and diligence that the authors put into preparing the response to my comment. The authors added a discussion on increases and declines of SOC and the realism of this. I have a couple critical points that arise from this:

“The lack of EC flux tower data (blue stars in Fig. S1) limits the constraining of NEE predictions”

- The argument for KGML was to combine ML and process-based models for better results in regions where data is scarce. In this argument, you say we would need flux towers there to get trustworthy estimates and the ML-version of ecosys is not enough.

Response: Thank you for raising this point. With this argument, we meant to say that KGML predictions can be further improved when flux tower data is available during the finetuning process. This is different from “KGML needs flux towers there to get trustworthy estimates” because “trustworthy” doesn’t mean that KGML predictions need to be perfect. In Fig. 2d (referred here as Fig. R1), our results have already demonstrated that KGML can better generalize to out-of-sample predictions than either PB or ML models. However, incorporating flux tower data can further improve the KGML performance in data-scarce situations. To avoid confusion, we have extended the explanation in the supplemental text S5:

“Furthermore, the absence of EC flux tower data in the southern region (indicated by blue stars in Fig. S1) may limit the performance of our model. While KGML-ag-Carbon has demonstrated better performance in data-scarce scenarios compared to traditional PB and pure ML models (Fig. 2; Fig. S9), incorporating flux tower data could enhance the reliability of our estimates. For instance, Fig. 2d highlights that constraining NEE with one site of flux tower data can improve the out-of-sample prediction accuracy, with the mean R-squared value improved from 0.91 (without data constraints) to 0.93.”

Figure R1. The performance of the *ecosys* model (green boxes), pure ML model (blue boxes), and KGML-ag-Carbon (red boxes) using different observed sample sizes for model training.

Comment: “The mass balance approach includes crop residues in Δ SOC estimation and so far there’s no good data to separate crop residues”

- I was asking about this point earlier. It is not clear what you mean by “separat(ing) crop residues”. Separate into which fractions and how would this affect the KGML estimates?

- Is it guaranteed that crop residues such as straw stay on the field? Is this what you assumed for KGML, or is there some data on how much crop residue stays on the field?

Response: Thank you for pointing out this confusion. The term "separate" in this context is not accurate and we meant to say “excluding the undecomposed crop residue from the estimated Δ SOC”. Our mass balance approach, Δ SOC = -NEE - Yield, calculates Δ SOC as a combination of turnovered SOC (humus) and undecomposed crop residue. In most Δ SOC measurements, only the turnovered SOC is considered. We didn’t exclude crop residue because currently there is no extensive dataset available to support such exclusion at a large scale and it is not guaranteed the crop residue will be undisturbed. To mitigate the influence of undecomposed crop residue in this study, we utilized multi-year averages of Δ SOC estimations from the mass balance approach and assumed that residue is not removed during the whole period. We acknowledge that this way may cause uncertainties of Δ SOC in a short study period and/or crop residue being removed, but it is the best we can do for now.

To clarify the KGML’s assumptions, we have discussed this point in discussion section 4.1.

“Our estimated Δ SOC represents a combination of crop residue and humus, while the majority of measurements typically focus on humus content.”

Then we have extended explanations in supplemental text S5.

“Moreover, our approach employs a mass balance method to estimate Δ SOC, which includes crop residues that may not be converted into SOC within the specified period. So far, there is no available data to effectively exclude crop residues from the estimation. Specifically, Δ SOC = -NEE - Yield = turnovered SOC (humus) + undecomposed crop residue. Unfortunately, there is a lack of comprehensive and large-scale data to effectively distinguish between turnovered SOC and the remaining crop residue. Using a multi-year average for Δ SOC estimation with the assumption that no residue will be removed, the term undecomposed crop residue will be minimized.”

Comment: While these added discussions are valuable and open, the reader would want to know why KGML predicts increases in some regions and losses in others. Is it management, climate or other factors?

Response: This is a good question but not so easy to answer. In response, we have (1) added spatial

distributions of the estimated Δ SOC into the main text to help visually examine the patterns, presented as Fig. 5a-c (referred to here as Fig. R2); and (2) added correlation analysis and multiple linear regression to attribute the predicted carbon increase/decrease to climate and soil factors (Fig. S14; referred to here as Fig. R3). Relevant discussions have been added to section 4.1 and supplemental text S5. In summary, our findings suggest that KGML-ag-Carbon predicts carbon losses in northern regions primarily due to relatively colder and drier conditions, and higher SOC stock levels, compared with southern regions. Contrastingly, warmer and wetter conditions and lower SOC stock mainly drive the predicted carbon increase in the south. The results also indicate that soil factors may play a more dominant role than climate in explaining the spatial patterns of the Δ SOC in the US Midwest. The findings are consistent with previous studies (Zhou et al., 2021 and 2023). For a more detailed explanation, please refer to the figures and the quoted texts from the revised manuscript below.

The revised part in discussion 4.1 is listed below:

“The high-resolution Δ SOC estimations reveal that the majority of changes fall within the range of -0.5% to 0.5% C/year (86%), which aligns with the observed ranges in other experimental studies^{36–41} (Fig. 5c). Notable patterns include a decline in SOC across southern Minnesota, northern Iowa, and northern eastern Illinois, as well as an increase in the southern US Midwest. These patterns are primarily influenced by soil factors (43% variance explained) and climate factors (11% variance explained). Relatively colder, drier conditions (fewer carbon inputs into the soil) and higher SOC stock levels (larger Rh) contribute to carbon losses in northern regions (Fig. S14). More detailed assessments of the Δ SOC patterns are available in Text S5.”

Figure R2. (a) The distribution of the Δ SOC estimation during 2000-2020 derived from the mass balance approach using KGML-ag-Carbon estimated 0.0025-degree-resolution carbon budgets; (b) Percentage fraction of the estimated Δ SOC in (a) compared to the SoilGrids SOC stock, limited to regions with over 50% corn or soybean planting; and (c) Histogram distribution of percentage fractions in (b).

Figure R3. The assessments of correlations between KGML-ag-Carbon input variables and GPP, NEE, Ra, Rh, Yield, Reco, Residue, and Δ SOC. (a) Pearson's correlation coefficients between each pair of variables; and (b) Spatial distributions of input variables. Climate factors including surface downward shortwave radiation (RADN), maximum air temperature (TMAX_AIR), the difference between the maximum and minimum air temperature (TDIF_AIR), maximum humidity (HMAX_AIR), the difference between the maximum and minimum humidity (HDIF_AIR), wind speed (WIND), and precipitation (PRECN). Soil factors include bulk density (TBKDS), sand content (TCSAND), silt content (TCSILT), water content at field capacity (TFC), water content at wilting point (TWP), saturated hydraulic conductivity (TKSat), soil organic carbon (TSOC), pH (TPH), and cation exchange capacity (TCEC). Each variable x was Z-normalized to standard normal distribution $N(0,1)$ using the equation $Z = (x - \mu)/\sigma$, where Z is the normalized variable; μ is the mean value of x ; and σ is the standard deviation of x .

A more thorough discussion is included in supplemental text S5:

“We implemented a mass balance approach to estimate Δ SOC from KGML-ag-Carbon estimated NEE and crop yield. Generally, Fig. S14a indicates relatively stronger correlations between NEE and input variables, compared with yield. Therefore, the spatial patterns of Δ SOC are likely shaped more by NEE. Since we define the positive direction of NEE as from the soil to the atmosphere, a negative correlation coefficient signifies a decrease in atmospheric carbon but may also indicate an increase in soil carbon. We have further decomposed the input factors into two categories: climate factors and soil factors, to better explain the spatial pattern of Δ SOC.

Climate factors influence the Δ SOC mainly through influencing the GPP (Fig. S14, Fig. 5). The GPP's heatmap patterns closely mirror that of Δ SOC in Fig. S14a. Regions in the southern areas tend to experience relatively higher radiation, temperature, precipitation, and humidity, especially when compared to their northern counterparts. As observed in Zhou et al. (2021)³, when air temperature

(strongly correlated with radiation as presented in Fig. S14b) is less than 30°C, GPP increases quickly along with the temperature increase. Meanwhile, the air humidity (strongly correlated with precipitation as presented in Fig. S14b) influences GPP through affecting crop stomatal conductance and the increase of humidity (when below 10hPa) will increase the GPP accordingly. Consequently, the southern regions exhibit higher GPP, contributing to an increased influx of carbon input into the soil (crop residue in Fig. S14a). This phenomenon potentially results in higher Δ SOC values in these areas. Furthermore, our observations indicate a strong positive correlation between Ra and specific climate parameters (i.e. radiation, temperature, humidity, and precipitation). However, it's crucial to note that the total Reco may not exhibit a correspondingly strong response (Fig. S14a). This could partly be attributed to the negative correlation observed between Rh and certain climate factors. This negative relationship may stem from the shared patterns between climate factors and soil properties such as TSOC, TFC, and TWP (relatively higher in the north and lower in the south). Nonetheless, it's important to acknowledge that these discrepancies could also arise from the partitioning of Reco into Ra and Rh components. The KGML-ag-Carbon, which was constrained using Reco, necessitates further validation and examination using the partitioned data of Ra and Rh to elucidate these correlations more definitively.

Soil factors influence the Δ SOC primarily by affecting Rh. We find that Rh exhibits heatmap patterns that are nearly the inverse of that of Δ SOC (Fig. S14a). Higher levels of TSOC tend to correlate with optimal soil conditions characterized by increased substrate availability for microbes and improved drainage (e.g. higher TFC and TWP). These favorable conditions then drive up Rh rates, leading to reduced NEE in the soil and ultimately contributing to a decrease in Δ SOC (Zhou et al. 2023)⁴. The northern regions exhibit higher levels of TSOC, TFC, and TWP when compared to their southern counterparts. Consequently, these regions often experience a net loss of carbon.”

“In summary, the spatial patterns of the estimated Δ SOC are mainly governed by regional climate and soil factors. The relatively warmer, wetter climate in the south, coupled with lower TSOC, predominantly contributes to carbon gain. Conversely, the northern regions, characterized by colder, drier conditions and higher TSOC levels, are primarily responsible for carbon loss. We conducted multiple linear regression analyses to further examine the total influence of climate or soil factors on Δ SOC. The R-squared values were 0.11 and 0.43 for climate and soil factors, respectively, indicating that the soil may play a relatively more dominant role in explaining the spatial pattern of Δ SOC. It's worth noting that this study doesn't delve into the influence of management practices, given its historical simulation approach employing predefined crop types and rotations. Evaluating various management practices would necessitate substantial research efforts and remains a prospective avenue for future investigations, which has been discussed with more detail in section 4.3.”

The methods used to generate spatial maps and histograms and conduct spatial pattern analysis have been added in method section 5.8:

“To generate 0.0025-degree-resolution Δ SOC estimations for the US Midwest (Fig. 5a-c), we employed the mass balance equation Δ SOC = - NEE - crop yield^{2,16,17} over the period spanning 2000 to 2020. Specifically, we regridded the 250-m-resolution NEE and crop yield estimations from KGML-ag-Carbon into 0.0025-degree estimations for use in the mass balance equation. To minimize the influence of surface crop residues (undecomposed), which do not contribute to Δ SOC but are counted as part of our Δ SOC estimations through the mass balance approach, we selected the 20-year averaged value of Δ SOC. We then focused on regions where more than 50% of the area was planted with corn or soybean crops (Fig.

S13a). The Δ SOC values were converted to percentage fractions (Fig. 5b) using ML-based SOC stocks derived from the SoilGrids¹⁸ (Fig. S13b). Specifically, we used corn and soybean fractions from CDL and CSDL data (Fig. S13a) to identify regions exclusively for corn and soybean agroecosystems (total fraction > 0.5). This alignment with our model's current training scope helped reduce the mixed pixel effect resulting from inputted remotely sensed GPP data from other ecosystems. The SoilGrids SOC stock (Fig. S13b) was derived from organic carbon density (OCD) in each layer of the 200-cm soil depth at a 250-m resolution¹⁸.

To attribute the spatial patterns of estimated Δ SOC, we conducted Pearson correlation analyses between the input variables (including seven climate variables and nine soil variables) and the target variables (including GPP, NEE, Ra, Rh, Yield, Reco, Residue, and Δ SOC) (Fig. S14a; Text S5). In our approach, each variable was temporally aggregated to a 20-year scale and Z-normalized using Eq. 11. The Residue variable was computed as the GPP - Ra - Yield, representing the net carbon return from plants to the soil. While GPP served as an input to the KGML-ag-Carbon model, we included it as a target variable in the correlation assessment due to its pivotal role in the carbon cycle. In addition, we have conducted multiple linear regression to assess the total influence of climate factors and soil factors on Δ SOC. For more comprehensive explanations, please refer to Text S5.”

References:

Zhou, W. et al. Quantifying carbon budget, crop yields and their responses to environmental variability using the *ecosys* model for U.S. Midwestern agroecosystems. *Agric. For. Meteorol.* 307, 108521 (2021).
Zhou, W. et al. How does uncertainty of soil organic carbon stock affect the calculation of carbon budgets and soil carbon credits for croplands in the U.S. Midwest? *Geoderma* 429, 116254 (2023).

Comment: It is strange to present the DeltaSOC only in Figure S14, deep in the supplement. You argue that the main point of the manuscript is the aggregation errors from high resolution to coarse resolution. However, these are merely a result of the patterns in high-resolution DeltaSOC in Figure S14. The spatial aggregation finding is only a secondary result in my mind. First, you have to discuss the fine-resolution DeltaSOC product; then, you can discuss spatial aggregation. This should be done with figures and text in the main manuscript.

The spatial aggregation analysis is somewhat trivial. You have high-resolution data and could do a mass-conserving aggregation instead of conventional averaging aggregation (which was done?). I only see the point of mismatch if the forcing data and covariates for upscaling were coarse and you wanted to quantify this error. The way you present Figure 5, is somewhat of an artificial experiment since there is not much reason to do this aggregation. Instead, I would like you to focus on why certain regions gain or lose carbon (first step). Then it makes sense to run KGML with coarse resolution forcing to see what we lose in detail (second step) and compare it to conventional aggregation (third step, Figure 5).

Response: We fully respect your opinion that “The spatial aggregation finding is only a secondary result in my mind”, but we would like to note that this type of information is timely and important for certain group of readers. There are indeed policy-relevant discussions in the US about at what spatial scale we can and/or need to make trustable carbon prediction, some argue for field level and some believe even county scale (roughly 0.5x0.5 deg) is sufficient. So this spatial aggregation analysis will provide a reference for those policy-relevant debates.

That being said, we very much appreciate your constructive suggestion this time by proposing an alternative way to do spatial aggregation analysis, which is indeed a better way that we haven't thought

about before. So following your suggestions, we (1) moved the spatial distribution of estimated Δ SOC to the main text as Figure 5a-c (referred to as Fig. R2 in our response); and (2) we conducted more rigorous experiments to examine the benefits of high-resolution carbon budget quantification (Fig. 5d-f, referred to as Fig. R4). Specifically, we ran the KGML-ag-Carbon model at a coarse resolution of 0.5 degrees as suggested (Fig. R4a), and analyzed the detailed losses between high-resolution and coarse-resolution products (Fig. R4b-c) to better illustrate the aggregation errors. Accordingly, we have revised the discussion in section 4.1 and added methods in section 5.8. More high-resolution benefit analysis results for other variables including GPP, NEE, Ra, Rh, and crop yield can be found in Fig. S15.

Figure R4. The distribution of the estimated Δ SOC during 2000-2020 at coarse resolution and the demonstration of the impact of coarse resolution on Δ SOC. (a) Δ SOC estimation derived from the mass balance approach using KGML-ag-Carbon estimated 0.5-degree-resolution carbon budgets; (b) Spatial distribution of differences between coarse-resolution (0.5 degrees) and fine-resolution (0.0025 degrees) Δ SOC estimations, relative to the SoilGrids SOC stock and limited to regions with over 50% corn or soybean planting; and (c) The histogram distribution of the differences in (b).

The revised part of the abstract is listed below:

“... our high-resolution approach can enrich overall 86% details for Δ SOC quantification than the conventional coarse-resolution approach.”

The revised discussion in section 4.1 is listed below:

“To underscore the necessity of high-resolution carbon budget and crop yield quantification, we have generated 0.0025-resolution Δ SOC estimations using the mass balance approach with KGML-ag-Carbon-predicted 250m-resolution NEE and crop yield data from 2000 to 2020 to represent the high-resolution product (Fig. 5a-c, method 5.8). For comparison, we have also generated 0.5-degree Δ SOC estimations using a similar approach, implementing KGML-ag-Carbon at a 0.5-degree resolution to represent the coarse-resolution product (Fig. 5d-f, method 5.8). ... Comparing coarse and high-resolution Δ SOC estimations shows notable differences (overall NRMSE = 86%) due to loss of detail (e.g. hot/cold spots) and relatively stronger mixed pixel effects in 0.5-degree pixels (Fig. 5d-e). The histogram distribution (Fig. 5f) indicates a difference ranging from -0.1 (10% quantiles) to 0.9 (90% quantile) %/year between coarse- and high-resolution estimations. This difference cannot be neglected when compared to the high-resolution Δ SOC histogram distributions (Fig. 5c). More detailed results regarding the differences between high-resolution and coarse-resolution GPP inputs, as well as Ra, Rh, NEE, and crop yield qualifications, are provided in Fig. S15.”

Comment: You still stick with the basis risk concept from agricultural insurance: ‘The concept of the “absolute error” in our study is similar to the “basis risk” concept in agricultural insurance (Benami et al. 2021), representing the potential consequences resulting from mismatches between local-scale factors and the regional baseline’

As described above, I think simply aggregating from fine to coarse resolution is a misleading way to look at this. Fine-scale information is available in your case, so there is no true coarse resolution comparison. Second, you still stick with insurance and financial logic – it is ultimately the task of the editors if this is within the scope of Nature Communications. I prefer insights into why KGML predicts carbon losses and gains in different regions.

Response: Thank you again for framing the point in a constructive way. We have now dropped the concept of basis risk or absolute error, and re-did the spatial aggregation analysis following in the way you have suggested.

We also agree with you to provide more insights into why KGML predicts carbon gains and losses in different regions, and we have added new analysis as detailed earlier. We respect your opinion about the financial logic, and hope you can understand what we have explained in the last point on why we wanted to keep at least some policy-relevant discussion beyond pure scientific results.

Comment: openness of code

Thanks for the clarification. Here, I see the responsibility of the editors to ensure that this indeed is happening as described at the same time when the paper is published. Ideally, there would be a way for reviewers to judge this now.

Response: Yes, to ensure transparency and facilitate the review process, we have compiled the pertinent code and sample data for all experiments presented in the manuscript. These materials have been shared with the editor on a confidential basis and will become accessible should the paper be published.

Comment: Point 5- overfitting the model

Thanks for these detailed responses. You may consider incorporating parts of this response into the material and methods or supplement to ensure that other readers directly find the answers to these questions in the manuscript.

Response: Thank you for your suggestion. We have added this information to the supplemental text S4 to ensure the audience can find the answers.

The added discussion in supplemental text S4 is listed below:

“It should be noted that our training configuration may not be a totally clean setup for data leakage, as the inputs for pretraining (2000-2018 in 31 states) and fine-tuning (2000-2020 in the US Midwest including 12 states) may share similar climate input information in three states. However, we've taken meticulous steps to mitigate the impact of input leakage to the utmost extent on robustness analysis (as illustrated in Fig. 2), by employing distinctly different outputs (synthetic data vs. observations) and utilizing input data at varying scales (county-level vs. pixel-level). Furthermore, to confirm the robustness of our KGML-ag-Carbon model, we have examined its performance in full factorial tests, as depicted in Fig. S11. Our findings demonstrate that even compared to a pure ML model with pretraining (indicated in red), the KGML-ag-Carbon model (depicted in light brown) maintains superior performance in all scenarios.”

Reviewers' Comments:

Reviewer #2:

Remarks to the Author:

I would like to thank the authors for their comprehensive revisions! I think the manuscript is now in a stage that it can be published.

I hope the source code of the trained model will be published in adequate form.

One minor point that should be added - how was the aggregation done, e.g. mean, mode or median as aggregation function? Please add this information to L957. To my mind, this could be important information to judge why the model overestimates at coarser input resolution.

Response Letter

Please be aware of the formatting of all responses:

1. Reviewer comment in **black**, response in **blue** and quotation from the main text in **red**;
2. The referred sections are based on the clean version of the revised manuscript, not the old or track-change version.

Response to Reviewer #2

Comments: I would like to thank the authors for their comprehensive revisions! I think the manuscript is now in a stage that it can be published.

I hope the source code of the trained model will be published in adequate form.

Response: Thank you for your positive feedback. As suggested, our trained model's source code and data are now publicly accessible on Zenodo.

Comments: One minor point that should be added - how was the aggregation done, e.g. mean, mode or median as aggregation function? Please add this information to L957. To my mind, this could be important information to judge why the model overestimates at coarser input resolution.

Response: We appreciate your attention to detail in our manuscript. To address your query, we have revised the sentence in Method 5.8:

“To achieve this, we employed a mean aggregation approach for each input variable, converting from 250-meter resolution to 0.5-degree resolution.”